# An excitatory lateral hypothalamic circuit orchestrating pain behaviors in mice

Justin N Siemian[1], Miguel A Arenivar[1], Sarah Sarsfield[1], Cara B Borja[1], Lydia J Erbaugh[1], Andrew L Eagle[2], Alfred J Robison[2], Gina Leinninger[2,3], Yeka Aponte[1,4]*

[1]Neuronal Circuits and Behavior Unit, National Institute on Drug Abuse Intramural Research Program, National Institutes of Health, Baltimore, United States; [2]Department of Physiology, Michigan State University, East Lansing, United States; [3]Institute for Integrative Toxicology at Michigan State University, East Lansing, United States; [4]The Solomon H. Snyder Department of Neuroscience, Johns Hopkins University School of Medicine, Baltimore, United States

**Abstract** Understanding how neuronal circuits control nociceptive processing will advance the search for novel analgesics. We use functional imaging to demonstrate that lateral hypothalamic parvalbumin-positive (LH[PV]) glutamatergic neurons respond to acute thermal stimuli and a persistent inflammatory irritant. Moreover, their chemogenetic modulation alters both pain-related behavioral adaptations and the unpleasantness of a noxious stimulus. In two models of persistent pain, optogenetic activation of LH[PV] neurons or their ventrolateral periaqueductal gray area (vlPAG) axonal projections attenuates nociception, and neuroanatomical tracing reveals that LH[PV] neurons preferentially target glutamatergic over GABAergic neurons in the vlPAG. By contrast, LH[PV] projections to the lateral habenula regulate aversion but not nociception. Finally, we find that LH[PV] activation evokes additive to synergistic antinociceptive interactions with morphine and restores morphine antinociception following the development of morphine tolerance. Our findings identify LH[PV] neurons as a lateral hypothalamic cell type involved in nociception and demonstrate their potential as a target for analgesia.

*For correspondence:
yeka.aponte@nih.gov

Competing interests: The authors declare that no competing interests exist.

## Introduction

Responding appropriately to environmental stimuli is vital to an organism's survival. Nociception facilitates survival via the detection of dangerous environmental stimuli, which organisms use to escape and avoid these threats (*Bolles and Fanselow, 1980*; *Tovote et al., 2015*). However, mal-adaptive processes following injury or infection can cause the transition to chronic pain, a clinical condition with great economic burden that is not well-addressed by current therapeutics (*Price et al., 2018*; *Grace et al., 2014*). The widespread failure of preclinical pain therapies to translate to the clinic may be due to the historical focus on studying acute, pain-stimulated nocifensive behaviors in naive animals such as paw withdrawal to heat, which are not maladaptive and necessitate the examination of off-target effects like sedation in separate assays (*Negus et al., 2015*). Rather than physical sensitization to painful stimuli, the more problematic components of chronic pain in humans are likely the loss of ability to perform standard daily life activities and development of comorbid depression (*Asmundson and Katz, 2009*; *Cleeland and Ryan, 1994*; *Dworkin et al., 2005*; *Elman et al., 2013*; *Negus et al., 2006*). As such, rodent studies searching for new analgesics have begun to investigate ethological behaviors like nesting that are suppressed by noxious stimulation (e.g., forgoing standard life activities) as well as the affective/emotional component of nociception with assays of noxious stimulus-induced aversion (e.g., comorbid depression) (*Negus et al., 2015*; *Johansen et al., 2001*; *Corder et al., 2019*). Identifying specific brain pathways capable of

managing these multiple components of chronic pain behavior and developing strategies for targeting them for translational use will advance the search for novel analgesics.

Decades ago, the lateral hypothalamus (LH) was identified as a brain region responsive to noxious stimuli that is capable of controlling pain-related behavioral responses and modulating neuronal activity in the periaqueductal gray area (PAG) (*Cox and Valenstein, 1965*; *Lopez et al., 1991*; *Dafny et al., 1996*; *Fuchs and Melzack, 1995*; *Behbehani et al., 1988*). Pharmacological experiments have implicated various neurotransmitters and receptors in the regulation of nociception by the LH-PAG pathway, including $\alpha_1$- and $\alpha_2$-adrenoceptors, cannabinoid 1 receptors (CNR1), hypocretin 2 receptors (HCRT2), tachykinin 1 receptors (TACR1; neurokinin 1 [NK1] receptor), and substance P (*Esmaeili et al., 2017*; *Holden et al., 2009*; *Holden et al., 2002*; *Holden and Naleway, 2001*). However, characterizing the specific LH cell types associated with nociception or other behavioral processes has only recently been enabled by modern neurobiological approaches.

While the LH circuits controlling food intake and reward have received intense focus over the past several years (*Jennings et al., 2015*; *Jennings et al., 2013*; *Qualls-Creekmore et al., 2017*; *Navarro et al., 2016*; *Barbano et al., 2016*; *Nieh et al., 2016*), those governing nociception have been understudied by comparison. Thus, with its diverse array of neuronal populations (*Mickelsen et al., 2019*), uncovering genetically defined LH circuits that regulate pain behavior may bring forth novel therapeutic targets. We previously described a small population of fast-spiking glutamatergic LH neurons expressing parvalbumin (LH[PV] neurons) that forms functional excitatory synapses in the ventrolateral periaqueductal gray area (vlPAG) and regulates acute thermal and chemical nociception (*Siemian et al., 2019*; *Kisner et al., 2018*). However, the broader therapeutic potential of LH[PV] neurons and their specific targets within the vlPAG have not yet been fully assessed.

Using in vivo calcium imaging, we demonstrate that LH[PV] neurons exhibit time-locked responses to acute hot or cold stimuli as well as increased activity following the administration of a persistent inflammatory irritant. Additionally, we show that chemogenetic modulation of LH[PV] neurons alters not only reflexive nociceptive behaviors over a timescale of hours but also restores noxious stimulus-suppressed behavior and ameliorates noxious stimulus-associated negative affect. In models of persistent inflammatory or neuropathic pain, optogenetic activation of LH[PV] neurons or their axonal projections in the vlPAG attenuates nociception. Furthermore, neuroanatomical tracing using modified rabies virus revealed that LH[PV] neurons preferentially target nociception-suppressing glutamatergic neurons over nociception-facilitating GABAergic neurons in the vlPAG. Interestingly, we observed that activation of an LH[PV] neuron pathway to the lateral habenula (LHb) can mediate aversion-like behavior but not nociception, suggesting pathway-specific behavioral effects of these neurons. Finally, we report that LH[PV] neuronal activation evokes additive to synergistic antinociceptive interactions with morphine and restores morphine antinociception following the development of morphine tolerance. Our findings identify LH[PV] neurons as a lateral hypothalamic cell type intricately involved in nociception and demonstrate their potential as a novel target for analgesic treatment or for use in combination therapies with current analgesics.

## Results

### In vivo functional imaging of LH[PV] neurons

LH[PV] neurons bidirectionally modulate responses to acute noxious stimuli (*Siemian et al., 2019*), but their activity in response to noxious stimuli in vivo has not yet been studied. To investigate this, we used the combination of in vivo endomicroscopy with a genetically encoded calcium indicator (GCaMP) to measure intensity fluctuations of calcium-sensitive fluorophores as an indicator of neuronal activity in LH[PV] cells during behavior. First, we expressed a green fluorescent calcium indicator in LH[PV] neurons by injecting a Cre recombinase-dependent viral vector driving the expression of GCaMP6s (*Chen et al., 2013*) in the LH of *Pvalb[Cre]* transgenic mice (*Hippenmeyer et al., 2005*). For detection of GCaMP6s fluorescence, we implanted a GRIN lens above the LH[PV] nucleus and interfaced the lens with a detachable miniscope (*Figure 1a, b*). In conjunction with established and open-source computational algorithms for data processing (*Friedrich et al., 2017*; *Zhou et al., 2018*; *Pnevmatikakis and Giovannucci, 2017*), we were able to visualize (*Figure 1c*) and extrapolate calcium (Ca$^{2+}$) traces from individual LH[PV] neurons over periods of behavioral testing (*Figure 1d*).

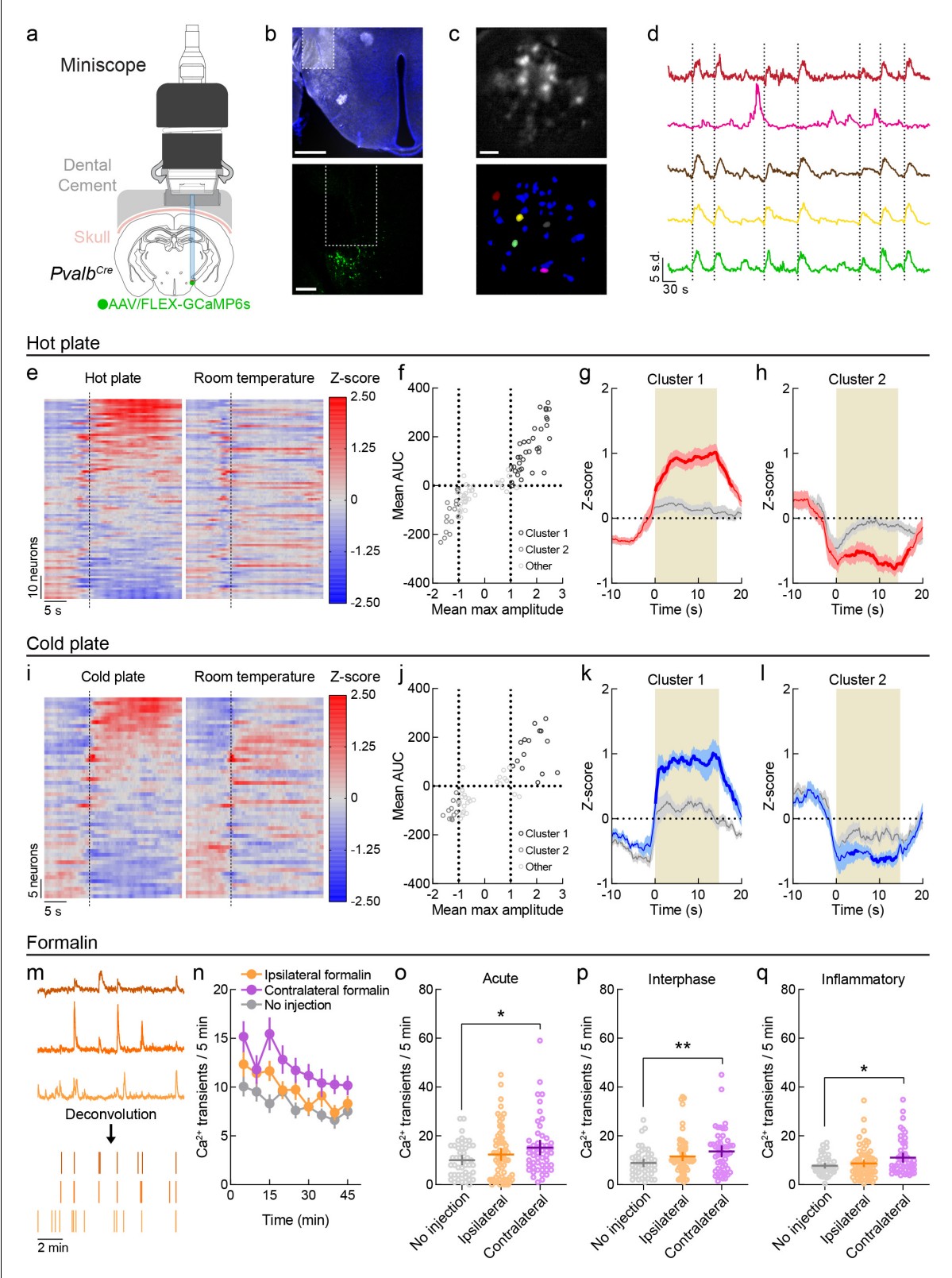

**Figure 1.** In vivo functional imaging of LH[PV] neurons. (**a**) Schematic configuration for deep-brain functional imaging from LH[PV] neurons in freely moving mice. Permission to publish miniscope drawing granted by Doric Lenses Inc. (**b**) Top: representative GRIN lens placement for functional imaging of LH[PV] neurons. Scale bar: 500 μm. Bottom: depiction of GRIN lens above GCaMP6s-expressing LH[PV] neurons. Scale bar: 200 μm. (**c**) Top: sample background-subtracted frame from a recording session. Bottom: spatial footprints of extracted neural segments. Scale bar: 100 μm. (**d**) Representative

*Figure 1 continued on next page*

*Figure 1 continued*

filtered traces from individual LH$^{PV}$ neurons. Dotted lines represent contacts with hot plate. (**e**) Z-scored Ca$^{2+}$ traces of LH$^{PV}$ neurons (87 neurons, three mice) averaged across exposures to a 51°C hot plate or a room temperature control surface. Dotted line represents contact with plate or control surface. (**f**) Clustering of 87 units by mean max amplitude and mean area under the curve (AUC) following hot plate surface contact. Dotted lines indicate the thresholds for inclusion into cluster 1 (mean max amplitude ≥ 1 and mean AUC ≥ 0) or cluster 2 (mean max amplitude ≤ 1 and mean AUC ≤ 0). (**g**) Neurons in cluster 1 (*n* = 35/87) displayed time-locked increases in activity in response to the hot plate as compared to the control room temperature surface. Two-way repeated-measures ANOVA on average Z-score per second revealed a significant time × stimulus interaction ($F_{(29, 986)}$ = 10.47, p<0.0001). Bonferroni multiple comparisons post-test significant between-stimulus differences are represented by the bolded red line. Red and gray shaded areas represent s.e.m. Tan shaded region represents average contact time with hot plate stimulus. (**h**) Neurons in cluster 2 (*n* = 16/87) displayed average decreases in activity in response to the hot plate as compared to the control room temperature surface. Two-way repeated-measures ANOVA on average Z-score per second revealed a significant time × stimulus interaction ($F_{(29, 435)}$ = 7.61, p<0.0001). Bonferroni multiple comparisons post-test significant between-stimulus differences are represented by the bolded red line. Red and gray shaded areas represent s.e.m. Tan shaded region represents average contact time with hot plate stimulus. (**i**) Z-scored Ca$^{2+}$ traces of LH$^{PV}$ neurons (53 neurons, three mice) averaged across exposures to a 4°C cold plate or a room temperature control surface. Dotted line represents contact with plate or control surface. (**j**) Clustering of 53 units by mean max amplitude and mean AUC following cold plate surface contact. Dotted lines indicate the thresholds for inclusion into cluster 1 (mean max amplitude ≥ 1 and mean AUC ≥ 0) or cluster 2 (mean max amplitude ≤ 1 and mean AUC ≤ 0). (**k**) Neurons in cluster 1 (*n* = 15/53) displayed time-locked increases in activity in response to the cold plate as compared to the control room temperature surface. Two-way repeated-measures ANOVA on average Z-score per second revealed a significant time × stimulus interaction ($F_{(29, 406)}$ = 5.94, p<0.0001). Bonferroni multiple comparisons post-test significant between-stimulus differences are represented by the bolded blue line. Blue and gray shaded areas represent s.e.m. Tan shaded region represents average contact time with cold plate stimulus. (**l**) Neurons in cluster 2 (*n* = 11/53, top) displayed average decreases in activity in response to the hot plate as compared to the control room temperature surface. Two-way repeated-measures ANOVA on average Z-score per second revealed a significant time × stimulus interaction ($F_{(29, 290)}$ = 2.05, p=0.0016). Bonferroni multiple comparisons post-test significant between-stimulus differences are represented by the bolded blue line. Blue and gray shaded areas represent s.e.m. Tan shaded region represents average contact time with cold plate stimulus. (**m**) Illustration of fluorescent trace deconvolution to estimated periods of neuronal firing. (**n**) Average deconvolved events per 5 min period following no injection (*n* = 46 neurons) or formalin injection in the hindpaw ipsilateral (*n* = 67 neurons) or contralateral (*n* = 51 neurons) to the brain hemisphere implanted with a GRIN lens. (**o–q**) Formalin induced fluctuations in LH$^{PV}$ neuronal activity in each phase of the formalin test. Mann–Whitney *U*-tests with Holm–Sidak correction for multiple comparisons revealed significantly higher Ca$^{2+}$ event frequency following contralateral formalin injection in the (**o**) acute (p=0.048), (**p**) interphase (p=0.0078), and (**q**) inflammatory phases (p=0.048) relative to no injection, whereas no significant differences were found between ipsilateral formalin and no injection (acute p=0.80, interphase p=0.18, inflammatory p=0.86). Lines and error bars indicate mean ±95% CI. See also *Figure 1—figure supplements 1* and *2*.

The online version of this article includes the following source data and figure supplement(s) for figure 1:

**Source data 1.** LH$^{PV}$ neuronal responses to acute thermal stimuli.

**Source data 2.** LH$^{PV}$ calcium transient frequency during formalin tests.

**Figure supplement 1.** Classification accuracy of noxious and neutral stimuli from LH$^{PV}$ neuronal activity.

**Figure supplement 1—source data 1.** Responses of cluster 1 and cluster 2 LH$^{PV}$ neurons to acute thermal stimuli used for trial-type decoding.

**Figure supplement 2.** Responses of LH$^{PV}$ neurons to acute thermal stimuli in individual mice and cell registration across sessions.

We first monitored Ca$^{2+}$ dynamics in LH$^{PV}$ neurons (*n* = 87 neurons, three mice) in response to an acute thermal hot plate stimulus and clustered the neurons according to their response properties (*Figure 1d−f*). In a subset of the recorded LH$^{PV}$ neurons ('Cluster 1,' *n* = 35/87 neurons), we observed time-locked increases in fluorescence in response to the 51°C hot plate relative to a room-temperature innocuous stimulus of similar visual and tactile properties, suggesting that this subpopulation of LH$^{PV}$ neurons becomes active in response to a thermal stimulus (*Figure 1g*). Another subset of neurons ('Cluster 2,' *n* = 16/87 neurons) exhibited an average decrease in activity in response to the hot plate relative to control stimulus (*Figure 1h*). We observed a similar profile of time-locked responses to a 4°C cold stimulus relative to a control innocuous stimulus (*Figure 1i, j*). One subset of the recorded neurons ('Cluster 1,' *n* = 15/53 neurons) was significantly activated in response to the cold plate relative to control stimulus (*Figure 1k*), while another subset ('Cluster 2,' *n* = 11/53 neurons) displayed significantly lower activity following the cold plate stimulus as compared to the control stimulus (*Figure 1l*). Within each cluster, we trained a support vector machine (SVM) classifier using averaged 10 s traces following contact with the noxious (hot/cold) or neutral surfaces and tested whether it could predict the stimulus type when given unlabeled traces. Remarkably, neuronal activity from each cluster except cluster 2 from the cold plate test could decode the correct stimulus type above chance levels (*Figure 1—figure supplement 1*). Cluster 1 and 2 neurons were observed in each of the mice tested. Thus, we registered cells across the hot plate and cold plate sessions to examine whether LH$^{PV}$ neurons exhibited consistent response profiles across tests (*Sheintuch et al., 2017*). Of the 33 total neurons that were detected in both sessions, only 6 remained in the same

cluster, and the area under the curves of the fluorescent traces of all 33 neurons did not significantly correlate between sessions, suggesting that the responses of LH[PV] neurons were generally variable across testing (*Figure 1—figure supplement 2*). Together, these results demonstrate that LH[PV] neuronal activity is modulated in response to acute thermal stimuli.

We next tracked LH[PV] neuronal activity over a longer timescale in response to a hindpaw injection of the chemical irritant formalin. Formalin induces discrete acute (0–5 min) and inflammatory (15–45 min) phases of pain behavior, separated by a brief interphase period (*Alhadeff et al., 2018*; *Dubuisson and Dennis, 1977*), allowing us to monitor changes in neuronal activity during each phase. Relative to recording sessions without formalin injection, we observed that the frequency of deconvolved $Ca^{2+}$ transients (*Figure 1m*) appeared to be generally higher following formalin injections in the hindpaw contralateral to the brain hemisphere in which the GRIN lens was implanted (*Figure 1n*), and statistical analyses of $Ca^{2+}$ event frequency within each period supported this observation (*Figure 1o−q*). Cell registration revealed that only three neurons were detected in more than three of these imaging sessions, thus we could not examine whether a neuron being in cluster 1 or 2 in the hot plate and cold plate tests impacted its response properties in the formalin tests (*Figure 1—figure supplement 2*). Together, these findings indicate that LH[PV] neurons display changes in spontaneous activity in response to several stimulus modalities, including both acute thermal stimuli and ongoing chemical inflammation.

## LH[PV] neurons regulate sensory and affective aspects of pain over long timescales

We next examined whether manipulating LH[PV] neuronal activity can alter noxious stimulation-suppressed behavior and negative affect, which may be better indicators of clinical utility than stimulus-evoked behaviors (e.g., reflexive withdrawal to acute thermal stimuli). To investigate this, we targeted these neurons for chemogenetic manipulations by bilaterally injecting Cre recombinase-dependent viral vectors driving the expression of either the excitatory designer receptor hM3D, the inhibitory designer receptor hM4D, or the fluorophore mCherry as control into the LH of *Pvalb^{Cre}* transgenic mice (*Figure 2a*). Activation of the designer receptors via administration of the ligand clozapine-*N*-oxide (CNO, 1 mg/kg, i.p.) evoked significant increases and decreases in $PWL_{HP}$ in LH[PV]: hM3D and LH[PV]:hM4D mice, respectively, as compared to mCherry controls, with effects detectable between 1 and 18 hr post-injection (*Figure 2b*). Thus, chemogenetic manipulations of these neurons alter nociception over a long timescale.

Pain in basic research is traditionally assessed by measuring 'pain-stimulated behavior' or the elicited reactions to noxious stimuli (e.g., paw withdrawal). However, clinical pain disorders often impact quality of life more profoundly by deterring actions normally performed when healthy (*Cleeland and Ryan, 1994*; *Dworkin et al., 2005*). Therefore, we next examined the effects of LH[PV] neuronal activity in a model of 'pain-suppressed behavior,' which measures a decrease in behavioral output following a noxious stimulus (*Negus et al., 2006*). Healthy mice normally collect nestlet pieces distributed throughout the home cage within 30 min and begin nest building, a natural behavior (*Negus et al., 2015*; *Diester et al., 2021a*; *Diester et al., 2021b*; *Figure 2c*); this was not affected by CNO administration across groups (*Figure 2d*, 'CNO + saline'). However, administration of acetic acid (0.6%; i.p.) significantly decreases nesting behavior; this was apparent in all three groups without LH[PV] manipulations (*Figure 2d*, 'Vehicle + acid'). Interestingly, activation of LH[PV] neurons in LH[PV]:hM3D mice prior to acetic acid injection prevented the reductions in nesting behavior (*Figure 2d*, 'CNO + acid') as compared to tests without CNO and to LH[PV]:Ctrl mice. In contrast, no changes were observed in LH[PV]:hM4D mice. Thus, LH[PV] activation not only decreases noxious stimulus-evoked behavior but also restores behaviors normally suppressed by noxious stimulation.

Pain results not only in overt behavioral changes but also negative affect, as made evident by the high comorbidity between pain and mood disorders (*Asmundson and Katz, 2009*; *Elman et al., 2013*). We sought to determine the role of LH[PV] neuronal activity on the affective, or emotional, component of a painful experience. For this, we used a place conditioning paradigm in which mice avoid a context paired with an aversive event (*Johansen et al., 2001*; *Alhadeff et al., 2018*; *Figure 2e*). After assessment of initial side preference of a two-chamber apparatus, we passively conditioned the mice by administering CNO (i.p.) with intra-plantar formalin to induce inflammation in the initially preferred side and CNO with intra-plantar saline in the initially less-preferred side. Mice were conditioned twice in each context on alternating days and then were given

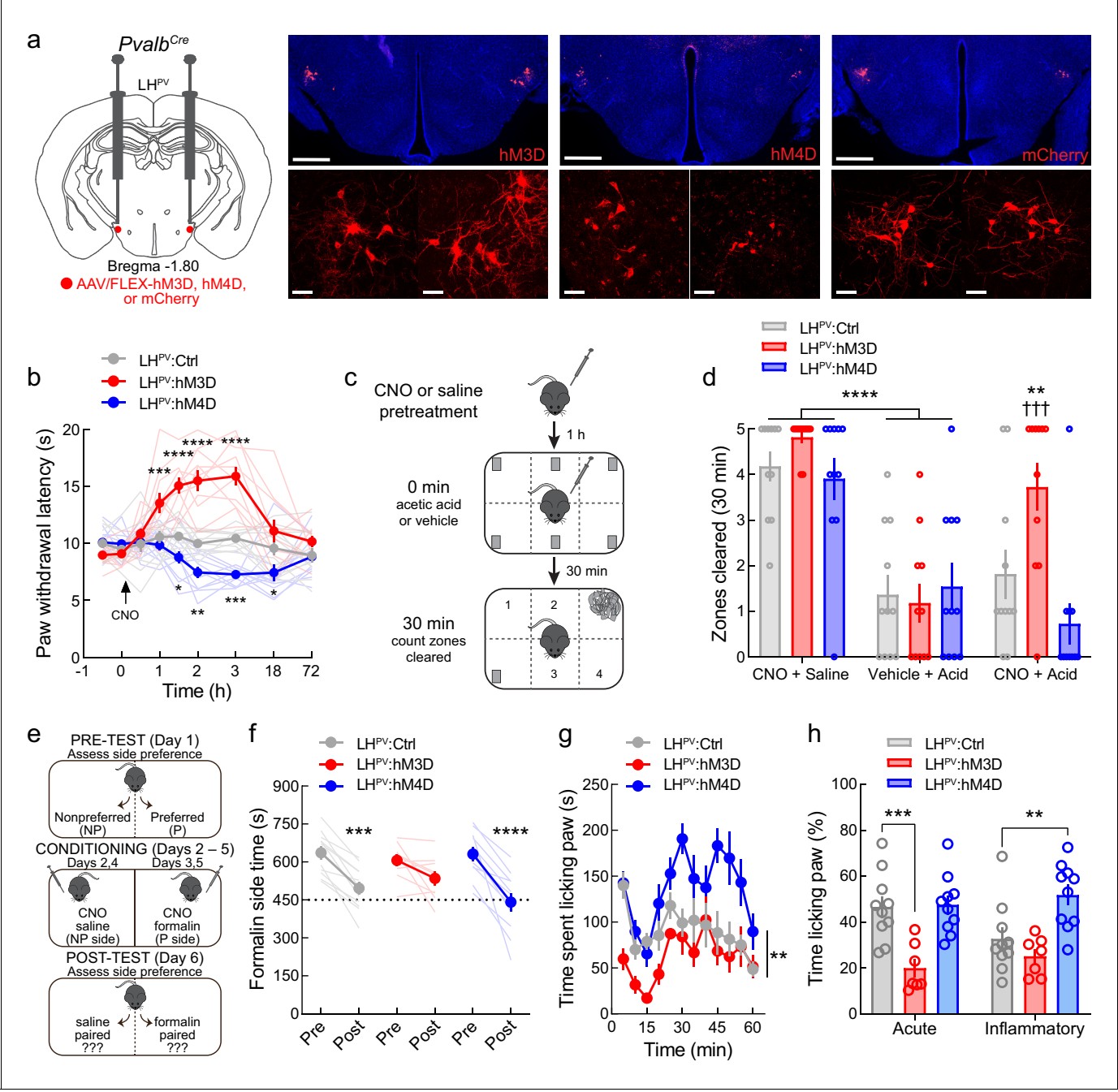

**Figure 2.** Chemogenetic modulation of LH[PV] neurons regulates pain-suppressed behavior and alters pain-associated negative affect. (**a**) Representative images of hM3D, hM4D, or mCherry expression in LH[PV] neurons. Scale bars: 500 μm, widefield; 50 μm, zoom. (**b**) Chemogenetic activation and inhibition of LH[PV] neurons evoked long-lasting significant increases and decreases in thermal pain thresholds, respectively (n = 11 mice per group; two-way mixed-model ANOVA group × time interaction, $F_{(16, 240)}$=14.15, p<0.0001). Significant differences from LH[PV]:mCherry mice were determined by Bonferroni multiple comparisons tests and are represented graphically, *p<0.05, **p<0.01, ***p<0.001, ****p<0.0001. (**c**) Schematic for pain-suppressed nesting assay. (**d**) Chemogenetic activation of LH[PV] neurons prevented the reductions in nesting behavior induced by i.p. injection of 0.6% acetic acid (10 ml/kg). Two-way mixed-model ANOVA revealed a significant group × test interaction (n = 11 mice per group; $F_{(4, 60)}$ = 4.17, p=0.0048). Bonferroni multiple comparisons post-tests revealed no differences in normal nesting behavior from clozapine-*N*-oxide (CNO) injections (p=0.92), and that acetic acid injection decreased nesting behavior across groups when administered without CNO (p<0.0001). Administration of CNO before acetic acid increased nesting behavior in LH[PV]:hM3D mice relative to LH[PV]:mCherry control mice (p=0.008; Cohen's *d* = 1.09) and tests without CNO (p=0.0002). (**e**) Schematic of the formalin place conditioning experiment. (**f**) Chemogenetic modulation of LH[PV] neurons altered the effects of formalin on place conditioning. Two-way mixed-model ANOVA revealed a significant group × test interaction (n = 11 mice per group; $F_{(2,28)}$ = 3.89, p=0.032). Bonferroni multiple comparisons post-tests showed significant shift in chamber preference in LH[PV]:mCherry (p=0.0001) and LH[PV]:hM4D mice (p<0.0001) but not

*Figure 2 continued on next page*

Figure 2 continued
LH[PV]:hM3D mice (p=0.094). (g) Time spent paw licking was altered by LH[PV] neuronal modulation (n = 10 LH[PV]:mCherry, 7 LH[PV]:hM3D, and 10 LH[PV]:
hM4D mice; two-way mixed-model ANOVA group × time interaction, $F_{(22, 264)}$ = 1.99, p=0.0064). (h) Acute and inflammatory phase paw licking were
differentially altered by LH[PV] neuronal activation and inhibition (n listed above; two-way mixed-model ANOVA group × phase interaction, $F_{(2,}$
$_{24)}$ = 4.33, p=0.025). Activation of LH[PV] neurons in LH[PV]:hM3D mice decreased acute (p=0.0004, Cohen's d = 2.07) but not inflammatory phase paw
licking (p=0.50), whereas LH[PV] neuronal inhibition in LH[PV]:hM4D mice increased inflammatory (p=0.0049, Cohen's d = 1.29) but not acute phase paw
licking (p>0.99).

free access to both chambers during a post-test to assess changes in place preference. As expected, LH[PV]:Ctrl mice lost preference to the formalin-paired context as compared to pre-formalin preference levels (*Figure 2f*). However, activation of LH[PV] neurons during conditioning attenuated this loss of place preference, whereas inhibition of LH[PV] neurons during conditioning permitted the loss of place preference (*Figure 2f*). Furthermore, the time spent paw licking in these sessions was bidirectionally affected by chemogenetic LH[PV] neuronal activation or inhibition (*Figure 2g, h*). LH[PV] neuronal activation decreased paw licking during the acute but not inflammatory phase, whereas inhibition increased paw licking in the inflammatory but not acute phase. Together, these results support a role for LH[PV] neurons both in pain behaviors and associated negative affect.

## Optogenetic activation of LH[PV] neurons attenuates persistent inflammatory pain-associated behaviors

Since LH[PV] neurons ameliorated moderately long-lasting behavioral effects of pain, we next sought to determine whether they could also alter nociceptive thresholds in traditional models of persistent pain behavior. We targeted LH[PV] neurons for optogenetic manipulations with bilateral injections of a Cre recombinase-dependent viral vector driving the expression of either channelrhodopsin (ChR2: tdTomato; light-sensitive neuronal activator) or GFP (control fluorophore) in the LH of *Pvalb*[Cre] transgenic mice and implanted optical fibers bilaterally above these neurons (*Figure 3a*). Activation of LH[PV] neurons in naive mice significantly increased paw withdrawal latency in response to a 51°C hot plate (PWL[HP], *Figure 3b*). However, activating these neurons did not change paw withdrawal threshold in the von Frey filament test (PWT[VF], *Figure 3c*), suggesting that these neurons regulate acute thermal but not mechanical nociception in healthy mice. Since LH[PV] neurons are glutamatergic (*Siemian et al., 2019*; *Kisner et al., 2018*) and activation of LH neurons expressing the vesicular glutamate transporter 2 (SLC17A6; LH[VGLUT2]) is aversive (*Jennings et al., 2013*), we also assessed the effects of LH[PV] neuronal activation in a real-time place preference (RTPP) assay in which photostimulation was paired with one-half of the behavioral arena. Activation of LH[PV] neurons was mildly aversive as mice spent significantly less time on the photostimulation-paired side (*Figure 3d*), suggesting that these neurons may play a role in reward- and aversion-like behaviors. Next, we injected complete Freund's adjuvant (CFA), a well-known inflammatory reagent (*Alhadeff et al., 2018*; *Fehrenbacher et al., 2012*; *Nagakura et al., 2003*), into the right hindpaw to cause inflammation and induce persistent hypersensitivity. We observed a significant decrease in nociceptive thresholds for both thermal and mechanical stimuli following these CFA injections (*Figure 3—figure supplement 1a, b*). Interestingly, activation of LH[PV] neurons after CFA evoked significant increases in both PWL[HP] and PWT[VF] (*Figure 3e, f*). Additionally, activation of LH[PV] neurons no longer triggered place avoidance (*Figure 3g*). Furthermore, we observed that the magnitude of the PWL[HP] response depends on the photostimulus frequency (*Figure 3—figure supplement 1c*) and that LH[PV] neuron-mediated antinociception was not strictly photostimulus-bound (*Figure 3—figure supplement 1d*) as the antinociceptive effects persisted for several minutes after photostimulation ceased. Together, these results indicate that LH[PV] neuronal activation attenuates hypersensitivity to both thermal and mechanical stimuli following the onset of inflammation.

## LH[PV] neurons target excitatory circuits within the vlPAG to regulate pain behaviors

LH[PV] neurons send dense projections to the vlPAG (*Kisner et al., 2018*; *Celio et al., 2013*), where they form functional excitatory synapses. We next examined whether this LH[PV]→vlPAG pathway also regulates nociception in models of persistent pain behavior. To specifically target and manipulate the LH[PV]→vlPAG pathway, we bilaterally injected a Cre recombinase-dependent viral vector driving

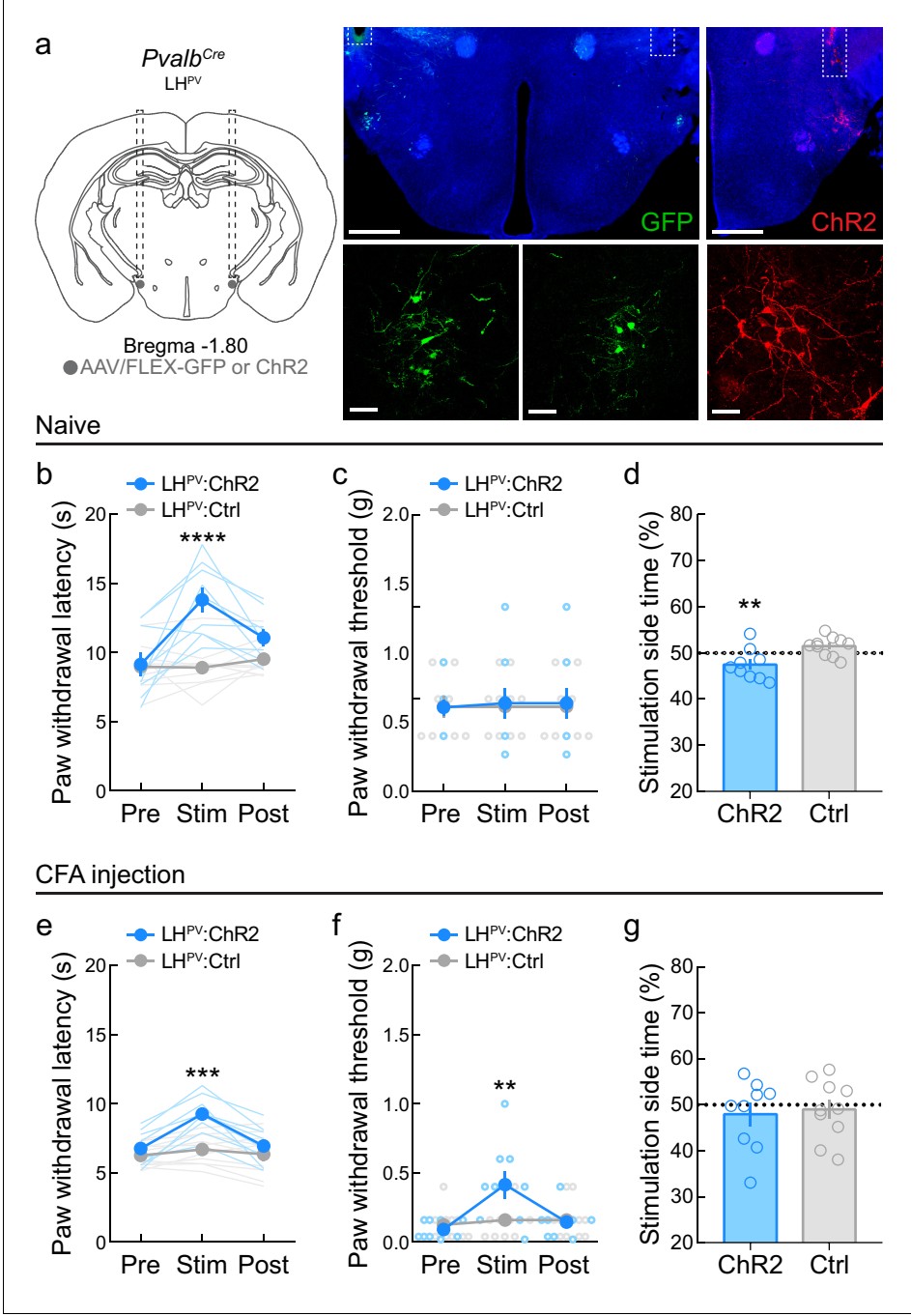

**Figure 3.** Optogenetic activation of LH[PV] neurons attenuates thermal and mechanical nociception following the induction of inflammatory pain. (**a**) Representative images of ChR2 or GFP expression in LH[PV] neurons and optical fiber implants above the lateral hypothalamus (LH). Scale bars: 500 µm, widefield; 50 µm, zoom. (**b**) Optogenetic activation of LH[PV] neurons in naive mice triggers thermal antinociception ($n$ = 9 ChR2 mice and 10 Ctrl mice). Two-way mixed-model ANOVA revealed a significant group × epoch interaction ($F(2, 34)$ = 14.01, p<0.0001), and Bonferroni multiple comparisons post-test showed that LH[PV]:ChR2 mice had significantly higher PWL$_{HP}$ during the photostimulation epoch than LH[PV]:Ctrl mice, p<0.0001; Cohen's $d$ = 2.22. (**c**) Optogenetic activation of LH[PV] neurons in naive mice does not affect mechanical nociception ($n$ = 9 ChR2 mice and 10 Ctrl mice). Two-way mixed-model ANOVA interaction, p=0.87. (**d**) Naive LH[PV]:ChR2 mice displayed significant real-time place avoidance to photostimulation relative to controls ($n$ = 9 ChR2 mice and 10 Ctrl mice, $t(17)$ = 3.15, p=0.0058, Cohen's $d$ = 1.43). (**e**) Optogenetic activation of LH[PV] neurons in mice 5 days following complete Freund's adjuvant (CFA) injection triggers increases in PWL$_{HP}$ ($n$ = 9 ChR2 mice and 10 Ctrl mice). Two-way mixed-model ANOVA revealed a

*Figure 3 continued on next page*

*Figure 3 continued*

significant group × epoch interaction ($F$(2, 34) = 15.05, p<0.0001), and Bonferroni multiple comparisons post-test showed that LH$^{PV}$:ChR2 mice had significantly higher PWL$_{HP}$ during the photostimulation epoch than LH$^{PV}$:Ctrl mice (p=0.0001; Cohen's $d$ = 2.08). (f) Optogenetic activation of LH$^{PV}$ neurons in mice 6 days following CFA injection triggers increases in PWT$_{VF}$ ($n$ = 9 ChR2 mice and 10 Ctrl mice). Two-way mixed-model ANOVA revealed a significant group × epoch interaction ($F$(2, 34) = 11.28, p=0.0002), and Bonferroni multiple comparisons post-test showed that LH$^{PV}$:ChR2 mice had significantly higher PWL$_{HP}$ during the photostimulation epoch than LH$^{PV}$:Ctrl mice (p=0.003; Cohen's $d$ = 1.11). (g) LH$^{PV}$:ChR2 mice did not display significant real-time place avoidance to photostimulation relative to controls 7 days post-CFA, p=0.75 ($n$ = 9 ChR2 mice and 10 Ctrl mice). See also *Figure 3—figure supplement 1*.

The online version of this article includes the following figure supplement(s) for figure 3:

**Figure supplement 1.** Effects of optogenetic activation of LH$^{PV}$ neurons following complete Freund's adjuvant (CFA) inflammatory pain induction.

the expression of channelrhodopsin (ChR2:tdTomato), the light-sensitive neuronal silencer archaerhodopsin (ArchT:GFP), or the fluorophore GFP (control) into the LH of *Pvalb*$^{Cre}$ mice and implanted optical fibers bilaterally above the vlPAG to specifically manipulate the axonal projections of LH$^{PV}$ neurons (*Figure 4a*). In naive mice, activation of the LH$^{PV}$→vlPAG pathway evoked increases in PWL$_{HP}$ but not PWT$_{VF}$, similar to somatic manipulations (*Figure 4b, c*), whereas inhibition of the pathway decreased both nociceptive thresholds (*Figure 4d, e*). However, in contrast to somatic manipulations, no effects were observed for either LH$^{PV}$→vlPAG activation or inhibition in the RTPP test (*Figure 4f*), suggesting that there were no changes in the overall affective state of the mice that may have contributed to these bidirectional effects on nociception. In healthy mice, we also observed that the magnitude of the PWL$_{HP}$ response during activation of the LH$^{PV}$→vlPAG pathway depends on photostimulus frequency. Moreover, these responses were not affected by systemic administration of the cannabinoid receptor 1 (CNR1 or CB1) antagonist/inverse agonist rimonabant (3 mg/kg, i.p.; *Figure 4g*) despite the PAG being an important site for cannabinoid-mediated antinociception (*Esmaeili et al., 2017*; *Finn et al., 2003*; *Maione et al., 2006*). These results suggest that blocking CB1 receptors does not affect antinociception driven by LH$^{PV}$→vlPAG circuitry.

We next investigated the effects of activating the LH$^{PV}$→vlPAG pathway in the spared nerve injury (SNI) model of neuropathy. Five days post-SNI, we observed significant decreases in thermal and mechanical thresholds (*Figure 4—figure supplement 1a, b*). Photostimulation of the LH$^{PV}$→vlPAG pathway evoked increases in both PWL$_{HP}$ and PWT$_{VF}$ (*Figure 4h, i*), and this remained during testing 25 days post-SNI (*Figure 4—figure supplement 1c, d*). Furthermore, no effects of LH$^{PV}$→vlPAG pathway activation were observed in the RTPP test post-SNI (*Figure 4j*). Due to the modest effects of LH$^{PV}$→vlPAG activation on mechanical thresholds, we predicted that this pathway may be more effective during inflammatory than neuropathic conditions. Therefore, in a new cohort of mice, we activated the LH$^{PV}$→vlPAG pathway before (*Figure 4k, l*) and after (*Figure 4m, n*) the induction of inflammation by CFA (*Figure 4—figure supplement 2a, b*) and observed robust effects on both thermal and mechanical nociceptive thresholds. Similar to the SNI cohort, LH$^{PV}$→vlPAG activation post-CFA did not affect RTPP, suggesting that there were no effects on reward- or aversion-related behaviors (*Figure 4o*). Furthermore, LH$^{PV}$→vlPAG pathway-mediated antinociception post-CFA was dependent on photostimulus frequency but not strictly photostimulus-bound (*Figure 4—figure supplement 2c, d*). Together, these results show that the LH$^{PV}$→vlPAG pathway regulates nociception in at least two models of persistent pain behavior and that its activation is more effective in attenuating inflammatory than neuropathic hypersensitivity.

Within the vlPAG, GABAergic and glutamatergic neurons play opposing roles in regulating nociception and defensive behavior (*Samineni et al., 2017*; *Tovote et al., 2016*). Although we previously showed that LH$^{PV}$ neurons form functional excitatory synapses with vlPAG neurons (*Siemian et al., 2019*), the identity of these post-synaptic targets remains unknown. Thus, we used a monosynaptic retrograde viral tracing strategy with a modified rabies virus (*Wickersham et al., 2007a*; *Wickersham et al., 2007b*) to identify the targets of LH$^{PV}$ neurons in the vlPAG. In *Slc17a6*-$^{Cre}$ and *Slc32a1*$^{Cre}$ mice (*Vong et al., 2011*), we injected starter cells in the vlPAG with Cre recombinase-dependent helper virus containing rabies glycoprotein G and the EnvA receptor for avian sarcoma leukosis virus (TVA) to express the proteins required for uptake and monosynaptic

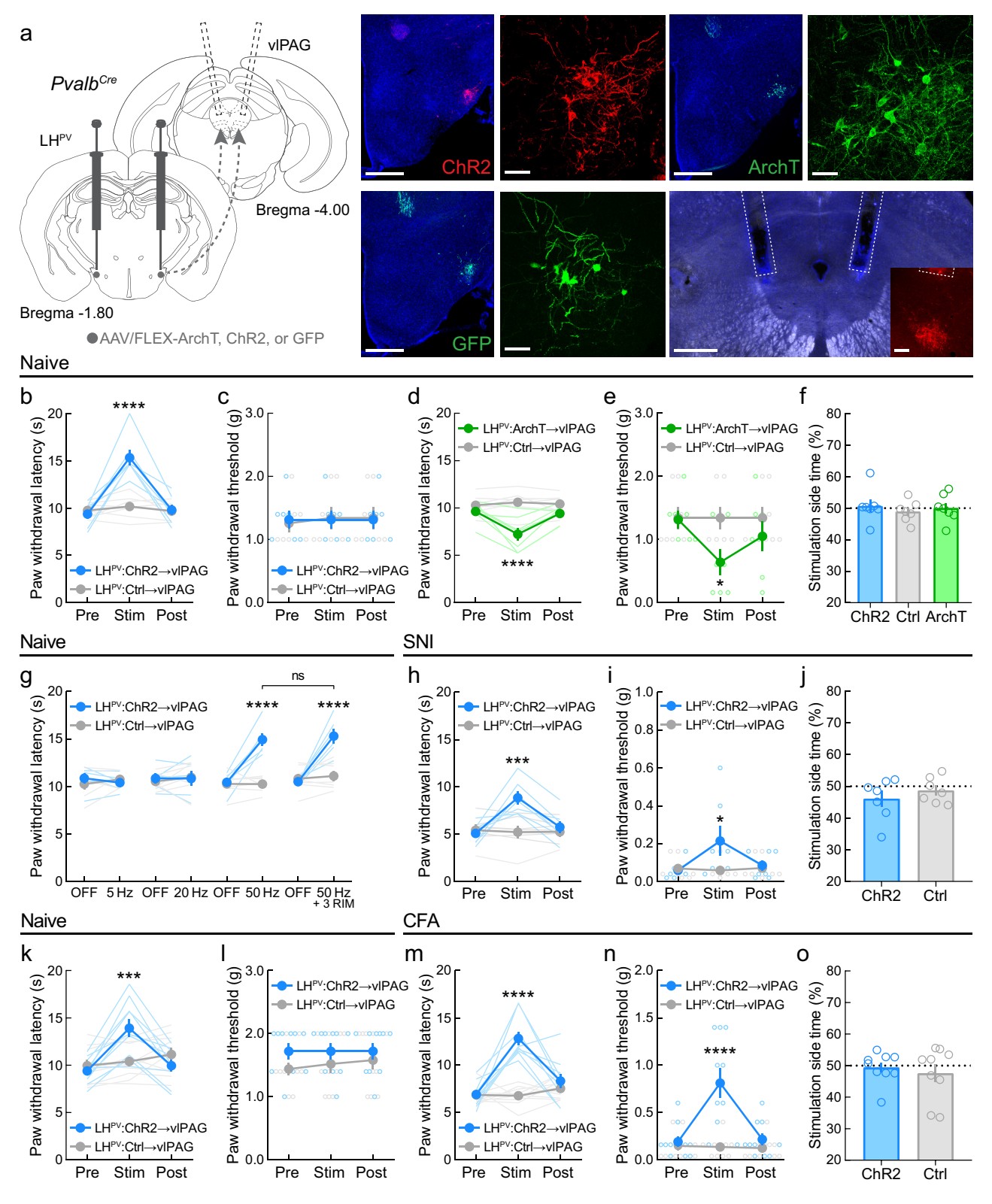

**Figure 4.** LH[PV]→vlPAG pathway mediates nociception in models of chronic neuropathic and inflammatory pain. (a) Representative images of ChR2, ArchT, or GFP expression in LH[PV] neurons and optical fiber implants above the ventrolateral periaqueductal gray area (vlPAG). Inset shows axons from LH[PV] neurons under the optical fiber. Scale bars: 500 μm, widefield; 50 μm, zoom; 100 μm, inset. (b) In naive mice, optogenetic activation of LH[PV] axonal projections in the vlPAG evokes thermal antinociception (n = 7 mice per group; two-way mixed-model ANOVA group × epoch interaction, *F*(2,

*Figure 4 continued on next page*

*Figure 4 continued*

24) = 25.19, p<0.0001, Bonferroni multiple comparisons post-test during photostimulation epoch, p<0.0001; Cohen's *d* = 2.85) but not (**c**) mechanical antinociception (p=0.38). (**d**) In naive mice, optogenetic inhibition of LH$^{PV}$ axonal projections in the vlPAG decreases both thermal (*n* = 7 mice per group; two-way mixed-model ANOVA group × epoch interaction, $F_{(2, 24)}$ = 11.96, p=0.0002, Bonferroni multiple comparisons post-test during photostimulation epoch, p<0.0001; Cohen's *d* = 2.13) and (**e**) mechanical thresholds (two-way mixed-model ANOVA group × epoch interaction, $F_{(2, 24)}$ = 12.10, p<0.0002, Bonferroni multiple comparisons post-test during photostimulation epoch, p=0.038; Cohen's *d* = 1.36). (**f**) Optogenetic activation or inhibition of the LH$^{PV}$→vlPAG pathway did not affect real-time place preference behavior in naive mice (*n* = 7 mice per group; one-way ANOVA, $F_{(2, 18)}$ = 0.28, p=0.76). (**g**) LH$^{PV}$→vlPAG activation-induced antinociception is dependent on photostimulus frequency but is not attenuated by the CB1 receptor antagonist rimonabant (3 mg/kg, i.p.; '3 RIM'). Two-way mixed-model ANOVA revealed a significant group × epoch interaction (*n* = 6 ChR2 mice and 7 Ctrl mice; $F_{(7, 77)}$ = 14.27, p<0.0001). Bonferroni multiple comparisons post-tests revealed between-group differences during the '50 Hz' and '50 Hz + 3 RIM' epochs (p<0.0001), but no within-group differences between these epochs (p>0.99). (**h**) Optogenetic activation of the LH$^{PV}$→vlPAG pathway evokes increases in PWL$_{HP}$ on day 5 post-spared nerve injury (SNI) (*n* = 7 mice per group; two-way mixed-model ANOVA group × epoch interaction, $F_{(2, 24)}$ = 12.86, p<0.0001, Bonferroni multiple comparisons post-test during photostimulation epoch, p=0.0002; Cohen's *d* = 2.04) and (**i**) PWT$_{VF}$ on day 6 post-SNI (two-way mixed-model ANOVA group × epoch interaction, $F_{(2, 24)}$ = 5.24, p<0.013, Bonferroni multiple comparisons post-test during photostimulation epoch, p=0.019; Cohen's *d* = 1.03). (**j**) On day 7 post-SNI, optogenetic activation of the LH$^{PV}$→vlPAG pathway did not affect real-time place preference behavior (*n* = 7 mice per group; p=0.39). (**k**) In a new cohort of naive mice, optogenetic activation of the LH$^{PV}$→vlPAG pathway evoked thermal (*n* = 10 mice per group; two-way mixed-model ANOVA group × epoch interaction, $F_{(2, 36)}$ = 23.64, p<0.0001, Bonferroni multiple comparisons post-test during photostimulation epoch, p=0.0009) but not (**l**) mechanical antinociception (p=0.31). (**m**) Optogenetic activation of the LH$^{PV}$→vlPAG pathway evokes increases in PWL$_{HP}$ on day 5 post-complete Freund's adjuvant (CFA) injection (*n* = 10 mice per group; two-way mixed-model ANOVA group × epoch interaction, $F_{(2, 36)}$ = 19.65, p<0.0001, Bonferroni multiple comparisons post-test during photostimulation epoch, p<0.0001; Cohen's *d* = 3.66) and (**n**) PWT$_{VF}$ on day 6 post-SNI (two-way mixed-model ANOVA group × epoch interaction, $F_{(2, 36)}$ = 24.63, p<0.0001, Bonferroni multiple comparisons post-test during photostimulation epoch, p<0.0001; Cohen's *d* = 1.88). (**o**) On day 7 post-CFA injection, optogenetic activation of the LH$^{PV}$→vlPAG pathway did not affect real-time place preference behavior (*n* = 9 mice per group; p=0.59). See also ***Figure 4—figure supplements 1–3***.

The online version of this article includes the following figure supplement(s) for figure 4:

**Figure supplement 1.** Effects of optogenetic activation of LH$^{PV}$ axonal projections in the ventrolateral periaqueductal gray area (vlPAG) following spared nerve injury (SNI) neuropathic pain induction.

**Figure supplement 2.** Effects of optogenetic activation of LH$^{PV}$ axonal projections in the ventrolateral periaqueductal gray area (vlPAG) following complete Freund's adjuvant (CFA) inflammatory pain induction.

**Figure supplement 3.** Behavioral outputs evoked by optogenetic activation of GABAergic LH$^{LEPR}$ axonal projections in the ventrolateral periaqueductal gray area (vlPAG).

propagation of modified rabies virus (***Figure 5a***). Three weeks later, we injected the EnvA-pseudo-typed G-deleted rabies virus RVdG-mCherry(EnvA) into the vlPAG. After an additional 3 weeks, mice were perfused, and brains were processed for histological assessment. LH-containing sections were immunostained with an anti-parvalbumin antibody and imaged using confocal microscopy (***Figure 5b***). Quantitative analyses revealed that more vlPAG$^{VGLUT2}$ neurons (14.89%; *n* = 63 of 423 neurons, three mice) than vlPAG$^{VGAT}$ neurons (6.96%; *n* = 22 of 316 neurons, six mice) are synaptically targeted by LH$^{PV}$ neurons (Chi-square = 11.18, ***p=0.0008, ***Figure 5c***). Since activation of glutamatergic neurons in the PAG was shown previously to decrease pain (***Samineni et al., 2017***), our findings suggest a potential role for LH$^{PV}$ neurons as an excitatory input to glutamatergic vlPAG circuitry.

To determine whether other lateral hypothalamic circuits encode for nociception, we examined the effects of manipulating LH leptin receptor expressing (LH$^{LEPR}$) neurons, which also project to the vlPAG, albeit a slightly more posterior region (***Schiffino et al., 2019***; ***Leinninger et al., 2009***). In contrast to LH$^{PV}$ neurons, LH$^{LEPR}$ neurons are predominantly GABAergic and their vlPAG axonal projections are more broadly distributed than those of LH$^{PV}$ neurons. Cre-dependent viruses driving ChR2 or GFP expression were injected into the LH of *Lepr$^{Cre}$* mice (***Leshan et al., 2006***; ***Figure 4—figure supplement 3a***), and we found that activation of the LH$^{LEPR}$→vlPAG pathway potentiated both thermal and mechanical nociception in healthy mice (***Figure 4—figure supplement 3b–e***). Moreover, activation of the LH$^{LEPR}$→vlPAG pathway was rewarding as LH$^{LEPR}$:ChR2→vlPAG mice spent more time on the photostimulation-paired side of the chamber than LH$^{LEPR}$:Ctrl→vlPAG control mice (***Figure 4—figure supplement 3f***). These results demonstrate that activation of lateral hypothalamic glutamatergic (LH$^{PV}$) and GABAergic (LH$^{LEPR}$) populations that project to the vlPAG attenuates and potentiates nociception, respectively.

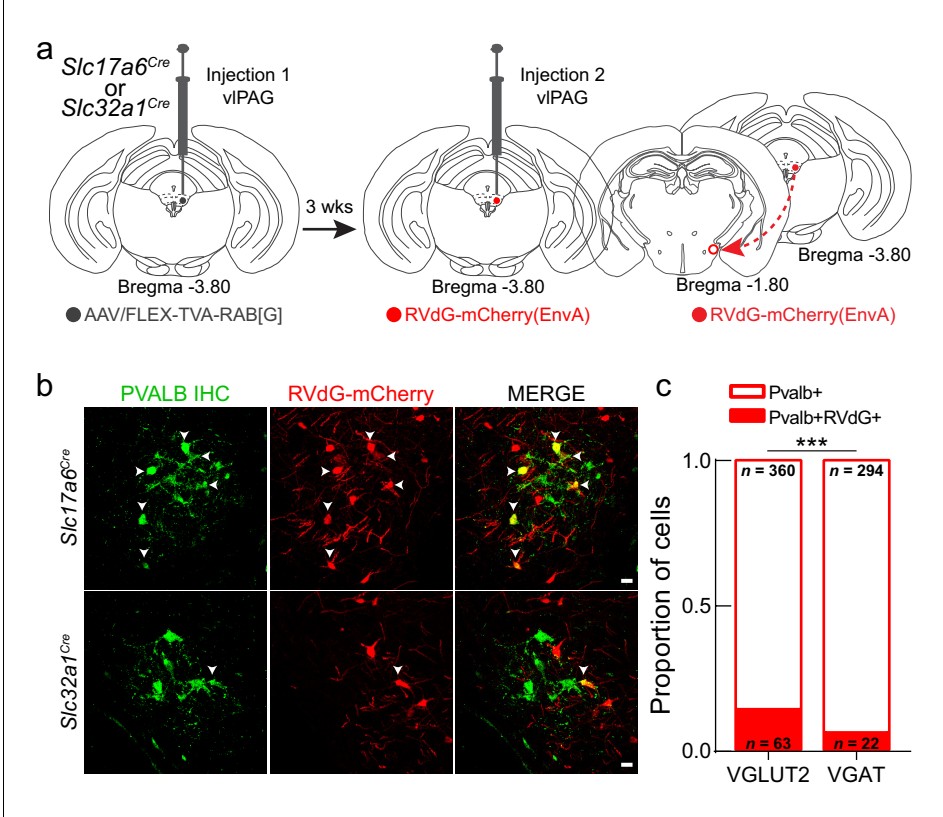

**Figure 5.** LH[PV] neurons preferentially target glutamatergic neurons in the ventrolateral periaqueductal gray area (vlPAG). (a) Schematic for modified rabies viral tracing strategy. (b) Images from *Slc17a6*[Cre] (top row) and *Slc32a1*[Cre] (bottom row) brain slices showing the overlap of RVdG-mCherry(EnvA) with LH[PV] neurons. Scale bars: 20 μm. (c) Proportion of LH[PV] neurons that express or do not express RVdG-mCherry(EnvA) in *Slc17a6*[Cre] or *Slc32a1*[Cre] mice. LH[PV] neurons were connected to a greater proportion of vlPAG[VGLUT2] neurons than vlPAG[VGAT] neurons (chi-square = 11.18, p=0.0008).

## Activation of LH[PV] axonal projections to the LHb triggers aversion

LH[PV] neurons also target other brain regions including the LHb (*Kisner et al., 2018*; *Celio et al., 2013*). Therefore, we examined whether LH[PV] neurons also modulate nociceptive processing via projections to the LHb. For this, we bilaterally injected a Cre recombinase-dependent viral vector driving the expression of either channelrhodopsin (ChR2:tdTomato) or the fluorophores GFP or tdTomato (control) into the LH of *Pvalb*[Cre] mice and implanted optical fibers bilaterally above the LHb to specifically activate the LH[PV]→LHb pathway (*Figure 6a*). Interestingly, activation of this LH[PV]→LHb circuitry did not evoke changes in nociceptive responses to an acute noxious thermal or mechanical stimulus in healthy mice (*Figure 6b–d*) or in mice with SNI-induced neuropathy when tested at 5 and 25 days post-surgery (*Figure 6—figure supplement 1*). Because the LHb is a brain region associated with reward- and aversion-related behaviors (*Stamatakis et al., 2016*; *Faget et al., 2018*), we also sought to determine whether activation of the LH[PV]→LHb pathway triggers such behaviors. We found that activation of this pathway in healthy mice was aversive as mice spent significantly less time on the photostimulation-paired side in the RTPP assays (*Figure 6e–g*). These results are consistent with previous findings demonstrating that broad activation of lateral hypothalamic glutamatergic axonal projections in the LHb is aversive (*Stamatakis et al., 2016*). Together, these findings demonstrate that LH[PV] neurons encode for distinct behavioral outputs depending on their targeted downstream regions: nociceptive processing via projections to the vlPAG and aversion-related behaviors through connections to the LHb.

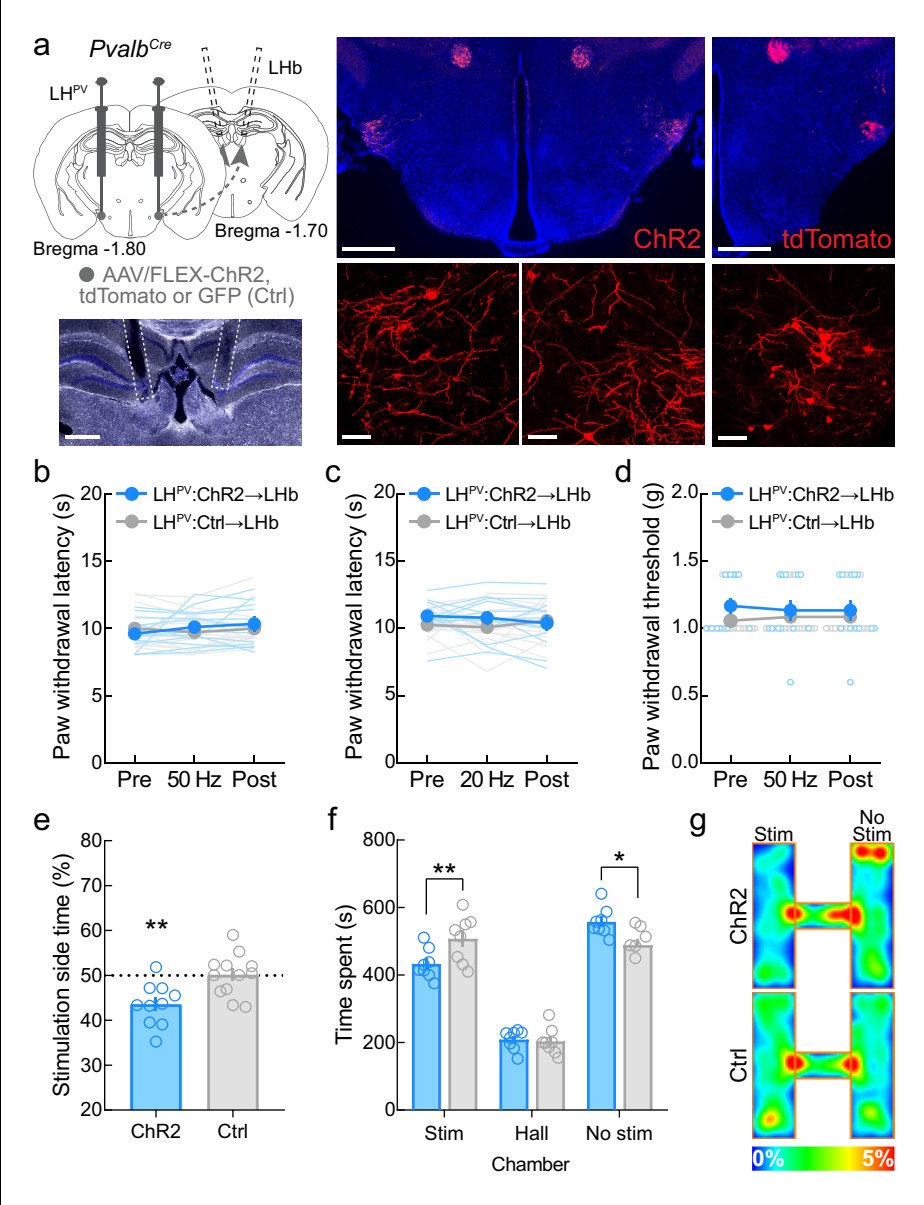

**Figure 6.** Activation of the LH$^{PV}$→LHb pathway triggers aversion but not antinociception. (**a**) Representative images of ChR2 and tdTomato expression in LH$^{PV}$ neurons and optical fiber implants above the lateral habenula (LHb). Scale bars: 500 μm, widefield; 50 μm, zoom. (**b**) Optogenetic activation of LH$^{PV}$ axonal projections in the LHb does not alter thermal nociception at 50 Hz photostimulus frequency (p=0.16) or (**c**) 20 Hz photostimulus frequency (p=0.23) in healthy mice ($n$ = 13 ChR2 mice and 14 Ctrl [GFP/tdTomato] mice). (**d**) Optogenetic activation of LH$^{PV}$ axonal projections in the LHb does not alter mechanical nociception at 50 Hz (p=0.15) in healthy mice ($n$ = 13 ChR2 mice and 14 Ctrl mice). (**e**) Optogenetic activation of the LH$^{PV}$→LHb pathway evokes significant real-time place aversion (p=0.0041; Cohen's $d$ = 1.38) in a standard rectangular one-chamber testing apparatus ($n$ = 10 ChR2 mice and 12 Ctrl mice). (**f**) Optogenetic activation of the LH$^{PV}$→LHb pathway also evokes real-time place aversion in a three-chamber testing apparatus ($n$ = 8 mice per group); two-way mixed-model ANOVA group × chamber interaction, $F_{(2, 28)}$ = 6.22, p=0.0058, Bonferroni's multiple comparisons post-test revealed the LH$^{PV}$:ChR2→LHb group spent less time in the photostimulation chamber (p=0.0089; Cohen's $d$ = 1.28) and more time in the no photostimulation chamber (p=0.016) than LH$^{PV}$:Ctrl→LHb control mice, but no differences were observed in hall zone occupancy (p>0.99). (**g**) Representative heatmaps of LH$^{PV}$:ChR2→LHb and LH$^{PV}$:Ctrl→LHb mice in a three-chamber real-time place preference session. See also *Figure 6—figure supplement 1*.

The online version of this article includes the following figure supplement(s) for figure 6:

*Figure 6 continued*

**Figure supplement 1.** Effects of optogenetic activation of LH$^{PV}$ axonal projections in the lateral habenula (LHb) following complete Freund's adjuvant (CFA) inflammatory pain induction.

## Antinociceptive interactions between LH$^{PV}$ neuronal activation and morphine

Since activation of LH$^{PV}$ neurons appears to reduce nociception as monotherapy, we last sought to examine the interaction between the antinociception induced by these neurons and the μ-opioid pain reliever morphine. For this, we performed a dose-addition analysis of CNO and morphine in LH$^{PV}$:hM3D and LH$^{PV}$:Ctrl mice. First, we determined the individual dose-response curves of CNO and morphine using a cumulative dosing procedure. As expected, CNO evoked dose-dependent PWL$_{HP}$ increases in LH$^{PV}$:hM3D (*Figure 7a*) but not LH$^{PV}$:Ctrl mice (*Figure 7b*), whereas morphine produced dose-dependent increases in both groups (*Figure 7a, b*). Next, the two drugs were combined in fixed proportions (1:1, 1:3, and 3:1) according to their relative potencies (ED$_{50}$) in the LH$^{PV}$:hM3D group. For example, the 1:1 ratio consisted of one unit of the morphine ED$_{50}$ (10.31 mg/kg) for every one unit of the CNO ED$_{50}$ (0.78 mg/kg). Fractions of these mixtures (e.g., the combined 0.125 ×, 0.25 ×, 0.5 ×, and 1 × ED$_{50}$ values of morphine and CNO) were administered consecutively by a cumulative dosing procedure to complete one dose-response curve test (*Figure 7a, b*). The shared dose-response curves were used to calculate the ED$_{50}$ of each drug within each mixture; these equi-effective points were plotted on an isobologram to visualize the nature of each interaction (*Figure 7c, d*). For LH$^{PV}$:hM3D mice, 1:3 and 1:1 morphine:CNO combinations fell within the range of additivity. Remarkably, the 3:1 morphine:CNO combination fell below the range of additivity, suggesting synergistic interactions between morphine and LH$^{PV}$ neuronal activation, indicating that activation of LH$^{PV}$ neurons enhanced the antinociceptive potency of morphine. Formal statistical comparison of expected and experimental ED$_{50}$ values confirmed this observation (Student's paired t-test: $t(7) = 2.92$, p=0.022). For LH$^{PV}$:Ctrl mice, no combinations significantly differed from the range of additivity, suggesting that CNO did not affect the antinociceptive potency of morphine in control subjects.

Finally, using the same mice, we investigated the effects of LH$^{PV}$ neuronal activation following the development of morphine tolerance. We administered morphine (32 mg/kg, i.p.) twice per day for 3 days, which caused a significant decrease in morphine-induced antinociception (*Figure 7e*). On day 5, CNO (1 mg/kg) evoked a significant increase in PWL$_{HP}$ and restored morphine-induced antinociception as compared to control mice (*Figure 7f*). We then treated these mice once per day over the following three days with a combination of 1 mg/kg CNO and 32 mg/kg morphine to assess the potential development of tolerance to this combination. However, no differences were observed on the day 9 test in LH$^{PV}$:hM3D mice as compared to day 5 (*Figure 7f*). Thus, activating LH$^{PV}$ neurons not only increases morphine potency acutely but also rescues morphine tolerance and may prevent subsequent tolerance development.

## Discussion

The LH is an important site for numerous survival-critical processes such as sleep, feeding, and reward (*Carter et al., 2009*; *Bonnavion et al., 2016*; *Stuber and Wise, 2016*). New technologies have enabled the identification of specific lateral hypothalamic populations associated with certain behaviors and the understanding of how the activity of such neurons drives behavior and relates to external factors. However, the cell types mediating many other LH-associated behaviors have received less attention. Nociception has historically been a less LH-prototypical process than one such as feeding, but LH circuits were nevertheless previously shown to respond to noxious stimuli, to control nociception, and to affect downstream circuits in the PAG, a critical brain region for pain regulation (*Cox and Valenstein, 1965*; *Lopez et al., 1991*; *Dafny et al., 1996*; *Fuchs and Melzack, 1995*; *Behbehani et al., 1988*). Cell-type-specific optogenetic manipulations showed that a small cluster of fast-spiking glutamatergic LH$^{PV}$ neurons projects to the vlPAG and modulates acute nociception in a μ-opioid-independent manner (*Siemian et al., 2019*). However, much remained to be

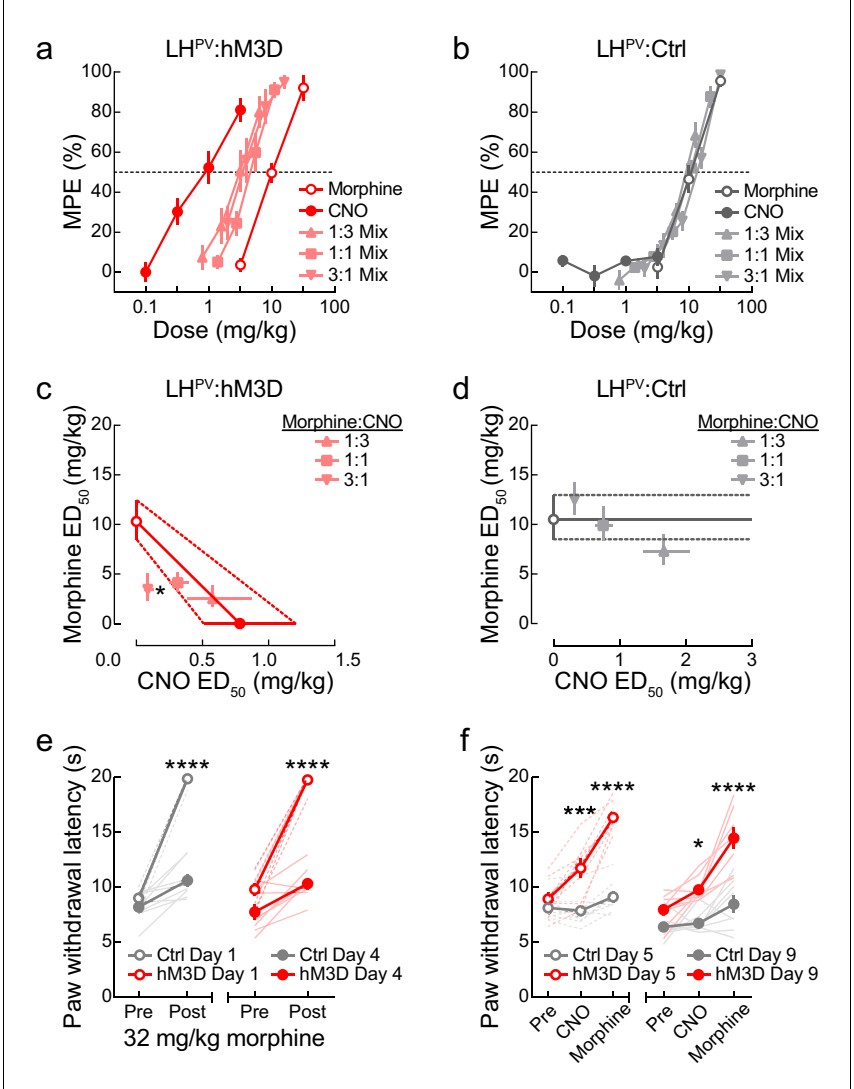

**Figure 7.** Antinociceptive interactions between LH[PV] neuronal activation and morphine. (**a**) Dose-response curves of clozapine-*N*-oxide (CNO) and morphine alone or in combinations of different fixed proportions in LH[PV]:hM3D and (**b**) LH[PV]:Ctrl mice in the hot plate test (*n* = 8 mice per group). (**c**, **d**) Isobolograms constructed from the data shown in panels (**a**) and (**b**). Each point represents the $ED_{50}$ ± 95% CI of each drug alone or in a mixture; ordinates represent the $ED_{50}$ value of morphine and abscissae represent the $ED_{50}$ value of CNO. In LH[PV]:hM3D mice, the 3:1 morphine:CNO mixture was significantly more potent than predicted by the hypothesis of additivity (paired Student's t-test, $t(7)$ = 2.92, p=0.022). (**e**) Both groups of mice developed significant antinociceptive tolerance to 32 mg/kg morphine when administered twice per day for 3 days. Three-way mixed-model ANOVA revealed a significant morphine × test interaction (*n* = 8 mice per group; $F(1, 14)$ = 134.7, p<0.0001), and Bonferroni multiple comparisons post-tests showed the antinociceptive effects of 32 mg/kg morphine were significantly lower on day 4 than day 1 (both p<0.0001). (**f**) Activation of LH[PV] neurons restored morphine potency, and further tolerance did not develop to combination treatment. Three-way mixed-model ANOVA revealed a significant treatment × group interaction (*n* = 8 mice per group; $F(2, 28)$ = 42.10, p<0.0001). Bonferroni multiple comparisons post-tests revealed that there were between-group differences in $PWL_{HP}$ evoked on day 5 by CNO (p=0.0006) and morphine (p<0.0001) and on day 9 by CNO (p=0.016) and morphine (p<0.0001). However, no within-group differences were observed between day 5 and 9 in LH[PV]:hM3D mice during CNO (p>0.99) or morphine treatment (p>0.99).

learned as to how LH[PV] neurons respond to noxious events and whether they could be targeted for therapies in scenarios outside of acute sensory stimulation.

One of the great challenges in understanding how dynamics in neuronal circuits control behavioral output is to determine when specific cell types are active, as well as the nature of the

relationship between this activity and behavior. Although direct manipulations of neuronal activity followed by behavioral examination are important for understanding this relationship, measuring changes in the activity patterns of neurons in awake behaving mice provides information as to how this circuit functions in the absence of experimenter-driven input. Using functional imaging to measure calcium dynamics, we gained insight as to how the activity of LH$^{PV}$ neurons correlates with nociception and show for the first time that LH$^{PV}$ neurons exhibit an array of time-locked responses to acute noxious thermal events. The involvement of LH$^{PV}$ neurons in holding information related to noxious events is supported by the finding that the neuronal activity could be used to decode noxious from innocuous stimuli. LH$^{PV}$ neuronal activity was also altered during formalin-induced inflammation. Formalin injection, which causes discrete phases of acute and inflammatory forms of pain behavior, into the hindpaw contralateral to the imaged LH hemisphere evoked increases in calcium transient frequencies that were more pronounced than those observed during injection of the paw ipsilateral to the imaged LH hemisphere, likely reflecting decussation of the nociceptive signal at the spinal level (*Dafny et al., 1996*; *Yoshida et al., 2019*; *Yamada et al., 2012*). It is worthwhile to note the advantages of using a single-photon miniscope, which enables single-cell resolution of neuronal activity. Other methods such as fiber photometry would likely not have revealed the changes we observed in LH$^{PV}$ neuronal activity to acute thermal stimuli, for which there were heterogeneous responses across neurons, as well as to formalin, which was reflected as an elevated rate of calcium transients that were asynchronous across neurons. Together, our functional imaging data suggest that LH$^{PV}$ neurons may become active during noxious events to signal or suppress nociception in mice.

In chemogenetic experiments designed to assess the broader therapeutic potential of LH$^{PV}$ neuron manipulation in pain disorders, we found that these neurons can bidirectionally modulate thermal nociception over long timescales, and thus may represent potential targets for extended duration analgesia. Moreover, activation of LH$^{PV}$ neurons significantly attenuated acetic acid-reduced nesting behavior, demonstrating that this manipulation not only decreases sensory pain but also permits the resumption of species-specific natural behaviors that are suppressed by noxious events. This model may be analogous to clinical interventions that allow patients undergoing chronic pain to resume daily activities such as exercising or performing occupational duties as opposed to removing pain at the expense of a reduced motivational capacity. We also observed that inhibition of LH$^{PV}$ neuronal activity in hM4D-expressing mice did not decrease nesting behavior in control tests, suggesting that inhibition of these neurons does not cause pain directly, but likely rather enhances sensitivity to noxious stimuli. In support of this, LH$^{PV}$ neuronal activation reduced formalin-associated negative affective pain, whereas this was nearly enhanced by inhibiting LH$^{PV}$ neurons. Behavioral scoring showed that sensory pain behavior was also bidirectionally modulated in this experiment, suggesting that LH$^{PV}$ neurons modulate both sensory and affective experiences.

Optogenetic activation of LH$^{PV}$ neurons decreased both thermal and mechanical nociception following the induction of a commonly used inflammatory pain model. These effects were likely attributable to LH$^{PV}$ projections to the vlPAG as LH$^{PV}$→vlPAG activation decreased thermal and mechanical thresholds in neuropathic and inflammatory models. In contrast, projections of LH$^{PV}$ neurons to the LHb regulated aversion as previously shown for the broader LH glutamatergic population (*Stamatakis et al., 2016*), but not nociception, suggesting that LH$^{PV}$ neurons regulate different behavioral outputs via different downstream projection areas. While systemic CB1 or μ-opioid antagonism does not affect LH$^{PV}$→vlPAG activation-induced antinociception (*Siemian et al., 2019*), the finding that sustained antinociception following extended LH$^{PV}$ somatic or LH$^{PV}$→vlPAG activation suggests that LH$^{PV}$ neurons may co-release neuropeptides that interact with downstream receptors to attenuate nociception or that other efferent circuits for antinociception are recruited during such activation that may function to decrease nociception. Importantly, in conjunction with the functional imaging data, the increased sensitivity observed upon inhibition of LH$^{PV}$ somas or the LH$^{PV}$→vlPAG pathway suggests that LH$^{PV}$ neurons may become active in response to a noxious stimulus to decrease its severity. It is important to briefly note that while a study found that sustained activation of archaerhodopsin evokes spontaneous synaptic release in ex vivo preparations (*Mahn et al., 2016*), this phenomenon has not been observed during in vivo electrophysiological recordings or precluded the observation of behavioral effects in the direction associated with the loss of presynaptic input when using photoinhibition times equal to or longer than the ones we employed here (*Jennings et al., 2013*; *Rozeske et al., 2018*).

Although LH[PV] neurons are functionally connected to neuronal circuits within the vlPAG, the heterogeneous behavioral effects driven by the intermingled vlPAG neuronal populations made it challenging to draw a clear circuit map from LH[PV] neurons to behavior through the vlPAG pathway. For instance, activation of vlPAG glutamatergic and GABAergic neurons decreased and increased nociception, respectively (*Samineni et al., 2017*). Therefore, we used a retrograde monosynaptic rabies tracing strategy to identify the preferred post-synaptic vlPAG targets of LH[PV] neurons. We found a higher proportion of LH[PV] neurons labeled following uptake of RVdG-mCherry(EnvA) in vlPAG[VGLUT2] compared to vlPAG[VGAT] neurons, suggesting that LH[PV] neurons may preferentially, yet not exclusively, target glutamatergic vlPAG neurons. In the context of previous work, excitatory input from LH[PV] neurons to vlPAG[VGLUT2] neurons would thus form a discrete antinociceptive pathway. However, experiments using techniques such as ChR2-assisted circuit mapping (CRACM) from LH[PV]:ChR2+-axonal projections onto postsynaptic vlPAG neurons followed by single-cell RT-qPCR analysis will be needed to elucidate how LH[PV] neurons regulate vlPAG microcircuitry and how activation of this LH[PV]→vlPAG pathway modulates nociceptive responses to noxious stimuli. The opposing behavioral outcomes during photostimulation of a GABAergic LH population, LH[LEPR] neurons, in the vlPAG further demonstrate the complex, heterogeneous nature of LH→PAG pathways and highlight the need for future circuit characterizations. Moreover, it is still unknown whether LH[PV] axonal projections to their target regions follow a one-to-one or one-to-many architecture. This is certainly an important question that has yet to be determined. However, our behavioral data suggest that these might be independent LH[PV] populations since we did not observe aversive-like effects during LH[PV]→vlPAG stimulation or antinociception during LH[PV]→LHb stimulation. Furthermore, the LH[PV]→LHb data also demonstrate that the antinociception evoked by activating the LH[PV]→vlPAG pathway was not due to antidromic stimulation effects. Future experiments will be needed to determine that these are indeed independent populations of LH[PV] neurons.

In a final series of experiments, we investigated the antinociceptive interactions between LH[PV] neuronal activation and the μ-opioid receptor agonist morphine. For a novel analgesic therapy to be useful, it must meet one of these three criteria: (1) possess analgesic properties alone, (2) facilitate analgesic action of existing treatments, or (3) decrease unwanted effects of existing treatments to make them more suitable for extended use (*Li and Zhang, 2011*). Our observation that LH[PV] neuronal activation attenuates nociception suggests that this manipulation meets the first criterion. Therefore, our last experiments were designed to assess the remaining criteria. To address the second criterion, we performed a dose-addition analysis between morphine and LH[PV] DREADD receptor activation by CNO as a standard pharmacological agent. We found that, depending on the proportion of drugs in the mixture, LH[PV] neuronal activation and morphine produced additive to synergistic interactions on thermal antinociception. The combination exhibiting the highest level of synergism required only a small stimulation of LH[PV] neuronal activity to greatly enhance morphine's potency. Importantly, we included a group of control mice without hM3D receptors, in which CNO did not alter morphine's potency. To address the third criterion above, we last investigated the effects of activating LH[PV] neurons following the development of tolerance to morphine-induced antinociception effect in the hot plate test. Here, chemogenetic activation of LH[PV] neurons evoked significant antinociception in morphine-tolerant mice, and more importantly, significantly restored morphine-induced antinociception. Remarkably, similar antinociceptive effects were maintained through another period of concurrent LH[PV] neuronal activation and morphine administration. Together, these findings show that LH[PV] neuronal activation can synergistically enhance acute morphine antinociception and restore its antinociceptive effects following the development of tolerance. Thus, activation of these LH[PV] neurons could be used to reduce the effective antinociceptive dose of morphine, helping to attenuate unwanted side effects such as respiratory depression and slow the rate of morphine tolerance.

An important point warranting further discussion is the contrast between our observations of divergent clusters of response patterns in LH[PV] neuronal activity during the hot and cold plate tests and the dominant behavioral phenotype of pain suppression when bulk activating these neurons. First, there is precedent to this contrast between diverse spontaneous activity patterns and more uniform behaviors driven by causal manipulations. For instance, LH[GABA] neurons responded to food locations, appetitive, or consummatory behaviors in a heterogeneous manner by either increasing or decreasing in activity at each event, yet when these neurons were bulk activated, mice ate voraciously, and when these neurons were ablated, mice ate less (*Jennings et al., 2015*). In the current

study, we observed that most neurons identified as cluster 1 or 2 in the hot plate assay did not maintain the same designation in the cold plate assay, and vice versa. Therefore, it seems likely that, in general, the responses of LH$^{PV}$ neurons are either not consistent over time or dependent on the type of stimulus applied. For instance, some LH$^{PV}$ neurons may specifically suppress heat pain, others may suppress cold pain, others may suppress chemical pain, and so on. As such, we think that the bulk activation of LH$^{PV}$ neurons with optogenetics or chemogenetics during stimulation with one specific noxious stimulus (e.g., heat) likely activates a stimulus-specific cluster of LH$^{PV}$ neurons (heat) as well as the clusters specific for other stimuli (cold, chemical, mechanical, etc.) to evoke antinociception, as opposed to having populations of LH$^{PV}$ neurons that are exclusively pronociceptive or antinociceptive, the effects of which could be potentially diluted during bulk activation of these neurons. However, further work will be needed to elucidate how noxious stimuli are responded to and encoded by LH$^{PV}$ neuronal activity.

Here, we provide a detailed characterization of LH$^{PV}$ neurons, clearly demonstrating that these neurons modulate nociception through a distinct downstream circuit. Moreover, we measured and correlated LH$^{PV}$ neuronal activity patterns during noxious events. Finally, we found that chemogenetic modulation of these neurons could potentially be used as a standalone analgesic therapy or in combination with current analgesics such as morphine. These results support the continued investigation of LH$^{PV}$ neurons as a target for novel analgesics and warrant new efforts to identify neuronal populations in humans for targeting in clinical settings.

# Materials and methods

## Key resources table

| Reagent type (species) or resource | Designation | Source or reference | Identifiers | Additional information |
|---|---|---|---|---|
| Genetic reagent (*Mus musculus*; male/female) | *Pvalb$^{Cre}$* | The Jackson Laboratory | RRID:IMSR_JAX:008069 | C57BL/6J background |
| Genetic reagent (*M. musculus*; male/female) | *Slc32a1$^{Cre}$* | The Jackson Laboratory | RRID:IMSR_JAX:028862 | C57BL/6J background |
| Genetic reagent (*M. musculus*; male/female) | *Slc17a6$^{Cre}$* | The Jackson Laboratory | RRID:IMSR_JAX:028863 | C57BL/6J background |
| Genetic reagent (*M. musculus*; male/female) | *Lepr$^{Cre}$* | M.G. Myers Jr., University of Michigan Medical School | RRID:IMSR_JAX:032457 | C57BL/6J background |
| Antibody | Anti-DsRed, rabbit polyclonal | Takara Bio, Inc | Cat # 632496 RRID:AB_10013483 | (1:1000) |
| Antibody | Anti-parvalbumin (PVALB), guinea pig polyclonal | Swant | Cat # GP72; RRID:AB_2665495 | (1:300) |
| Antibody | Anti-rabbit Alexa Fluor 488, goat polyclonal | Thermo Fisher Scientific | Cat # A11034 RRID:AB_2576217 | (1:500) |
| Antibody | Anti-guinea pig Alexa Fluor 488, donkey polyclonal | Jackson ImmunoResearch Laboratories | Cat # 706-545-148; RRID: AB_2340472 | (1:500) |
| Antibody | Anti-guinea pig Alexa Fluor 647, donkey polyclonal | Jackson ImmunoResearch Laboratories | Cat # 706-605-148; RRID: AB_2340476 | (1:500) |
| Recombinant DNA reagent | rAAV2/9-CAG-FLEX-GCaMP6s-WPRE-SV40 | Addgene | RRID:Addgene_10084; Addgene viral prep 100842-AAV9 | $5.0 \times 10^{12}$ GC/ml |
| Recombinant DNA reagent | rAAV2/rh10-hSYN-DIO-hM3D(Gq)-mCherry | University of North Carolina (UNC) Vector Core | RRID:Addgene_44361 | $2.0 \times 10^{12}$ GC/ml |
| Recombinant DNA reagent | rAAV2/rh10-hSYN-DIO-hM4D(Gi)-mCherry | UNC Vector Core | RRID:Addgene_44362 | $2.0 \times 10^{12}$ GC/ml |

*Continued on next page*

*Continued*

| Reagent type (species) or resource | Designation | Source or reference | Identifiers | Additional information |
|---|---|---|---|---|
| Recombinant DNA reagent | rAAV2/9-hSYN-DIO-mCherry | Addgene | RRID:Addgene_50459; Addgene viral prep 50459-AAV9 | $2.1 \times 10^{13}$ GC/ml |
| Recombinant DNA reagent | rAAV2/1-CAG-FLEX-rev-ChR2-tdTomato | Addgene | RRID:Addgene_18917; Addgene viral prep 18917-AAV1 | $6.9 \times 10^{12}$ GC/ml |
| Recombinant DNA reagent | rAAV2/9-CAG-FLEX-ArchT-GFP | UNC Vector Core | RRID:Addgene_29777 | $4.7 \times 10^{12}$ GC/ml |
| Recombinant DNA reagent | rAAV2/9-CAG-FLEX-GFP | University of Pennsylvania (U Penn) Vector Core | RRID:Addgene_51502 | $3.3 \times 10^{13}$ GC/ml |
| Recombinant DNA reagent | rAAV2/1-CAG-FLEX-tdTomato | U Penn Vector Core | RRID:Addgene_51503 | $4.5 \times 10^{13}$ GC/ml |
| Recombinant DNA reagent | rAAV2/9-CAG-FLEX-tdTomato | U Penn Vector Core | RRID:Addgene_51503 | $4.1 \times 10^{13}$ GC/ml |
| Recombinant DNA reagent | rAAV2/8-hSYN-FLEX-TVA-Rabies B19G (TVA+) | Michigan Diabetes Research Center Molecular Genetics Core, University of Michigan | | $4 \times 10^{12}$ GC/ml |
| Recombinant DNA reagent | EnvA-ΔG-Rabies-mCherry | Michigan Diabetes Research Center Molecular Genetics Core, University of Michigan | | $1 \times 10^{10}$ pfu/ml |
| Chemical compound, drug | Clozapine *N*-oxide (CNO) | Tocris Bioscience | Cat # 4936; PUBCHEM:135445691 | |
| Chemical compound, drug | Acetic acid | Sigma-Aldrich | Cat # 320099; PUBCHEM:176 | |
| Chemical compound, drug | Formalin | Macron Fine Chemicals | Cat # 5016–02; PUBCHEM:712 | |
| Chemical compound, drug | Complete Freund's adjuvant (CFA) | Sigma-Aldrich | Cat # F5881 | |
| Chemical compound, drug | Morphine | National Institute on Drug Abuse Drug Supply Program | PUBCHEM:5288826 | |
| Software, algorithm | ANY-maze video tracking system v5 | Stoelting Co. | RRID:SCR_014289 | |
| Software, algorithm | Doric Neuroscience Studio v5.1 | Doric Lenses Inc | RRID:SCR_018569 | |
| Software, algorithm | FIJI/ImageJ v1.52p | https://imagej.net/Fiji | RRID:SCR_002285 | |
| Software, algorithm | Prism 8 | GraphPad | RRID:SCR_002798 | |
| Software, algorithm | Miniscope Analysis Pipeline | *Etter, 2021* | | |
| Software, algorithm | CellReg | *Sheintuch et al., 2021* | | |
| Software, algorithm | MATLAB | MathWorks | RRID:SCR_001622 | R2019a & R2020a |
| Other | Snap-in Imaging Cannula Model L-V | Doric Lenses Inc | | GRIN lenses |
| Other | Basic Fluorescence Snap-In Microscopy System – Deep Brain | Doric Lenses Inc | | In vivo imaging system |

Further information and requests for resources and reagents should be directed to and will be fulfilled by Yeka Aponte (yeka.aponte@nih.gov).

## Experimental model and subject details

### Animals

All experimental protocols were conducted in accordance with the National Institutes of Health Guide for the Care and Use of Laboratory Animals and with the approval of the National Institute on Drug Abuse and Michigan State University Animal Care and Use Committees. Male and female heterozygous *Pvalb*$^{Cre}$ mice (RRID:IMSR_JAX:008069; C57BL/6J background, The Jackson Laboratory, Bar Harbor, ME, USA), *Slc32a1*$^{Cre}$ mice (RRID:IMSR_JAX:028862, C57BL/6J background, The Jackson Laboratory), *Slc17a6*$^{Cre}$ mice (RRID:IMSR_JAX:028863, C57BL/6J background, The Jackson Laboratory), and *Lepr*$^{Cre}$ mice (RRID:IMSR_JAX:032457; C57BL/6J background, kindly provided by M.G. Myers Jr., University of Michigan Medical School, MI, USA) were used in this study. Mice were maintained at the National Institute on Drug Abuse animal facility under standard housing conditions. Up to five mice of the same sex were group housed under a 12 hr light-dark cycle at 20–24°C and 40–60% humidity with free access to water and food (PicoLab Rodent Diet 20, 5053 tablet, LabDiet/Land O'Lakes Inc, St. Louis, MO, USA). For behavior experiments, 6- to 8-week-old male and female mice (~18–25 g) were randomly assigned to experimental groups while maintaining littermate or age-matched and gender-matched controls. Following stereotaxic surgeries, mice were individually housed.

In all experiments, biological replicates were defined as 'parallel measurements of biologically distinct samples that capture random biological variation,' and technical replicates were defined as 'repeated measurements of the same sample that represent independent measures of the random noise associated with protocols or equipment' (*Blainey et al., 2014*).

## Surgical procedures

For in vivo functional imaging experiments, mice were anesthetized with isoflurane and placed onto a stereotaxic apparatus (David Kopf Instruments, Tujunga, CA, USA). After exposing the skull by a minor incision, a small hole (<1 mm diameter) was drilled unilaterally (bregma, −1.78 mm; midline, +1.38 mm) for virus injection and GRIN lens insertion. A sterile, beveled 25-gauge needle was inserted into the center of the craniotomy stopping approximately 50 µm above the dorsal-ventral coordinate for the lens implant and remaining in place for 4–5 min to create a path for the implant. Next, rAAV2/9-CAG-FLEX-GCaMP6s-WPRE-SV40 was injected offset from the center of the craniotomy (100 nl; rate: 25 nl/min; RRID:Addgene_100842; Addgene viral prep 100842-AAV9; titer: 5.0 × 10$^{12}$ GC/ml) into the LH of *Pvalb*$^{Cre}$ mice (bregma, −1.78 mm; midline, +1.365 mm; skull surface, −5.38 mm) by a pulled glass pipette (20–30 µm tip diameter) with a micromanipulator (Narishige International USA Inc, Amityville, NY, USA) controlling the injection speed. The injection is offset to avoid damaging the tissue in the lens field of view. After injection, a 500-µm-diameter GRIN lens (Snap-in Imaging Cannula Model L-V; Doric Lenses Inc, Québec, QC, Canada) was lowered into the center of the craniotomy (bregma, −1.78 mm; midline, +1.38 mm; skull surface, −5.28 mm). Implants were affixed to the skull with C&B Metabond Quick Adhesive Cement System (Parkell, Inc, Edgewood, NY, USA). Subsequently, mice were individually housed for 3−4 weeks for post-surgical recovery and viral transduction.

For behavioral experiments, mice were anesthetized with isoflurane and placed onto a stereotaxic apparatus (David Kopf Instruments). After exposing the skull by a minor incision, small holes (<1 mm diameter) were drilled bilaterally for virus injection. For experiments targeting parvalbumin neurons in the LH (LH$^{PV}$), 40 nl of an adeno-associated virus was injected bilaterally (rate: 25 nl/min) into the LH of *Pvalb*$^{Cre}$ mice (bregma, −1.80 mm; midline, ±1.40 mm; skull surface, −5.40 mm) or *Lepr*$^{Cre}$ mice (bregma, −1.50 mm; midline, ±0.90 mm; skull surface, −5.40 mm) by a pulled glass pipette (20–30 µm tip diameter) with a micromanipulator (Narishige International USA Inc) controlling the injection speed.

Viruses used for chemogenetic experiments include (1) rAAV2/rh10-hSYN-DIO-hM3D(Gq)-mCherry (RRID:Addgene_44361; UNC Vector Core viral prep; titer: 2.0 × 10$^{12}$ GC/ml), (2) rAAV2/rh10-hSYN-DIO-hM4D(Gi)-mCherry (RRID:Addgene_44362; UNC Vector Core viral prep; titer: 2.0 × 10$^{12}$ GC/ml), and (3) rAAV2/9-hSYN-DIO-mCherry (RRID:Addgene_50459; Addgene viral prep 50459-AAV9; titer: 2.1 × 10$^{13}$ GC/ml).

Viruses used for optogenetic experiments include (1) rAAV2/1-CAG-FLEX-rev-ChR2-tdTomato (RRID:Addgene_18917; Addgene viral prep 18917-AAV1; titer: 6.9 × 10$^{12}$ GC/ml), (2) rAAV2/9-

CAG-FLEX-ArchT-GFP (RRID:Addgene_29777; University of North Carolina [UNC] Vector Core viral prep; titer: $4.7 \times 10^{12}$ GC/ml), (3) rAAV2/9-CAG-FLEX-GFP (RRID:Addgene_51502; University of Pennsylvania [U Penn] Vector Core viral prep; titer: $3.3 \times 10^{13}$ GC/ml), (4) rAAV2/1-CAG-FLEX-tdTomato (RRID:Addgene_51503; U Penn Vector Core viral prep; titer: $4.5 \times 10^{13}$ GC/ml), or (5) rAAV2/9-CAG-FLEX-tdTomato (RRID:Addgene_51503; U Penn Vector Core viral prep; titer: $4.1 \times 10^{13}$ GC/ml).

For somatic-targeted optogenetic experiments, optical fibers were implanted bilaterally above $LH^{PV}$ somas (bregma, −1.80 mm; midline, ±1.40 mm; skull surface, −5.00 mm; no angle). For experiments targeting $LH^{PV}$ axonal projections within the vlPAG or the LHb, optical fibers were implanted bilaterally at 10° angles above $LH^{PV}$ axonal projections in the vlPAG (bregma, −4.00 mm; midline, ±1.00 mm; skull surface, −2.90 mm) or LHb (bregma, −1.70 mm; midline, ±0.90 mm; skull surface, −2.90 mm). For experiments targeting $LH^{LEPR}$ axonal projections within the vlPAG, optical fibers were implanted bilaterally at 10° angles above $LH^{LEPR}$ axonal projections in the vlPAG (bregma, −4.84 mm; midline, ±0.90 mm; skull surface, −2.50 mm). The axonal projections of $LH^{LEPR}$ neurons in the vlPAG are more posterior than those of $LH^{PV}$ neurons, which is why this more posterior coordinate was used (*Schiffino et al., 2019*; *Leinninger et al., 2009*). Implants were affixed to the skull with cyanoacrylate adhesive and C&B Metabond Quick Adhesive Cement System (Parkell, Inc). Subsequently, mice were individually housed for 3−4 weeks for post-surgical recovery and viral transduction.

## In vivo functional imaging

A miniature microscope with an integrated LED was used to image GCaMP6s fluorescence in $LH^{PV}$ neurons through an implanted GRIN lens (Basic Fluorescence Snap-In Microscopy System – Deep Brain; Doric Lenses Inc). $LH^{PV}$:GCaMP6s mice underwent five imaging sessions (hot plate, cold plate, ipsilateral formalin, contralateral formalin, and no formalin). Before each imaging session, GRIN lenses were briefly cleaned with isopropanol and mice were gently restrained while the snap-in microscope was secured to the baseplate for alignment with the implanted GRIN lens. Mice were then given approximately 5 min to acclimate to the microscope and tether. Grayscale TIFF images were collected at 10 frames per second (100 ms exposure) using Doric Neuroscience Studio software version 5.1 (RRID:SCR_018569). The LED power was calibrated between 10% and 50% (0.2–1.2 mW of 458 nm blue light). At the beginning of each session, imaging was synchronized with behavioral video recordings for later alignment. Sample size estimates were derived from a previous study using miniscope recordings in the hypothalamus (*Betley et al., 2015*).

For the hot plate tests, mice were placed on a 51°C hot plate (IITC Life Science, Woodland Hills, CA, USA) or a room temperature black cardboard surface with similar visual and tactile properties for 10–12 trials per stimulus. Mice were removed from the hot plate when typical behavioral responses were observed (e.g., paw withdrawal or paw licking). For the cold plate tests, mice were placed on a 4°C aluminum block or a room temperature white cardboard surface for 8–9 trials per stimulus. Mice were removed from the cold plate when paw checking or withdrawal responses were observed. 1−2 min interstimulus intervals were used for these tests. For the formalin tests, mice received a 20 µl intra-plantar injection of 2% formalin (Cat # 5016–02; Macron Fine Chemicals/Avantor, Radnor, PA, USA) diluted in saline. 47 min videos were captured, and formalin was injected into one of the hindpaws at the 2 min mark. For the 'no injection' test, no formalin was administered. These tests were separated by at least 5 days to minimize photobleaching and inter-test effects.

### Image processing

Image analyses were performed using MATLAB scripts available in the Miniscope Analysis pipeline (https://github.com/etterguillaume/MiniscopeAnalysis). First, images were motion-corrected using the Non-Rigid Motion Correction (NoRMCorre) package (*Pnevmatikakis and Giovannucci, 2017*) and downsampled spatially and temporally by factors of 3. Motion-corrected, downsampled videos were then processed using Constrained Non-negative Matrix Factorization for Endoscopic data (CNMF-E) to extract individual neural segments, denoise their signals, demix signals from nearby neurons, and deconvolve calcium transients for estimation of neuronal firing (*Friedrich et al., 2017*; *Zhou et al., 2018*; *Pnevmatikakis et al., 2016*).

### Imaging and behavioral analysis

For the hot plate and cold plate tests, filtered traces were Z-score normalized and smoothed with a rolling average of 3 frames. The 30 s activity traces surrounding stimulus presentations (10 s before to 20 s after) were averaged within each stimulus to form an average peri-stimulus activity trace per neuron. Neurons were assigned into clusters for further analysis if they displayed one of the two following phenotypes: (cluster 1) the average peak Z-score amplitude was $\geq 1.0$ and the AUC of the trace following the stimulus was positive, or (cluster 2) the average peak Z-score amplitude was $\leq -1.0$ and the AUC of the trace following the stimulus was negative. The remaining neurons that did not meet either of these criteria were considered non-responsive to the noxious stimuli and were not analyzed further.

For decoding analysis, average traces for each neuron were constructed for hot plate, cold plate, or neutral stimulus trials for the 10 s period following stimulus onset. Principal component analysis was used to reduce the dimensionality of the averaged traces while maintaining 99% of the variance. 80% of the resulting traces from each cluster of neurons (see above) were used to train an SVM classifier in MATLAB (built-in function) to distinguish between activity resulting from a neutral stimulus and a hot or cold plate stimulus. The resulting classifier was used to predict which stimulus generated the remaining traces, and the predictions were compared to the known stimuli labels to determine the accuracy of the classifier. This process was repeated 100 times with random subsets of training data to obtain a distribution of test accuracies. To determine the significance of the test accuracy distribution, the labels of the testing dataset were randomly shuffled 100 times. Each label permutation was compared to the predictions obtained from one of the previously trained classifiers to form a null distribution of chance accuracies. A cumulative Gaussian curve was fit to the cumulative frequency distributions (0.10 bin size) of the test and null accuracies, and the distribution means were compared using the extra sum-of-squares F-test in Prism.

For the formalin experiments, deconvolved signals were used to bin estimated $Ca^{2+}$ transients for every 5 min period of the test. For statistical comparison, we averaged the number of events per 5 min period within each phase of the formalin test (0–5 min, acute; 6–15 min, interphase; 16–45 min, inflammatory).

## Optical manipulations

Optical fiber implants were coupled to patch cords connected to lasers (Doric Lenses Inc) via rotary joints mounted over behavioral testing areas. Optical fiber implants were custom-made and assessed for output efficiency $\geq 80\%$. Laser output was controlled by Doric Neuroscience Studio software version 5.1 (RRID:SCR_018569). For photostimulation experiments, 450 nm laser diodes were used to deliver 5 ms pulses of 10–15 mW light at a frequency of 5–100 Hz. For photoinhibition experiments, 520 nm laser diodes were used to deliver 10–15 mW of constant light.

## Behavioral experiments

Mice were habituated to experimenter handling for 3 days prior to experiments, and all experiments were performed during the light cycle. Mice were acclimated to behavioral rooms for at least 1 hr before experiments began. Across experimental and control groups, mice were gender-matched and age-matched or littermates. By design, sample sizes were 8–12 mice based on (*Bolles and Fanselow, 1980*) previous literature using similar procedures (*Negus et al., 2015*; *Jennings et al., 2015*; *Jennings et al., 2013*; *Siemian et al., 2019*; *Alhadeff et al., 2018*) and (*Tovote et al., 2015*) estimates of exclusion rates following histology. Mice were excluded from analysis if viral expression and fiber placement were not observed in at least one hemisphere after histological assessment (see Histology).

### Pain-suppressed nesting assay

Single-housed mice were tested in their home cages, which were initially supplemented with nestlet. Mice were acclimated to the procedure room for at least 1 hr before testing and had access to food and water in their cages throughout test sessions. At the start of each test, mice were pretreated with saline or CNO (1 mg/kg, i.p.; PUBCHEM:135445691; Cat # 4936; Tocris Bioscience, Minneapolis, MN, USA). After 1 hr, the existing nest was removed from each home cage, and a new nestlet cut into six small, equal-sized pieces was placed into the home cage, distributed across zones

divided by a 3 × 2 grid in the cage. The mouse was then given an i.p. injection of 0.6% acetic acid (PUBCHEM:176; Cat # 320099; Sigma-Aldrich) in saline (10 ml/kg) or saline alone (10 ml/kg) and returned to the home cage. Measurements of the number of nestlet pieces collected were taken at 10, 30, 60, and 100 min post-acetic acid injection (*Negus et al., 2015*) by an experimenter blinded to the treatment group. The data from the 30 min time point were presented. At least 5 days separated tests to minimize inter-test effects (*Negus et al., 2015*).

## Formalin place conditioning

Place conditioning experiments were performed in a two-chamber apparatus separated by a wall with a small door that could be closed with a divider. The chambers were defined by tactile, visual, and olfactory cues. One chamber had a metal grid floor, walls decorated with tan and black alternating vertical stripes, and almond scent. The other chamber had a smooth white floor, walls decorated with white circles on a tan background, and orange scent. The front wall of each chamber remained clear, and sessions were recorded using video cameras aimed through this wall using ANY-maze software. Pilot experiments showed that mice consistently preferred the metal grid side at a rate of 60–70% per 15 min test. In comparison to using an unbiased design, this biased design permits pre-assigning groups at surgery with less potential for mismatched side preference at pretest. All mice used in this study preferred the metal grid side in the 15 min pretest on day 1, and this side was assigned for pairing with formalin treatment.

Over the next 4 days, mice received one training session per day with the center door closed and only one chamber accessible; formalin sessions were video recorded for later behavioral scoring of paw licking behavior by a blinded scorer; some videos were difficult to view the mouse to score licking behavior and were removed from this analysis (two mCherry, three hM3D, and one hM4D). All sessions were preceded by an injection of CNO (1 mg/kg, i.p.; Tocris Biosciences) to control for potential subjective effects of LH$^{PV}$ manipulation in the absence of inflammatory pain. On even days (sessions 2 and 4), mice received an intra-plantar injection of saline (20 μl) in the hindpaw and were immediately placed in the initially non-preferred side for 60 min. On odd days (sessions 3 and 5), mice received a 20 μl hindpaw intra-plantar injection of 2% formalin (Cat # 5016-02; Macron Fine Chemicals/Avantor, Radnor, PA, USA) diluted in saline and were placed in the initially preferred chamber for 60 min. The formalin-treated paw was different on each of the two condition sessions. On day 6, untreated mice were placed back in the testing arena with free access to both chambers and the sessions were analyzed with ANY-maze video tracking system v5 (RRID:SCR_014289; Stoelting Co., Wood Dale, IL, USA).

## Thermal nociception (hot plate test)

A cylindrical plexiglass enclosure was placed on a 51°C hot plate (IITC Life Science). For optogenetic experiments, patch cords were connected, and mice were placed in a holding chamber for an initial 3 min period. Mice were gently transferred to the hot plate and the latency to paw withdrawal (PWL$_{HP}$) was measured. A latency of 20 s was defined as complete analgesia and used as a cutoff time to avoid tissue injury. Following this measurement, mice were removed from the hot plate and photomanipulations commenced for 3 min in the holding chamber after which mice were placed back on the hot plate for a second PWL$_{HP}$ measurement. Photomanipulations ceased for another 3 min period in the holding chamber before a final PWL$_{HP}$ measurement. For frequency-response experiments, this procedure was repeated for each frequency, except only one 3 min 'laser-OFF' period separated photostimulation epochs. For experiments examining the effects of longer photostimulation, 50 Hz photostimulation was delivered every other second over 20 min, and PWL$_{HP}$ was measured at the end of the photostimulation period and at 5, 10, and 20 min post-photostimulation. For experiments examining the effects of rimonabant on photostimulation-induced antinociception, rimonabant (3 mg/kg, i.p., dissolved in a vehicle of 8% Tween-80 in saline; PUBCHEM:5360515; Cat # 9000484; Cayman Chemical, Ann Arbor, MI, USA) was administered in a volume of 10 ml/kg 30 min prior to photostimulation. For chemogenetic experiments, CNO (1 mg/kg, i.p.; PUBCHEM:135445691; Cat # 4936; Tocris Bioscience) was administered after the second PWL$_{HP}$ measurement and measurements were taken periodically after (0.5–72 hr).

## Mechanical nociception (von Frey test)

Mice were habituated for 20 min in cylindrical plexiglass enclosures on a fine mesh grid floor. For optogenetic experiments, patch cords were connected, and mice were placed in a holding chamber for an initial 3 min period. Von Frey filaments ranging from 0.008 g to 4 g were used to determine paw withdrawal threshold ($PWT_{VF}$), which was defined as the lowest strength filament eliciting a behavioral response in at least two out of three applications. Briefly, measurements started with the lowest strength filament, and the filament strength was increased until paw withdrawal responses reliably occurred in at least two out of three applications. This procedure was repeated for each hindpaw in three epochs as described above for $PWL_{HP}$ measurements: pre-photostimulation, photostimulation, and post-photostimulation.

## Real-time place preference

RTPP sessions were performed in a standard rat cage with opaque black siding filled with a thin layer of clean rodent bedding, except for a subset of $LH^{PV} \rightarrow LHb$ mice that were also tested in a three-chamber apparatus consisting of two identical black-walled chambers separated by a narrow hall section, and the entire apparatus was filled with a thin layer of clean rodent bedding. Patch cords were connected, and mice were placed into the chamber. Photostimulation (50 Hz) or photoinhibition was paired with one side of the chamber, which remained constant across all tests. For $LH^{LEPR} \rightarrow vlPAG$ experiments, 20 Hz photostimulation was used (*Schiffino et al., 2019*). Tests lasted for 10 min ($LH^{PV}$ somatic manipulations) or 20 min (axonal projection manipulations). At the end of the sessions, the percentage of time spent on the laser-paired side was calculated by ANY-maze video tracking system v5 (RRID:SCR_014289; Stoelting Co.).

## Persistent inflammatory pain

Following initial behavioral tests after stereotaxic surgery and viral transduction, CFA (Cat # F5881; Sigma-Aldrich, St. Louis, MO, USA) was diluted 1:1 in saline and injected (20 µl) into the plantar surface of one hindpaw under brief isoflurane anesthesia (*Alhadeff et al., 2018*). Behavioral tests resumed 5 days post-CFA.

## Persistent neuropathic pain

Following initial behavioral tests after stereotaxic surgery and viral transduction, the SNI model was used for induction of neuropathic pain. Briefly, under isoflurane anesthesia, the tibial and common peroneal nerves were axotomized while the sural nerve was spared (*Decosterd and Woolf, 2000*; *Suter et al., 2003*). Behavioral tests resumed 5 days post-SNI.

## Dose-addition analysis

For the experiment examining interactions between CNO and morphine, tests were conducted according to a cumulative dosing procedure, in which $PWL_{HP}$ measurements are taken immediately prior to i.p. drug administration, then 60 min after drug administration immediately before the next drug administration. When administered alone, CNO was tested across a dose range of 0.1–3.2 mg/kg, and morphine was tested across a dose range of 3.2–32 mg/kg. For combination tests, these dose-measurement cycles continued until near 100% maximal effect was achieved corresponding to the predetermined cutoff time of 20 s. Raw $PWL_{HP}$ values for CNO and morphine were transformed into percent maximum possible effect (%MPE) values according to the formula %MPE = [(post-drug $PWL_{HP}$ – pre-drug $PWL_{HP}$) / (cutoff time – pre-drug $PWL_{HP}$) $\times$ 100]. %MPEs were averaged within each group (± s.e.m.) and plotted as a function of dose. Log($ED_{50}$) values were determined from the %MPE dose-response curve via linear regression and averaged within the group to calculate the $ED_{50}$ (±95% confidence interval [CIs]) for each drug, except for CNO in the mCherry control group, which did not produce 50% effect levels. Morphine was obtained from the National Institute on Drug Abuse Drug Supply Program (PUBCHEM:5288826).

To examine the antinociceptive interactions between CNO and morphine, a fixed-proportion dose-addition analysis method was used (*Siemian et al., 2018*; *Tallarida, 2010*; *Negus et al., 2009*). For this analysis, CNO and morphine were combined in fixed proportions (1:1, 1:3, and 3:1) and administered using the cumulative dosing procedure as described. The actual doses of the drugs in the combination were determined by the relative potencies of each drug (based on the

ED$_{50}$ values) in the LH$^{PV}$:hM3D group. For example, the 1:1 ratio consisted of one unit of the morphine ED$_{50}$ (10.31 mg/kg) for every one unit of the CNO ED$_{50}$ (0.78 mg/kg). By this method, the 1:3 ratio contained 0.5 × ED$_{50}$ of morphine and 1.5 × ED$_{50}$ of CNO and the 3:1 ratio contained 1.5 × ED$_{50}$ of morphine and 0.5 × ED$_{50}$ of CNO. Fractions of these mixtures (the combined 0.125 ×, 0.25 ×, 0.5 ×, 1 ×, and 2 × ED$_{50}$ values of morphine and CNO) were administered consecutively by the cumulative dosing procedure to complete one dose-effect curve test. At least 1 week separated each test to avoid the development of tolerance and inter-test effects. Furthermore, a morphine-alone dose-response curve was taken 1 week after the last combination test, which showed that the morphine ED$_{50}$ had not significantly changed (mCherry mice first morphine ED$_{50}$ 10.51 mg/kg, second morphine ED$_{50}$ 10.26 mg/kg; hM3D mice first morphine ED$_{50}$ 10.31 mg/kg, second morphine ED$_{50}$ 10.45 mg/kg). The shared dose-response curves were used to calculate the ED$_{50}$ of each drug within each mixture. Isobolograms plotting the ED$_{50}$ values of each drug were constructed to visually represent the nature of the drug interactions as additive, infra-additive, or supra-additive (synergistic).

Dose-addition analysis was performed as described previously (*Siemian et al., 2018*; *Tallarida, 2000*). When both drugs were active in an assay, expected additive ED$_{50}$ values (±95% CL) (Z$_{add}$) were calculated from the equation Z$_{add}$ = fA + (1 − f)B, where A is the ED$_{50}$ of morphine alone, B is the ED$_{50}$ of CNO alone, and f is the fractional multiplier of A in the computation of the additive total dose (e.g., f = 0.5 when fixed ratio was 1:1). When only one drug was active (i.e., morphine in the mCherry control group), the hypothesis of additivity predicts that the inactive drug (i.e., CNO) should not contribute to the effects of the mixture, and the equation reduces to Z$_{add}$ = A/$\rho$A, where $\rho$A is the proportion of morphine in the total drug dose. Experimental ED$_{50}$ values (Z$_{mix}$) were determined from the 1:3, 1:1, and 3:1 combinations and were defined as the sum of the ED$_{50}$ values of both drugs in the combination. Given the within-subject experimental design, Z$_{add}$ and Z$_{mix}$ values were analyzed with paired two-tailed Student's t-tests to determine differences between expected and experimental ED$_{50}$ values.

## Morphine tolerance study

One week after the last morphine-alone dose-response curve, we induced morphine tolerance in LH$^{PV}$:Ctrl and LH$^{PV}$:hM3D mice by administering 32 mg/kg morphine (i.p.) twice per day, separated by approximately 8 hr. We measured PWL$_{HP}$ before and 1 hr after the first injection on day 1 and the first injection on day 4 (the seventh injection overall) to verify tolerance development. On day 5, three PWL$_{HP}$ measurements were taken: pre-injection, 1 hr post-CNO injection, and 1 hr post-morphine injection. On days 6–8, a combined injection of 32 mg/kg morphine and 1 mg/kg CNO was administered once per day. On day 9, the day 5 test was repeated to measure potential development of tolerance to the morphine/CNO mixture.

## Histology

Mice were deeply anesthetized with isoflurane and transcardially perfused with 1× phosphate buffered saline (PBS) followed by 4% paraformaldehyde (PFA) in 1× PBS. Whole brains were removed and post-fixed in 4% PFA overnight at 4°C and subsequently transferred to 1× PBS for storage at 4°C until further processing. Coronal brain sections (50 μm thick) were collected in 1× PBS using a Leica VT1200 vibratome (Leica Biosystems GmBH, Wetzlar, Germany). In some instances, DsRed immunostaining was required to visualize viral transduction. Sections were blocked for 1 hr at room temperature in 1× PBS with 0.3% Triton X-100% and 3% normal goat serum. After blocking, sections were incubated with rabbit anti-DsRed antibody (1:1000 Cat # 632496/RRID:AB_10013483; Takara Bio, Inc, Mountain View, CA, USA) in block solution for 20 hr at 4°C. Tissue was then washed 4 × 10 min in 1× PBS followed by incubation in goat anti-rabbit Alexa Fluor 488 antibody (1:500 Cat # A11034/RRID:AB_2576217; Thermo Fisher Scientific, Waltham, MA, USA) in block solution for 1.5 hr at room temperature. After secondary antibody incubation, sections were washed 4 × 10 min in 1× PBS. All sections were mounted with DAPI-Fluoromount-G aqueous mounting medium (Electron Microscopy Sciences, Hatfield, PA, USA) onto Superfrost Plus glass slides (VWR International, Radnor, PA, USA). Images were taken with an AxioZoom.V16 fluorescence microscope (Carl Zeiss Microscopy LLC, Thornwood, NY, USA).

## Recombinant rabies virus tracing

For retrograde monosynaptic tracing experiments, *Slc32a1*[Cre] and *Slc17a6*[Cre] mice were anesthetized with ketamine/xylazine (90/10 mg/kg i.p.) and placed onto a stereotaxic apparatus (David Kopf Instruments). After exposing the skull by a minor incision, a small hole (<1 mm diameter) was drilled unilaterally for helper virus injection. 40 nl of Cre-dependent AAV8/hSyn-FLEX-TVA-Rabies B19G (TVA+) was injected unilaterally (rate: 10 nl/min; titer: $4 \times 10^{12}$ GC/ml) into the vlPAG (bregma, −3.90 mm; midline, ±0.2 mm; skull surface, −3.20 mm) by a 25-gauge Hamilton syringe (500 nl). 3–4 weeks later, mice were injected with 100 nl of the recombinant rabies viral vector (EnvA-ΔG-Rabies-mCherry; titer: $1 \times 10^{10}$ pfu/ml) at the same vlPAG coordinate. Both viruses were graciously provided by the Michigan Diabetes Research Center Molecular Genetics Core, University of Michigan. 3–4 weeks after the recombinant rabies virus injection, mice were deeply anesthetized with isoflurane and transcardially perfused with 1× PBS followed by 4% PFA in 1× PBS. Whole brains were removed and post-fixed in 4% PFA overnight at 4°C and subsequently cryoprotected by equilibration in 30% sucrose in 1× PBS at 4°C, flash frozen in isopentane on dry ice, and stored at −80°C. Tissue was embedded in Tissue-Tek O.C.T. Compound (Sakura Finetek USA, Inc, Torrance, CA, USA) for cryosectioning. Coronal brain sections (50 µm thick) were collected in 1× PBS using a Leica CM3050 S cryostat (Leica Biosystems GmBH, Wetzlar, Germany). Sample size estimates were derived from a previous study using the same methodology (*Jennings et al., 2013*).

For parvalbumin (PVALB) immunostaining, sections containing the hypothalamus were blocked for 1 hr at room temperature in 1× PBS with 0.3% Triton X-100% and 3% normal donkey serum. After blocking, sections were incubated with guinea pig anti-PVALB antibody (1:300 Cat # GP72; RRID:AB_2665495; Swant, Marly, Switzerland) in block solution for 16 hr at 4°C. Tissue was then washed 4 × 10 min in 1× PBS followed by incubation in donkey anti-guinea pig Alexa Fluor 488 or 647 antibody (1:500 Cat # 706-545-148/RRID:AB_2340472 or Cat # 706-605-148/RRID:AB_2340476; Jackson ImmunoResearch Laboratories, Inc, West Grove, PA, USA) in block solution for 1.5 hr at room temperature. After secondary antibody incubation, sections were stained for 5 min with 4′,6-diamidino-2-phenylindole, dilactate (DAPI 1:5000; Thermo Fisher Scientific) in 1× PBS followed by 3 × 10 min washes in 1× PBS. Sections were mounted with Fluoromount-G aqueous mounting medium (Electron Microscopy Sciences) onto Superfrost Plus glass slides (VWR International). Z-stacks (30 µm) containing the LH[PV] region were imaged with an LSM 700 microscope using a 20× air objective (Carl Zeiss Microscopy LLC). Maximum intensity projections were manually counted using Fiji v1.52p software (RRID:SCR_002285) with the cell counter plugin (*Schindelin et al., 2012*). Sections were anatomically matched to ensure that the same regions were analyzed across samples. Additionally, sections containing the PAG were mounted as described, and images were taken with an AxioZoom.V16 fluorescence stereomicroscope using a 7× digital magnification to assess the injection site for mistargeted or lacking virus expression. After PAG assessment, one *Slc32a1*[Cre] and one *Slc17a6*[Cre] sample was excluded from the analysis as viral expression was not observed in the vlPAG.

## Statistics

Graphs and statistics for behavioral experiments were prepared with GraphPad Prism 8 software (RRID:SCR_002798; GraphPad, La Jolla, CA, USA). All data are plotted as mean ± s.e.m., except for $Ca^{2+}$ event frequency data and isobolograms, which are plotted as mean ± 95% CI, and cell counts, which are plotted in 'part-of-whole' format. Paired or unpaired Student's two-tailed t-tests, one-way, two-way, or three-way mixed model ANOVAs with Bonferroni or Dunnett's post-tests for multiple comparisons corrections were used to analyze all behavioral data, as appropriate. Mann–Whitney *U*-tests with Holm–Sidak correction for multiple comparisons were used to analyze $Ca^{2+}$ event frequency data from the formalin tests. A chi-square test was used to compare cell counts in the rabies tracing experiment. For all statistical tests, p<0.05 was considered significant.

## Acknowledgements

The authors acknowledge with gratitude C Lupica for discussions and comments on the manuscript, MG Myers Jr for kindly providing the *Lepr*[Cre] mice, BT Laing for performing SNI surgeries, T Larson for assistance with imaging, the NIDA IRP Histology Core, in particular L Shen and C Mejias-Aponte,

for technical assistance with histology, and NIDA IRP Visual Media, in particular A Russell and L Brick, for brain slice drawings. Mouse clip art was adapted from Openclipart.org (Creative Commons CC0). Modified rabies tracing vectors were graciously provided by the Michigan Diabetes Research Core, funded by NIH P30-DK020572. AJ Robison and AL Eagle are supported by NIMH R01-111604, NIDA R01-040621, NICHD R01-072968, and NINDS R01-085171. GM Leinninger is supported by NIDDK RO1-DK103808. Y Aponte is supported by the National Institute on Drug Abuse Intramural Research Program (NIDA IRP), U.S. National Institutes of Health (NIH).

## Additional information

### Funding

| Funder | Grant reference number | Author |
| --- | --- | --- |
| National Institute on Drug Abuse | Intramural Research Program | Justin N Siemian<br>Miguel A Arenivar<br>Sarah Sarsfield<br>Cara B Borja<br>Lydia J Erbaugh<br>Yeka Aponte |
| National Institute of Diabetes and Digestive and Kidney Diseases | P30-DK020572 | Andrew L Eagle<br>Alfred J Robison<br>Gina Leinninger |
| National Institute of Diabetes and Digestive and Kidney Diseases | RO1-DK103808 | Gina Leinninger |
| National Institute of Mental Health | R01-111604 | Andrew L Eagle<br>Alfred J Robison |
| National Institute on Drug Abuse | R01-040621 | Andrew L Eagle<br>Alfred J Robison |
| Eunice Kennedy Shriver National Institute of Child Health and Human Development | R01-072968 | Andrew L Eagle<br>Alfred J Robison |
| National Institute of Neurological Disorders and Stroke | R01-085171 | Andrew L Eagle<br>Alfred J Robison |

The funders had no role in study design, data collection and interpretation, or the decision to submit the work for publication.

### Author contributions

Justin N Siemian, Conceptualization, Data curation, Software, Formal analysis, Validation, Investigation, Visualization, Methodology, Writing - original draft, Writing - review and editing; Miguel A Arenivar, Cara B Borja, Andrew L Eagle, Investigation, Writing - review and editing; Sarah Sarsfield, Resources, Formal analysis, Investigation, Visualization, Writing - original draft, Writing - review and editing; Lydia J Erbaugh, Software, Formal analysis, Writing - review and editing; Alfred J Robison, Gina Leinninger, Conceptualization, Resources, Supervision, Funding acquisition, Project administration, Writing - review and editing; Yeka Aponte, Conceptualization, Resources, Formal analysis, Supervision, Funding acquisition, Validation, Visualization, Methodology, Writing - original draft, Project administration, Writing - review and editing

### Author ORCIDs

Justin N Siemian ⓘD https://orcid.org/0000-0003-3337-7333
Sarah Sarsfield ⓘD https://orcid.org/0000-0002-0527-8154
Yeka Aponte ⓘD https://orcid.org/0000-0002-5967-2579

### Ethics

Animal experimentation: All experimental protocols were conducted in accordance with the National Institutes of Health Guide for the Care and Use of Laboratory Animals and with the approval of the

National Institute on Drug Abuse and Michigan State University Animal Care and Use Committees. All of the animals were handled according to approved institutional animal care and use committee protocols (NIDA 19-CNRB-116, 19-CNRB-127, and 20-CNRB-132; MSU 201900103). Surgeries were performed under either isoflurane or ketamine/xylazine anesthesia, and every effort was made to minimize suffering.

## Decision letter and Author response
Decision letter https://doi.org/10.7554/eLife.66446.sa1
Author response https://doi.org/10.7554/eLife.66446.sa2

## Additional files

### Supplementary files
• Transparent reporting form

### Data availability
All data generated or analyzed during this study are included in the manuscript and supporting files. Source data files have been provided for Figure 1.

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
