## [Decision Letter]

**Acceptance summary:**

In this work, Siemian and co-authors investigate a cluster of parvalbumin-expressing excitatory neurons in the lateral hypothalamus and their role in modulating pain states and pain-associated behavior. Overall, this work is well-executed and convincing and will be broadly interesting to both the hypothalamic circuits and pain neuroscience communities, and will contribute to our understanding of the role of hypothalamic neurons in modulating pain responses and pain behavior.

**Decision letter after peer review:**

Thank you for submitting your article "An excitatory lateral hypothalamic circuit orchestrating sensory and affective pain" for consideration by *eLife*. Your article has been reviewed by 5 peer reviewers, including Peggy Mason as the Reviewing Editor and Reviewer #2, and the evaluation has been overseen by Michael Taffe as the Senior Editor. The following individuals involved in review of your submission have agreed to reveal their identity: Robert Gereau (Reviewer #1); Alexander C Jackson (Reviewer #3); Gregory Corder (Reviewer #4); Asaf Keller (Reviewer #5).

Essential revisions:

– Discussion of the inhibited vs excited LH neurons.

– Responses of single neurons to multiple stimuli – coordinated or independent.

– Are there dual projections to PAG and LHab (if no data, then a thorough acknowledgment and discussion of this issue)?

– Concerns over mixed effects of Arch on synaptic transmission.

– Persistent over chronic is preferred for the time scales used here.

– Small sample sizes from low numbers of animals and small effects make the presentation of raw data points all the more desirable.

– Flesh out intro beyond to introduce the component reactions to noxious stimulation that you will discuss – nocifensive, suppression of everyday activities and the intersection with aversion that eventually is discussed with respect to both LH and LHab.

*Reviewer #1:*

This is an interesting study from the Aponte lab that builds on their prior work that identified lateral hypothalamic parvalbumin-expressing, glutamatergic neurons (LH^PV^ neurons) as regulators of pain behaviors in mice. Here, the authors explore how painful stimuli impact activity of the LH^PV^ neurons in vivo, and further examine the impact of manipulating activity of these neurons in the context of inflammatory and neuropathic pain, as well as the effects on affective aspects of pain.

Using miniscope recordings, the authors find that there are two main populations of LH^PV^ neurons – the majority of which are activated in association noxious stimulation, but a subset that are inhibited.

Studies using chemogenetics and optogentics, all with appropriate control groups, define clearly the effects of activating these LH^PV^ neurons, and show that the send important functional projections to the periaqueductal grey.

The manuscript misses a potentially important opportunity in defining what these different classes (pain-activated vs inhibited) neurons in the LH do, but there is still much that we learn here.

One technical issue that needs to be addressed is around the use of ArchT for inhibition of LH→PAG terminals, as ArchT can have complicated actions at terminals.

The authors identify differences in projections to the lateral habenula vs. PAG, which are interesting and suggest different functional outputs. It will be important to understand if these are unique cell populations, or just dual projections from the same population of neurons in the LH.

Overall, this represents a significant contribution to the field. I commend the authors on the thorough characterization of their behaviors and detailed exploration of the optogenetic stimulation frequency space – which is largely missing in most papers. There are a number of technical questions that should be addressed to further strengthen the manuscript, and it is worth considering whether some of the data presented here would be better reserved for a future publication to make the present manuscript more focused on the main points outlined in the manuscript title.

Comments for the authors:

1. The imaging studies in figure 1 are intriguing and provide some very interesting insights – but the presentation raises a number of questions.

a. Overall – the data show that there are LH^PV^ neurons that seem to show increased activity to heat and cold, and some that show decreased activity. The authors call this cluster 1 and 2 respectively. This is interesting, and points to the importance of using the miniscope approach to get cellular level resolution, as it appears that simple fiber photometry may have shown no change in activity in response to these stimuli. This should be highlighted.

b. Similar trends are seen for hot plate and cold plate. If you look at the single cell level – are neurons that classify as cluster 1 for hot plate also in cluster 1 for cold plate? Presumably the same mice were tested for both assays. This is not clear.

c. The z scores show something rather interesting – there is clearly a period of suppressed activity in cluster 1 cells and increased activity in cluster 2 cells preceding contact with the plate. This is evident also in the "control" group. What do the authors think of this? The inclusion of the control groups here was absolutely critical, as it shows that there are indeed differences. Related – the post-hoc analysis states significant differences are illustrated by the bold line – but this is not easy to see. Am I correct in seeing that there is no significant differences in cluster 2 for the cold stimulus vs. control? The manuscript states significant differences, but the posthoc either does not support this or it is not clear from the bold shading of the line in Figure 1l.

d. The cluster 1 neurons are more prevalent, and the authors comment that the response magnitude is larger than those for cluster 2. This should be supported by statistical comparisons validating this claim.

e. For the formalin data – given the results from the hot and cold plate, it is difficult to know how to interpret these findings. Were these cells recorded here specifically cluster 1 cells as determined from the hot and cold plate? Again- were these the same mice tested for hot and cold plate? An analysis of these data for formalin separately for neurons defined as cluster 1 and 2 for hot/cold would be much more informative than including all together.

f. The interpretation of formalin responses as "long-term" seems unwarranted. In fact, the response to formalin is ongoing. So – this would suggest sustained responses to ongoing nociception.

g. Related – it would be of interest to analyze the formalin data as Ca^2+^ transients time-locked to nocifensive responses (flinching, lifting, licking). Did you collect videos of the mice that would allow this analysis? Are there also "cluster 1" and "cluster 2" type responses associated with flinching/licking?

2. The authors go on to study the impact of manipulating LH^PV^ activity on behaviors, but sort of ignore the fact that there are these two populations of neurons with respect to activity in response to noxious stimulation. It is unfortunate that the interesting findings of the two different clusters of LH^PV^ neurons – one inhibited by noxious stimulation and one activated – is not built upon here – but is largely ignored. The subsequent studies go about activating or inhibiting all LH^PV^ neurons to test their effects on sensory and affective aspects of pain. The challenge with this approach, based on the imaging data, is that there is a mixture of activity patterns in response to noxious stimuli. For example – heat responses. As you are targeting LH^PV^ neurons globally – you will be hitting both cluster 1 and cluster 2 neurons. I know that it would be a huge task to sort out what is different about these two populations – but this caveat at least needs significant discussion. It would be very interesting to be able to determine what is different about the cluster 1 and cluster 2 neurons. This is not addressed here, and I think may be beyond the scope of this study. However – this should be discussed. Why are some neurons inhibited, some activated? Do these neurons have different projection targets?

3. Activating the LH^PV^ neurons leads to reductions in responses to painful stimuli, and reversal of pain-suppressed behaviors. Given that the majority of these neurons are activated by painful stimuli, what does this mean for the role of LH^PV^ neurons in pain? You allude to this briefly in the discussion, but this is the major question in my mind in terms of the major conclusions we can make from this nice study. Are LH^PV^ neurons mediating endogenous analgesia during painful stimuli? If so, then this could be mediating a DNIC type response. It would be very interesting to know if you can elicit DNIC in mice, and block this by inhibiting these LH^PV^ neurons.

4. The use of ArchT for photoinhibition of the LH^PV^ → vlPAG terminals is problematic given reports that activation of Arch actually produces increases in spontaneous synaptic release at some terminals. It is imperative that the authors characterize the effects of Arch activation on synaptic transmission (evoked and spontaneous) at this synapse to allow interpretation of the findings.

5. The results re: the projections to the LHb are also very interesting. Are the neurons that project to the LHb and vlPAG distinct, or do individual neurons send process to both LHb and vlPAG? A double retrograde tracing experiment would expand the scope of what we learn here significantly.

*Reviewer #2:*

This is a thorough, multipronged study which supports a role for parvalbumin-positive neurons in the lateral hypothalamus (LH-Pv+) in modulating the nocifensive and affective / motivational consequences of noxious stimulation and morphine suppression of the same. The authors provide evidence that both nocifensive and affective / motivational consequences are suppressed by activation of glutamatergic LH-PV+ neurons whereas activation of GABAergic LH projections (expressing the leptin receptor) result in place aversion.

Experiments also demonstrate that LH-PV+ activation combines in an additive and at one ratio in a synergistic manner with morphine and that activation of LH-PV+ neurons reverses tolerance to morphine.

The nociceptive-specific nature of the sensory responses is not convincing given that there is a comparison between stimulus and not-stimulus rather than between innocuous stimulus and noxious stimulus. The authors may argue that room temperature is a thermal stimulus and perhaps that is true, depending on the animal's skin temperature which was not measured. But a better comparison would be between a 45C or so stimulus and the 51C used. I am not suggesting that the authors add such expts rather I recommend a more careful interpretation of the data collected. This concern is strengthened by the response to ipsilateral formalin which raises the issue of whether the "responses" may be in fact correlates of efferent function involved in autonomic or state arousal. Yet, the remaining experiments mitigate this concern.

In pairing the (idiosyncratically) preferred side with CNO-inhibition, the preference for either side goes to 50%. Thus inhibition of LH-PV+ neurons takes away side preference but does NOT produce side avoidance as is stated in lines 153-54.

Please jog the gray symbols in Figure 4 supp 1, 2 laterally so they can be seen.

The LH-PV+ preference for vglut2 over vgat PAG neurons is by less than a factor of 3. Given what we know about the unimportance of absolute numbers of synapses in dictating the responses of lemniscal pathways in the thalamus, this result is interesting but with this amount of information uninterpretable. Tone this down.

*Reviewer #3:*

In this study, Siemian and co-authors significantly expand upon their previous work (Siemian et al., 2019) examining the role of parvalbumin-positive glutamatergic neurons in the LH (LH^PV^), and their projections to the vlPAG, in the modulation of pain responses. In their previous work, the authors showed that optogenetic activation or inhibition of LH^PV^ neurons suppresses or potentiates, respectively, responses to a noxious thermal stimulus and that this effect may be mediated by an excitatory projection to the vlPAG. The current manuscript is exciting in that it describes a much more detailed investigation of this circuit, using in vivo activity monitoring and manipulation (both chemo- and optogenetic) in multiple models of pain and pain-related behavior (both acute and chronic) as well as a neuroanatomical elucidation of this circuitry. Overall, this work uses a wide-spectrum of modern tools to address the circuit-basis of LH^PV^ modulation of pain states and pain-associated behavior. This well-executed work contributes to filling a significant gap in our understanding of the role of LH neurons in modulating pain responses and pain behavior.

In their current work, the authors start by observing in vivo calcium signals with single cell resolution using GRIN-lens endomicroscopy of LH^PV^ neurons in PV-cre mice in several pain assays. They found that a subset of LH^PV^ neurons were activated by an acute thermal stimulus (hotplate), a much smaller subset were inhibited and large proportion did not respond. In another set of experiments, a subset of neurons were found to be activated by an acute noxious cold stimulus (coldplate), a small subset were inhibited and large proportion did not respond. They also found that some LH^PV^ neurons responded to formalin injection over longer time-scales. These experiments successful provide solid evidence that at least a subpopulation of LH^PV^ neurons respond to noxious stimuli which is a key finding. Less clear is how the heterogeneity observed in these responses may be linked to the circuit manipulations in the subsequent experiments, which engage the whole population of LH^PV^ neurons, and how specific LH^PV^ neurons are in modulating pain pathways.

In the next series of experiments, the authors manipulate LH^PV^ neurons using chemogenetic and optogenetic methods in PV-cre mice, in a variety of standard pain assays. They first found that chemogenetic activation of LH^PV^ neurons suppressed pain responses, while inhibition of LH^PV^ neurons potentiated pain responses (increases and decreases in thermal pain thresholds respectively). In an interesting assay of homecage behavior, they further found that chemogenetic activation of LH^PV^ neurons prevented a pain-associated reduction in both nesting behavior and place aversion, which strengthens the argument that activation of LH^PV^ neurons is antinociceptive and seems to diminish pain-associated aversion. Finally, optogenetic activation of LH^PV^ neurons appeared to diminish thermal and inflammatory pain, consistent with their chemogenetic results. However, optogenetic activation of LH^PV^ neurons also produced mild place avoidance in an RTPP assay, somewhat at odds with their chemogenetic results. Overall, the chemo and optogenetic manipulation experiments succeeded in showing that activation of LH^PV^ neurons has antinociceptive effects.

Next, the authors significantly extended their previous work by further probing the role of LH^PV^ projections to both the vlPAG and LHb. Optogenetic stimulation of LH^PV^ inputs to the vlPAG increased both acute thermal pain and more chronic neuropathic pain thresholds while inhibition decreased it, consistent with direct somatic manipulations. Interestingly, these manipulations did not appear to affect place preference leading to their examination of LH^PV^ projections to the LHb, activation of which was aversive, consistent with previous work showing that the broader population of LH-VGLUT2 to LHb pathway is aversive. To address whether excitatory or inhibitory vlPAG populations may be targeted by LH^PV^ neurons, the authors used rabies tracing and found that although a very small percentage of LH^PV^ neurons appeared to innervate the vlPAG, a greater proportion of excitatory than inhibitory neurons were targeted. Finally, the authors found that LH^PV^-induced antinociception enhanced and morphine-induced antinociception and rescues morphine tolerance.

Overall these data are novel and suggest that even a very tiny cluster of LH neurons has a rather outsized role to play in pain modulation. These findings would contribute to our limited understanding of both the role of hypothalamic circuits in pain modulation and the intersection of reward, aversion and pain.

Comments for the authors:

– Although the in vivo GRIN lens experiments provide evidence that a subpopulation of LH^PV^ neurons respond to noxious stimuli, there's less clarity on the following questions:

1. Does the subpopulation of LH^PV^ neurons activated by noxious thermal stimulation overlap with the subpopulation of neurons activated by noxious cold stimulation, ie is a specific subset of these neurons broadly tuned to respond to noxious thermal and cold stimuli or are they separate? The authors could provide more clarity on whether the cold/hot assays were done in the same animals and if not, provide some commentary on this question.

2. What is the degree of inter-animal variability in the data? Did some animals contribute more or less to the population(s) that responded to thermal/cold stimuli?

3. Given that only a relatively small subpopulation of LH^PV^ neurons are activated by noxious stimuli, with another subpopulation inhibited, how does one interpret the chemo- and optogenetic data in which all LH^PV^ neurons are either activated or inhibited? This should be addressed as a caveat in the discussion.

– The rabies tracing experiment is well-done and informative but is a somewhat indirect way of showing preferential innervation of different vlPAG populations. Directly addressing this is a technically difficult problem and perhaps the subject of a follow-up study but a discussion of the caveats would be helpful. One possible caveat in this experiment is that, for the sake of argument, a simply larger population of excitatory vlPAG neurons may skew the representation of retrogradely-labeled LH^PV^ neurons despite perhaps an equal rate of connectivity with excitatory and inhibitory neurons.

– Another outstanding question is whether the same population of LH^PV^ neurons are innervating both the vlPAG (antinociception) and the LHb (aversion) or if they are entirely different subpopulations. One way of approaching this experimentally is to inject two different conjugated CTBs, one in the vlPAG and the other in the LHb to determine if retrogradely-labeled LH^PV^ neurons overlap or not. If this experiment is not possible, addressing this in the discussion would be helpful.

– In the Discussion, the authors refer to LH^PV^ neurons as a whole in the context of pain modulation when their data actually implicates only a comparatively small subpopulation of these neurons in modulating pain responses, evidenced by their in vivo monitoring and rabies tracing. This language should be adjusted in the discussion given that LH^PV^ neurons are clearly not functionally monolithic. Furthermore, a short discussion of any known heterogeneity among LH^PV^ neurons would be useful here.

– Finally, in terms of the writing, the manuscript is overall quite well-written.

However, the introduction could be improved. The authors start with a general description of pain and pain behavior and quickly transition to the role of the lateral hypothalamus in pain. It would be useful to make that transition more fluid and more detailed given that this is rationale for studying these cells in the context of pain. This could be accomplished by a more general description of the LH and how the repertoire of behaviors associated with the LH are relevant to pain responses. For example, the section of text: "While the LH circuits controlling food intake and reward have received intense focus over the past several years, those governing nociception have gone understudied by comparison. Thus, with its diverse array of neuronal populations, uncovering genetically defined LH circuits that regulate pain may bring forth a novel therapeutic target" should be expanded upon somewhat and include some key references. At the moment, there are no references here.

– Also, the description of the role of the LH in pain in the introduction is sparse and should include a slightly more detailed description of the literature with key references. This can be included either as part of the introduction or in the discussion. Also, there are some key references missing here. An example includes a set of papers describing a role for an LH substance P projection to the PAG in mediating antinociception (Holden and Naleway, 2001; Holden et al., 2002; Pizzi and Holden, 2008; Pizzi et al., 2009)

*Reviewer #4:*

The miniscope calcium imaging data are excellent. For all imaging studies, summary statistics on which cells or how many cells are from which specific animal should be graphically shown somewhere. From the current display one cannot tell if the active and inhibited cells are from just one animal or are observed in all animals. It would beneficial to see several "per animal" metrics, i.e. showing the data in Figure 1 E and I not as a merge of all neurons across animals but for each individual mouse, as well as show the proportion of active and inhibited cells per subject.

1. The title states "sensory and affective pain", which I read initially as the paper would present data on the dichotomy of the sensory and emotional qualities of pain perception, as originally defined by Melzack 1968. However, the assays used (primarily reflexive tests of hypersensitivity) measure only sensory aspects of behavior. In the text though (in comparison to the title), the authors use "affect" to refer to "pain suppressed behaviors" / "pain induced co-morbidities", which I agree are partly captured by the nest building assay. The title to more accurately reflect the data, and remove "affective pain".

2. The miniscope data find two Clusters of cell-types functionally defined by being active or inhibited to noxious stimuli. This result is nice. However, it is disconnected from the remainder of the data, and as currently presented is a very elaborate, but far superior, Fos-like experiment showing that some cells are active while others are not, but it is not clear how and to what function these cell-types contribute to results presented after Figure 1. To avoid a difficult and time-consuming miniscope experiment to image projection neurons to PAG vs. habenula, I suggest a tracing-topology study to link this data to the other very important and interesting results in Figure 6 on differential results seen in the LH projections to the PAG vs habenula. For example, injection of a retrograde-AAV-FlpO recombinase into the PAG (and in a separate cohort, injection into habenula) of Pv-Cre mice with a Cre-ON/Flp-On intersectional fluorophore into the hypothalamus plus a noxious stimulus to induce FOS in these cells. Several conclusions could be drawn to compliment the miniscope functional data and the behavioral data, namely 1) are there collateral projections from LH◊PAG that also project to the habenula; 2) what proportion of these projection cells are nociceptive (FOS+); 3) are there functionally distinct cell-types (e.g. Clusters 1 and 2) that preferentially send projections to PAG or habenula (i.e. Fluorphore+ and FOS+/FOS-). This type of minor experiment might also provide additional insight to understand the two activity Clusters and how the bulk optogenetic/chemogenetic activation of LH^Pv^ vlPAG neurons is antinociceptive when this experimental design "turns on" the nociception-inhibited Cluster-2.

3. I would consider not referring to the 7-day timepoint post Spared Nerve Injury, a model of "chronic neuropathic pain". This is still an acute surgical neuropathy model, whereas most "chronic" designations should be reserved for the 3+ week timepoint, at a minimum.

4. No axonal terminals can be seen in the image of Figure 4A nor Supp Figure 3. Are there other images from these mice illustrating that this projection connects with this more posterior portion of the vlPAG? The group's prior 2019 paper shows dense LH^PV^ innervation of the very anterior PAG (superocularmotor region), in contrast to some fiber placements in this article, which are almost 1.0 mm apart. In addition to the new images to confirm axons under the fiber tracts, I would request that images and quantification be provided for LH-PV axon densities across the anterior-posterior axis of the PAG. This will also be very helpful for the reader to link the past work with the current manuscript, as well as make sense of the choice of A-P coordinates for the fibers (-4.0 and -4.8 [Leptin-Cre study]) and for the RABV tracing experiment (which was done at the anterior -3.8 coordinate which was also used in the 2019 paper). Alternatively, performing a patch-clamp experiment (as done by the group in the 2019 paper) of Vglut2 or Vgat PAG neurons in the posterior PAG would confirm this connection, since the provided images do not show any axons in this region.

5. Even though it was stated by the authors that it created some clutter, I would still suggest to show all individual dots and lines for behavior throughout the figures (perhaps make the lines 50% transparent)

*Reviewer #5:*

In 2019 the Aponte lab (10.1038/s41598-019-48537-y) reported that a small cluster of lateral hypothalamic neurons that express the calcium-binding protein parvalbumin (LH-PV neurons) modulate nociception in mice. They showed that photostimulation of these neurons suppresses nociception to an acute, noxious thermal stimulus, and that photoinhibition potentiates thermal nociception. They also showed that these neurons form functional excitatory synapses on neurons in the ventrolateral periaqueductal gray (vlPAG), and that photostimulation of these axons mediates antinociception. Finally, they showed that the anti-nociceptive effect appears to occur independently of opioidergic mechanisms. Many of these findings are replicated here.

In the present study they add to these findings by demonstrating, with the use of calcium imaging from behaving mice, that the LH-PV neurons respond to noxious stimuli. They also demonstrate that projections of these neurons to vlPAG affect both sensory and affective aspects of pain, whereas projections to the habenula appear to affect only the affective/aversive components.

The conclusions of this paper are mostly supported by the data, but some detailed aspects could to be clarified (laid out below).

1. Some of the effects reported appear rather small, and some reported differences might be driven by outliers. For example, data in Figure 1 d,g,h,k,l represent changes smaller than 2 standard deviations, or less than one Z score. Data in Figure 4i suggest very small effects on paw withdrawal thresholds. A consideration of whether these small changes are functionally meaningful would be particularly useful. Differences depicted in data in Figure 1 o,p,q appear to driven by a small number of outliers; even if these data survive tests of statistical outliers, one wonders why the vast majority of experiments show no differences.

Related to this, it would be useful to know what criteria were used to ensure that parametric analyses are appropriate. It is not clear why, in some comparisons, both Bonferroni and Dunnett's multiple comparisons are used on the same datasets. And, depicting variances as confidence intervals, instead of SEM, will likely be more informative.

Sample sizes are quite small. Although a large number of neurons are depicted, they were collected from a small number (e.g. 3) mice. At the very least, showing data from individual mice (instead of pooling data from all mice) will help determine how reproducible the results are.

2. Some neurons appear to increase their activity in response to stimuli, whereas others decrease their activity (Figure 1). It would be informative if the authors discuss this intriguing finding.

3. The doses of morphine that were effective appear rather high (Figure 7). Discuss?

4. Do individual LH-PV neurons project to both vlPAG and habenula? If so, can we exclude the possibility that terminal photostimulation antidromically activated LH neurons and their unintended axonal targets?

If these are independent projections, can the authors discuss how the LH projections to vlPAG vs to habenula might be regulated or balanced during different states?

[Editors' note: further revisions were suggested prior to acceptance, as described below.]

Thank you for resubmitting your work entitled "An excitatory lateral hypothalamic circuit orchestrating pain in mice" for further consideration by *eLife*. Your revised article has been evaluated by Michael Taffe as the Senior Editor and a Reviewing Editor.

The manuscript has been improved but there are some remaining issues that need to be addressed, as outlined below:

The revisions have exacerbated a problem in terminology. Pain is a percept. Noxious stimulation is a stimulus. Pain behavior is a package of skeletal muscle-mediated behaviors that ambiguously may include (or not) autonomic reactions. Thus, the title is problematic – you have little information on the pain percept (CPP and that is it). The bulk of your data speaks to pain behavior. And stimuli are not painful until proven so. Stimuli should be described as noxious (if they are indeed in that range).

Many of these issues converge in the sentence "As such, rodent studies searching for new pain interventions have begun to investigate ethological behaviors like nesting that are suppressed by pain (e.g., forgoing standard life activities) as well as the affective/emotional component of pain with assays of pain-induced aversion (e.g., comorbid depression) (5, 11, 12) identifying specific brain pathways capable of managing these multiple components of chronic pain and developing strategies for targeting them for translational use will advance the search for novel pain therapies."

Possible changes include: new pain interventions/therapies → new analgesic interventions, therapies; suppressed by pain → suppressed during pain behavior; assays of pain-induced aversion → assays of noxious stimulus-induced aversion; components of chronic pain → components of chronic pain behavior.

The suggestion that "the bulk activation of LH^PV^ neurons with optogenetics or chemogenetics likely activates more than the necessary number of them (hot responders + cold responders) to evoke uniform antinociception effects in a given assay" is not clear. Please explain how this could work. Related to this, if the responses to the noxious stimulation are so easily dismissed in favor of "bulk activation," what are the implications, the import, if any, of these responses? If without import, then why are they shown?

Typo in line 274 "since-cell".

---

## [Author Response]

Essential revisions:– Discussion of the inhibited vs excited LH neurons.– Responses of single neurons to multiple stimuli – coordinated or independent.

First, we decided to address the comments regarding (a) inhibited versus excited LH^PV^ neurons and (b) responses of individual LH^PV^ neurons to multiple stimuli (coordinated or independent) together, since they are closely related. For this, we performed cell registration across the five calcium imaging sessions (Figure 1—figure supplement 2). We found 33 neurons detected in both the hotplate and coldplate sessions in which we examined time-locked responses. Of those 33 neurons, only 6 neurons remained in a particular cluster in both sessions (e.g., cluster 1 in both the coldplate and hotplate), and an additional 2 neurons switched to the opposite cluster. Therefore, it seems likely that, in general, the responses of LH^PV^ neurons are either not consistent over time or dependent on the type of stimulus applied. As such, we think that the bulk activation of LH^PV^ neurons with optogenetics or chemogenetics likely activates more than the necessary number of them (hot responders + cold responders) to evoke uniform antinociception effects in a given assay, as opposed to having populations of LH^PV^ neurons that are exclusively pronociceptive or antinociceptive, the effects of which could be potentially diluted during bulk activation of these neurons. We have added a narrative for these points in the Discussion section (Discussion, page 23 – 24, lines 459 – 476).

– Are there dual projections to PAG and LHab (if no data, then a thorough acknowledgment and discussion of this issue)?

It is still unknown whether LH^PV^ axonal projections to their target regions follow a one-to-one or one-to-many architecture. This is certainly an important question that has yet to be determined. We thank the reviewers for suggesting various ways to address this point. However, these concerns seem to have mainly arisen from the assumption that certain LH^PV^ neurons consistently comprise the cluster 1 or cluster 2 archetype. Our cell registration analysis now demonstrates that this is not the case. Furthermore, our behavioral data suggest that these might be independent populations since we did not observe aversive-like effects during LH^PV^→vlPAG stimulation or antinociception during LH^PV^→LHb stimulation. Future experiments will be needed to firmly draw this conclusion, combined with elucidating the functional roles of each of these potentially independent populations. Of note, we have added some significant discussion regarding this issue (Discussion, page 21 – 22, lines 426 – 433).

– Concerns over mixed effects of Arch on synaptic transmission.

We indeed acknowledge the findings of Mahn et al. 2016 on silencing thalamocortical synapses with archaerhodopsin (eArch3.0). This study is commonly cited to discourage the use of eArch3.0 at axonal projections. We are certainly aware of the biophysical limitations of optogenetic inhibition at terminals. However, during that study Mahn and colleagues concluded that halorhodopsin (eNpHR3.0) is the most suitable tool for silencing synaptic terminals even though eNpHR3.0 also shows strong light-off rebound responses. Thus, when outlining our experiments, we checked whether other rigorous studies in the hypothalamus and other brain regions used eArch3.0 as silencer. For example, Jennings et al., 2013 used eArch3.0 to inhibit BNST^VGAT^ neurons projecting to the lateral hypothalamus. While their validation experiments tested 5 s photoinhibition, the behavioral experiments used 10 min of constant photoinhibition. This robustly suppressed feeding behavior in food-deprived mice without affecting food intake during the light-off periods. Of note, their findings have been the precedent of many subsequent impactful primary research and review articles. Another remarkable example is a study showing the recordings of postsynaptic lateral/ventrolateral PAG (l/vlPAG) neurons while photoinhibiting ArchT-expressing dmPFC terminals over a duration of 3 min (Rozeske et al., 2018). Please notice that this was the same duration of photoinhibition that we used in our present study. Additionally, Rozeske and colleagues observed a sustained reduction in l/vlPAG post-synaptic firing rates over the duration of photoinhibition in over half of the recorded neurons (probably due to some neurons that were not connected), and this finding is strengthened by the inclusion of a channelrhodopsin (ChR2) group which showed the opposite direction of effect. In further support, their behavioral experiments showed that photostimulation and photoinhibition of the mPFC to l/vlPAG pathway produced opposing effects in a contextual fear discrimination paradigm, whereby photostimulation and photoinhibition decreased and increased freezing behavior, respectively. Collectively, these studies demonstrate that archaerhodopsin activation at axonal terminals silences synaptic transmission and evokes the predicted overall inhibitory effect on neuronal circuits *in vivo* and during behavior.

Once more, we would like to emphasize the biophysical constraints of photoinhibition at terminals but also cite a few other points from our own findings. First, we demonstrated that archaerhodopsin works when used for LH^PV^ somatic inhibition, both during electrophysiological and behavioral experiments (Siemian et al., 2019). Second, archaerhodopsin and channelrhodopsin manipulations evoked opposite behavioral effects when used in either LH^PV^ somas or their axonal terminals in the vlPAG. Furthermore, these effects mirror what we observed during chemogenetic manipulations of these neurons. Together, we hope that the cited previous studies *in vivo* using archaerhodopsin at axonal terminals in combination with the consistency of our current behavioral findings are sufficient to ease concerns regarding this issue. We have added a discussion for this point and cited both studies Mahn et al. 2016 and Rozeske et al., 2018 to highlight our awareness of this issue (Discussion, page 20, lines 403 – 408).

– Persistent over chronic is preferred for the time scales used here.– Small sample sizes from low numbers of animals and small effects make the presentation of raw data points all the more desirable.

We thank the reviewers for these suggestions and understand their preference for displaying raw data points. Thus, we have changed the word “chronic” to “persistent” when discussing our experimental findings of pain models lasting less than 21 days. Additionally, for the functional imaging experiments, we have plotted the data for the hotplate and coldplate assays for each individual mouse, along with the contribution from each mouse to cluster 1 and 2 during both tests. We also added the individual data points for the vast majority of behavioral experiments.

– Flesh out intro beyond to introduce the component reactions to noxious stimulation that you will discuss – nocifensive, suppression of everyday activities and the intersection with aversion that eventually is discussed with respect to both LH and LHab.

We appreciate the reviewer’s helpful recommendations to improve our Introduction. We have expanded this section by both commenting on such components and providing more background on LH^PV^ neurons and other key studies that examined the role of the lateral hypothalamus in pain (Introduction, page 3-4, lines 34 – 41, 46 – 51, 56 – 61).

Please see below our point-by-point responses to the individual reviewers’ comments. We sincerely appreciate the reviewers’ valuable comments and suggestions to improve the quality of our work, and we hope that these revisions will now make our work suitable for publication at *eLife*.

Reviewer #1:This is an interesting study from the Aponte lab that builds on their prior work that identified lateral hypothalamic parvalbumin-expressing, glutamatergic neurons (LH^PV^ neurons) as regulators of pain behaviors in mice. Here, the authors explore how painful stimuli impact activity of the LH^PV^ neurons in vivo, and further examine the impact of manipulating activity of these neurons in the context of inflammatory and neuropathic pain, as well as the effects on affective aspects of pain.Using miniscope recordings, the authors find that there are two main populations of LH^PV^ neurons – the majority of which are activated in association noxious stimulation, but a subset that are inhibited.Studies using chemogenetics and optogentics, all with appropriate control groups, define clearly the effects of activating these LH^PV^ neurons, and show that the send important functional projections to the periaqueductal grey.The manuscript misses a potentially important opportunity in defining what these different classes (pain-activated vs inhibited) neurons in the LH do, but there is still much that we learn here.One technical issue that needs to be addressed is around the use of ArchT for inhibition of LH→PAG terminals, as ArchT can have complicated actions at terminals.The authors identify differences in projections to the lateral habenula vs. PAG, which are interesting and suggest different functional outputs. It will be important to understand if these are unique cell populations, or just dual projections from the same population of neurons in the LH.Overall, this represents a significant contribution to the field. I commend the authors on the thorough characterization of their behaviors and detailed exploration of the optogenetic stimulation frequency space – which is largely missing in most papers. There are a number of technical questions that should be addressed to further strengthen the manuscript, and it is worth considering whether some of the data presented here would be better reserved for a future publication to make the present manuscript more focused on the main points outlined in the manuscript title.

We appreciate the reviewer’s enthusiastic assessment of our work and helpful suggestions.

Comments for the authors:1. The imaging studies in figure 1 are intriguing and provide some very interesting insights – but the presentation raises a number of questions.a. Overall – the data show that there are LH^PV^ neurons that seem to show increased activity to heat and cold, and some that show decreased activity. The authors call this cluster 1 and 2 respectively. This is interesting, and points to the importance of using the miniscope approach to get cellular level resolution, as it appears that simple fiber photometry may have shown no change in activity in response to these stimuli. This should be highlighted.

We have added a narrative in the Discussion section to emphasize the advantage of using a single-photon miniscope, which enables single-cell resolution analysis (Discussion, page 19, lines 366 – 371).

b. Similar trends are seen for hot plate and cold plate. If you look at the single cell level – are neurons that classify as cluster 1 for hot plate also in cluster 1 for cold plate? Presumably the same mice were tested for both assays. This is not clear.

We apologize for this ambiguity. The same three mice were indeed used for both assays. Per request, we used the CellReg pipeline to register cells across sessions. While a fair number of the same cells overall were detected during the hotplate and coldplate tests (*n* = 33), only 6 neurons remained in the same cluster between both sessions. We also observed 2 additional neurons that flipped from cluster 1 in the hotplate test to cluster 2 in the coldplate test. Thus, it appears that largely different populations of LH^PV^ neurons are modulated by hot and cold noxious stimuli. This information has been added together with mouse-by-mouse breakdown data in Figure 1—figure supplement 2.

c. The z scores show something rather interesting – there is clearly a period of suppressed activity in cluster 1 cells and increased activity in cluster 2 cells preceding contact with the plate. This is evident also in the "control" group. What do the authors think of this? The inclusion of the control groups here was absolutely critical, as it shows that there are indeed differences. Related – the post-hoc analysis states significant differences are illustrated by the bold line – but this is not easy to see. Am I correct in seeing that there is no significant differences in cluster 2 for the cold stimulus vs. control? The manuscript states significant differences, but the posthoc either does not support this or it is not clear from the bold shading of the line in Figure 1l.

We think that the suppressed or increased activity preceding contact with the surfaces may be related to the nature of the Z-score calculation. What we show in the figures is the Z-score extracted from the entire session-long trace of these neurons, averaged across trials to create the peri-stimulus trace. A neuron that greatly increases in activity in confined epochs throughout the test (e.g., when contacting the hotplate) will by rule lower its resting Z-score baseline into negative values. The opposite would be true for neurons that presumably have some baseline amount of activity and pause for confined epochs during the test; the resting Z-score will be pushed into positive values. A resting Z-score of zero would indicate a neuron that either did not have much of a dynamic range during that session or had relatively the same amount of increases and decreases in fluorescence throughout the session. It has become popular to normalize the pre-stimulus period to zero, but we chose not to do this here to depict the traces in a less-processed form. Z-scoring the extracted fluorescent traces was important to normalize responses across neurons (some brighter than others), but to re-zero the Z-score traces would cause the neurons that had a lower resting Z-score to be overweighted in the Z-score space.

Furthermore, the fact that these pre-stimulus phenomena occurred on control trials, but then increase/decrease to around 0 suggests that perhaps there was some expectation of the mice to feel something noxious upon contact with the surface. However, despite this increase from pre-stimulus to post-stimulus during control trials, the neurons do not increase/decrease their activity to the same degree as the noxious trials.

Addressing the significant differences illustrated by bold lines for cluster 2 in Figure 1l—we apologize for this oversight as we realized that the bold segments were not shown on the submitted version of this figure. We corrected the panel in the revised version.

d. The cluster 1 neurons are more prevalent, and the authors comment that the response magnitude is larger than those for cluster 2. This should be supported by statistical comparisons validating this claim.

We greatly appreciate the reviewer’s interest in these data. After rereading our discussion on the response magnitude of each cluster, we realized that our discussion was more appropriate for the initially submitted version of this figure. For that previous version we used unsupervised k-means clustering instead of the thresholding method used in the full submission that was sent to reviewers. Since inclusion into cluster 1 or cluster 2 was defined at thresholds at equivalent absolute values on either side of 0 (e.g., ± 1 Z-score), this causes the magnitude of the clusters to be very similar. Thus, we have removed the section of the sentence comparing the magnitudes between clusters.

e. For the formalin data – given the results from the hot and cold plate, it is difficult to know how to interpret these findings. Were these cells recorded here specifically cluster 1 cells as determined from the hot and cold plate? Again, were these the same mice tested for hot and cold plate? An analysis of these data for formalin separately for neurons defined as cluster 1 and 2 for hot/cold would be much more informative than including all together.

We apologize for this ambiguity. The same three mice were used for the hotplate, coldplate, and formalin tests. The neurons reported for the formalin test are all recorded neurons, not specifically cluster 1 or 2 as determined from the hot and cold plate experiments. As stated above we found only 6 neurons that consistently classified as cluster 1 or 2 across the hot and cold plate tests, suggesting that the responses of these neurons are either dependent on the noxious stimulus or variable over time.

That being said, when we performed CellReg analysis within each mouse across all five tests (i.e., hotplate, coldplate, contralateral formalin, ipsilateral formalin, no injection) we found that neurons were predominantly only detected in 1 session, and that only 1 neuron was detected in all 5 sessions (please see the pie chart in panel g of Figure 1 —figure supplement 2). Thus, our current data again suggest that the neuronal responses are either (a) determined on an individual basis according to the stimulus, (b) variable across time due to GCaMP-expressing cell turnover, or (c) that some combination of these factors occurred. In any case, our current data do not suggest that these neurons can consistently be labeled as “cluster 1” or “cluster 2” in one test and expect similar phenotypes in a different test. Future work should examine the responses to multiple different noxious stimuli within the same test, which would help to decrease the impact of cell turnover and determine whether different neurons in fact do respond to different types of stimuli.

f. The interpretation of formalin responses as "long-term" seems unwarranted. In fact, the response to formalin is ongoing. So – this would suggest sustained responses to ongoing nociception.

We appreciate the reviewer’s attention to phrasing on this section. We have changed the Results section to remove “long-term” (page 7, lines 122 – 123) and instead indicate that the neuronal activity changes by an ongoing stimulus (page 8, line 138).

g. Related – it would be of interest to analyze the formalin data as Ca^2+^ transients time-locked to nocifensive responses (flinching, lifting, licking). Did you collect videos of the mice that would allow this analysis? Are there also "cluster 1" and "cluster 2" type responses associated with flinching/licking?

This is indeed a great suggestion. We recorded the contralateral formalin behavioral sessions from two out of the three mice presented here. However, we did not proceed with undertaking the suggested type of analysis since the weight of the miniscope appeared to burden the neck muscles of the mice over the duration of the 45 min formalin test. During the first few minutes of the experiment the mice licked their paws as normal untethered mice do. However, they soon began to take breaks to rest their heads, during which time they did not lick their paws or otherwise perform standard formalin-associated behaviors. As such, we did not think that these behavioral events accurately represented the level of chemical irritation the mice were experiencing. For this reason, we simplified this analysis and binned the activity into the well-defined acute/interphase/inflammatory epochs.

For the sake of speculation, it seems fair to assume that at least some neurons would show time-locked responses. However, even in the hot/cold plate tests, the main increase in activity occurs upon contact with the surface, not necessarily upon paw withdrawal. As such, LH^PV^ neuronal activity may precede bouts of licking/flinching at times when the mouse is sensing the buildup of irritation but not yet performing the behavior. Interestingly, these would likely ‘look’ more like cluster 2 neurons when time-locked to the actual behavior, as the fluorescence would be lower post-behavior than pre-behavior. In future work we plan to optimize these types of assays for finer-grained integration of neuronal activity and spontaneous behavior.

2. The authors go on to study the impact of manipulating LH^PV^ activity on behaviors, but sort of ignore the fact that there are these two populations of neurons with respect to activity in response to noxious stimulation. It is unfortunate that the interesting findings of the two different clusters of LH^PV^ neurons – one inhibited by noxious stimulation and one activated – is not built upon here – but is largely ignored. The subsequent studies go about activating or inhibiting all LHPV neurons to test their effects on sensory and affective aspects of pain. The challenge with this approach, based on the imaging data, is that there is a mixture of activity patterns in response to noxious stimuli. For example – heat responses. As you are targeting LH^PV^ neurons globally – you will be hitting both cluster 1 and cluster 2 neurons. I know that it would be a huge task to sort out what is different about these two populations – but this caveat at least needs significant discussion. It would be very interesting to be able to determine what is different about the cluster 1 and cluster 2 neurons. This is not addressed here, and I think may be beyond the scope of this study. However – this should be discussed. Why are some neurons inhibited, some activated? Do these neurons have different projection targets?

We recognize this important point and greatly appreciate the reviewer’s understanding of the challenges to dissect the differences between these two populations. Moreover, we agree that presently, this is beyond the scope of our current study. However, we would like to acknowledge that there is some precedent for observing heterogeneity (increasing/decreasing cells) upon single-cell calcium imaging, but a singular dominant behavioral phenotype driven by bulk activation of all neurons. For example, Jennings et al., 2015 showed that LH^GABA^ neurons respond to food locations in a heterogeneous manner by either increasing or decreasing in activity (Figure 5), and further that while the neurons typically increase in activity during appetitive or consummatory behaviors (Figure 6), sizable populations were found decreasing in activity during such behaviors (Figure S7). Yet, when these neurons are bulk activated, mice eat voraciously (Figure 1-2) and when the neurons are ablated, mice eat less (Figure 3).

In the context of our study, we agree that it would be great to further subdivide our manipulations of LH^PV^ neurons on the basis of the cluster 1 and 2 responses. However, our analysis included with this revision in fact found that the recorded LH^PV^ neurons did not consistently remain in cluster 1 or 2 across the hot and coldplate tests, suggesting that their responses are either dependent upon the noxious stimulus or otherwise somewhat variable on an individual basis across testing. Nevertheless, future work using the same noxious stimuli over multiple sessions may indeed identify constant neurons that classify as cluster 1 or 2. In the event that neurons could be distinguished based on consistent response properties, then as stated, it would require several experiments to determine whether those would be identifiable on the basis of projection field, gene expression, tonic activity patterns, or some other factor, with the caveat that the distinguishing factor would enable us to selectively target one or both clusters for manipulation of neuronal activity using optogenetics or chemogenetics. Moreover, the small number of LH^PV^ neurons and their restricted location within the most lateral part of the lateral hypothalamus makes such “functional dissection” endeavors more challenging than for larger, more broadly distributed, and more overtly heterogeneous neuronal populations. However, we agree that this will be an important question to address in future work. Using retrograde Cre-dependent GCaMP from different output regions to record projection-specific LH^PV^ neurons is one method that comes to mind to elucidate whether these clusters can be segregated based on projection. We have added some additional discussion to take this single cell heterogeneity versus bulk manipulation conflict more into account (Discussion, page 23, lines 459 – 476).

3. Activating the LH^PV^ neurons leads to reductions in responses to painful stimuli, and reversal of pain-suppressed behaviors. Given that the majority of these neurons are activated by painful stimuli, what does this mean for the role of LH^PV^ neurons in pain? You allude to this briefly in the discussion, but this is the major question in my mind in terms of the major conclusions we can make from this nice study. Are LH^PV^ neurons mediating endogenous analgesia during painful stimuli? If so, then this could be mediating a DNIC type response. It would be very interesting to know if you can elicit DNIC in mice, and block this by inhibiting these LH^PV^ neurons.

Our current impression is that LH^PV^ neurons, at least those in cluster 1, become active in response to pain, thus temporarily inhibiting it. When this ability is lost, mice become more sensitive to pain, whereas when this is bolstered, mice become less sensitive to pain. Thus, we would speculate that these neurons are a source of endogenous analgesia. It does not appear to quite be a DNIC response since activation of LH^PV^ neurons does not cause pain (see Results of nesting experiment) or cause robust place aversion (particularly during stimulation of axonal terminals in the vlPAG, which did reduce pain responses). However, future investigations of the role of LH^PV^ neurons in the DNIC response could certainly be warranted.

4. The use of ArchT for photoinhibition of the LH^PV^ → vlPAG terminals is problematic given reports that activation of Arch actually produces increases in spontaneous synaptic release at some terminals. It is imperative that the authors characterize the effects of Arch activation on synaptic transmission (evoked and spontaneous) at this synapse to allow interpretation of the findings.

We indeed acknowledge the findings of Mahn et al. 2016 on silencing thalamocortical synapses with archaerhodopsin (eArch3.0). This study is commonly cited to discourage the use of eArch3.0 at axonal projections. We are certainly aware of the biophysical limitations of optogenetic inhibition at terminals. However, during that study Mahn and colleagues concluded that halorhodopsin (eNpHR3.0) is the most suitable tool for silencing synaptic terminals even though eNpHR3.0 also shows strong light-off rebound responses. Thus, when outlining our experiments, we checked whether other rigorous studies in the hypothalamus and other brain regions used eArch3.0 as silencer. For example, Jennings et al., 2013 used eArch3.0 to inhibit BNST^VGAT^ neurons projecting to the lateral hypothalamus. While their validation experiments tested 5 s photoinhibition, the behavioral experiments used 10 min of constant photoinhibition. This robustly suppressed feeding behavior in food-deprived mice without affecting food intake during the light-off periods. Of note, their findings have been the precedent of many subsequent impactful primary research and review articles. Another remarkable example is a study showing the recordings of postsynaptic lateral/ventrolateral PAG (l/vlPAG) neurons while photoinhibiting ArchT-expressing dmPFC terminals over a duration of 3 min (Rozeske et al., 2018). Please notice that this was the same duration of photoinhibition that we used in our present study. Additionally, Rozeske and colleagues observed a sustained reduction in l/vlPAG post-synaptic firing rates over the duration of photoinhibition in over half of the recorded neurons (probably due to some neurons that were not connected), and this finding is strengthened by the inclusion of a channelrhodopsin (ChR2) group which showed the opposite direction of effect. In further support, their behavioral experiments showed that photostimulation and photoinhibition of the mPFC to l/vlPAG pathway produced opposing effects in a contextual fear discrimination paradigm, whereby photostimulation and photoinhibition decreased and increased freezing behavior, respectively. Collectively, these studies demonstrate that archaerhodopsin activation at axonal terminals silences synaptic transmission and evokes the predicted overall inhibitory effect on neuronal circuits *in vivo* and during behavior.

Once more, we would like to emphasize the biophysical constraints of photoinhibition at terminals but also cite a few other points from our own findings. First, we demonstrated that archaerhodopsin works when used for LH^PV^ somatic inhibition, both during electrophysiological and behavioral experiments (Siemian et al., 2019). Second, archaerhodopsin and channelrhodopsin manipulations evoked opposite behavioral effects when used in either LH^PV^ somas or their axonal terminals in the vlPAG. Furthermore, these effects mirror what we observed during chemogenetic manipulations of these neurons. Together, we hope that the cited previous studies *in vivo* using archaerhodopsin at axonal terminals in combination with the consistency of our current behavioral findings are sufficient to ease concerns regarding this issue. We have added a discussion for this point and cited both studies Mahn et al. 2016 and Rozeske et al., 2018 to highlight our awareness of this issue (Discussion, page 20, lines 403 – 408).

5. The results re: the projections to the LHb are also very interesting. Are the neurons that project to the LHb and vlPAG distinct, or do individual neurons send process to both LHb and vlPAG? A double retrograde tracing experiment would expand the scope of what we learn here significantly.

It is still unknown whether LH^PV^ axonal projections to their target regions follow a one-to-one or one-to-many architecture. This is certainly an important question that has yet to be determined. We thank the reviewer for suggesting a way to address this point. However, our behavioral data suggest that these might be independent populations since we did not observe aversive-like effects during LH^PV^→vlPAG stimulation or antinociception during LH^PV^→LHb stimulation. Of note, our LH^PV^→LHb data also demonstrates that the anticonception evoked by activating the LH^PV^→vlPAG pathway was not due to antidromic stimulation effects. Future experiments will be needed to determine that these are indeed independent populations of LH^PV^ neurons. We have added some significant discussion regarding this issue (Discussion, page 21 – 22, lines 426 – 433).

Reviewer #2:This is a thorough, multipronged study which supports a role for parvalbumin-positive neurons in the lateral hypothalamus (LH-Pv+) in modulating the nocifensive and affective / motivational consequences of noxious stimulation and morphine suppression of the same. The authors provide evidence that both nocifensive and affective / motivational consequences are suppressed by activation of glutamatergic LH-PV+ neurons whereas activation of GABAergic LH projections (expressing the leptin receptor) result in place aversion.Experiments also demonstrate that LH-PV+ activation combines in an additive and at one ratio in a synergistic manner with morphine and that activation of LH-PV+ neurons reverses tolerance to morphine.The nociceptive-specific nature of the sensory responses is not convincing given that there is a comparison between stimulus and not-stimulus rather than between innocuous stimulus and noxious stimulus. The authors may argue that room temperature is a thermal stimulus and perhaps that is true, depending on the animal's skin temperature which was not measured. But a better comparison would be between a 45C or so stimulus and the 51C used. I am not suggesting that the authors add such expts rather I recommend a more careful interpretation of the data collected. This concern is strengthened by the response to ipsilateral formalin which raises the issue of whether the "responses" may be in fact correlates of efferent function involved in autonomic or state arousal. Yet, the remaining experiments mitigate this concern.

We thank the reviewer for these suggestions and agree that temperature gradient analysis could be used in future studies to determine whether LH^PV^ neurons are technically temperature- or pain-sensing neurons or signaling arousal states. Thus, we have adjusted our wording on the Results section for the functional imaging data by removing several instances of the word “noxious”.

In pairing the (idiosyncratically) preferred side with CNO-inhibition, the preference for either side goes to 50%. Thus inhibition of LH-PV+ neurons takes away side preference but does not produce side avoidance as is stated in lines 153-54.

We appreciate this distinction and understand that “avoidance” of one side technically indicates the outright preference (>50% time) for the other side, and not a lowered occupancy relative to baseline which is how we had used the term. Thus, the wording in this section has been rearranged to discuss the results in terms of losing preference for the initially preferred side (Results, page 10, lines 176 – 180).

Please jog the gray symbols in Figure 4 supp 1, 2 laterally so they can be seen.

As suggested, we have reformatted the symbols on these figures for better visualization of individual data points.

The LH-PV+ preference for vglut2 over vgat PAG neurons is by less than a factor of 3. Given what we know about the unimportance of absolute numbers of synapses in dictating the responses of lemniscal pathways in the thalamus, this result is interesting but with this amount of information uninterpretable. Tone this down.

Determining how LH^PV^ neurons regulate vlPAG microcircuitry is indeed a subject for our future studies. In our previous work and current study, we extensively discussed previous experiments in mice using selective chemogenetic manipulation of neuronal activity in the vlPAG. Such work demonstrated that glutamatergic or GABAergic neurons play opposing roles in nociception and defensive behaviors (Samineni et al., 2017; Tovote et al., 2016). Thus, we would speculate that glutamatergic LH^PV^ neurons modulate nociceptive processing by excitatory control of glutamatergic neurons in the vlPAG to attenuate nociception. Our findings that more vlPAG^VGLUT2+^ neurons (15%) than vlPAG^VGAT^ (7%) neurons are synaptically targeted by LH^PV^ neurons further support this idea. Alternatively, LH^PV^ neurons may function by activating inhibitory interneurons in the vlPAG that provide local inhibitory control of vlPAG^VGAT^ neurons to suppress nociception. Future experiments will use techniques such as ChR2-assisted circuit mapping (CRACM) from LH^PV^-ChR2^+^ axonal projections onto postsynaptic vlPAG neurons followed by single-cell RT-qPCR analysis to elucidate how LH^PV^ neurons regulate vlPAG microcircuitry. We have expanded this narrative in the revised version of the Discussion (Discussion, page 21, lines 419 – 423).

Reviewer #3:In this study, Siemian and co-authors significantly expand upon their previous work (Siemian et al., 2019) examining the role of parvalbumin-positive glutamatergic neurons in the LH (LH^PV^), and their projections to the vlPAG, in the modulation of pain responses. In their previous work, the authors showed that optogenetic activation or inhibition of LH^PV^ neurons suppresses or potentiates, respectively, responses to a noxious thermal stimulus and that this effect may be mediated by an excitatory projection to the vlPAG. The current manuscript is exciting in that it describes a much more detailed investigation of this circuit, using in vivo activity monitoring and manipulation (both chemo- and optogenetic) in multiple models of pain and pain-related behavior (both acute and chronic) as well as a neuroanatomical elucidation of this circuitry. Overall, this work uses a wide-spectrum of modern tools to address the circuit-basis of LH^PV^ modulation of pain states and pain-associated behavior. This well-executed work contributes to filling a significant gap in our understanding of the role of LH neurons in modulating pain responses and pain behavior.In their current work, the authors start by observing in vivo calcium signals with single cell resolution using GRIN-lens endomicroscopy of LH^PV^ neurons in PV-cre mice in several pain assays. They found that a subset of LH^PV^ neurons were activated by an acute thermal stimulus (hotplate), a much smaller subset were inhibited and large proportion did not respond. In another set of experiments, a subset of neurons were found to be activated by an acute noxious cold stimulus (coldplate), a small subset were inhibited and large proportion did not respond. They also found that some LH^PV^ neurons responded to formalin injection over longer time-scales. These experiments successful provide solid evidence that at least a subpopulation of LH^PV^ neurons respond to noxious stimuli which is a key finding. Less clear is how the heterogeneity observed in these responses may be linked to the circuit manipulations in the subsequent experiments, which engage the whole population of LH^PV^ neurons, and how specific LH^PV^ neurons are in modulating pain pathways.In the next series of experiments, the authors manipulate LH^PV^ neurons using chemogenetic and optogenetic methods in PV-cre mice, in a variety of standard pain assays. They first found that chemogenetic activation of LH^PV^ neurons suppressed pain responses, while inhibition of LH^PV^ neurons potentiated pain responses (increases and decreases in thermal pain thresholds respectively). In an interesting assay of homecage behavior, they further found that chemogenetic activation of LH^PV^ neurons prevented a pain-associated reduction in both nesting behavior and place aversion, which strengthens the argument that activation of LH^PV^ neurons is antinociceptive and seems to diminish pain-associated aversion. Finally, optogenetic activation of LH^PV^ neurons appeared to diminish thermal and inflammatory pain, consistent with their chemogenetic results. However, optogenetic activation of LH^PV^ neurons also produced mild place avoidance in an RTPP assay, somewhat at odds with their chemogenetic results. Overall, the chemo and optogenetic manipulation experiments succeeded in showing that activation of LH^PV^ neurons has antinociceptive effects.Next, the authors significantly extended their previous work by further probing the role of LH^PV^ projections to both the vlPAG and LHb. Optogenetic stimulation of LH^PV^ inputs to the vlPAG increased both acute thermal pain and more chronic neuropathic pain thresholds while inhibition decreased it, consistent with direct somatic manipulations. Interestingly, these manipulations did not appear to affect place preference leading to their examination of LH^PV^ projections to the LHb, activation of which was aversive, consistent with previous work showing that the broader population of LH-VGLUT2 to LHb pathway is aversive. To address whether excitatory or inhibitory vlPAG populations may be targeted by LH^PV^ neurons, the authors used rabies tracing and found that although a very small percentage of LH^PV^ neurons appeared to innervate the vlPAG, a greater proportion of excitatory than inhibitory neurons were targeted. Finally, the authors found that LH^PV^-induced antinociception enhanced and morphine-induced antinociception and rescues morphine tolerance.Overall these data are novel and suggest that even a very tiny cluster of LH neurons has a rather outsized role to play in pain modulation. These findings would contribute to our limited understanding of both the role of hypothalamic circuits in pain modulation and the intersection of reward, aversion and pain.Comments for the authors:– Although the in vivo GRIN lens experiments provide evidence that a subpopulation of LH^PV^ neurons respond to noxious stimuli, there's less clarity on the following questions:1. Does the subpopulation of LH^PV^ neurons activated by noxious thermal stimulation overlap with the subpopulation of neurons activated by noxious cold stimulation, ie is a specific subset of these neurons broadly tuned to respond to noxious thermal and cold stimuli or are they separate? The authors could provide more clarity on whether the cold/hot assays were done in the same animals and if not, provide some commentary on this question.

We apologize for this ambiguity. The same three mice were indeed used for both assays. For better clarification, we used the CellReg pipeline to register cells across the hotplate and coldplate sessions. While a fair number of the same cells overall were detected during the hotplate and coldplate tests (*n* = 33), only 6 neurons remained in the same cluster between both sessions. We also observed 2 additional neurons that flipped from cluster 1 in the hotplate test to cluster 2 in the coldplate test. Thus, it appears that largely different populations of LH^PV^ neurons are modulated by hot and cold noxious stimuli. This information has been added together with mouse-by-mouse breakdown data in Figure 1—figure supplement 2.

2. What is the degree of inter-animal variability in the data? Did some animals contribute more or less to the population(s) that responded to thermal/cold stimuli?

We appreciate the reviewer’s interest in these data. Thus, in combination with the response above, we have added individual heatmaps depicting neuronal activity per mouse over these two tests, along with pie charts showing the contribution of each mouse to each response cluster (Figure 1 —figure supplement 2).

3. Given that only a relatively small subpopulation of LH^PV^ neurons are activated by noxious stimuli, with another subpopulation inhibited, how does one interpret the chemo- and optogenetic data in which all LH^PV^ neurons are either activated or inhibited? This should be addressed as a caveat in the discussion.

We acknowledge this important point brought up by the reviewer. However, we would like to emphasize that there is some precedent for observing heterogeneity (increasing/decreasing cells) upon single-cell calcium imaging, but a singular dominant behavioral phenotype driven by bulk activation of all neurons. For example, Jennings et al., 2015 showed that LH^GABA^ neurons respond to food locations in a heterogeneous manner by either increasing or decreasing in activity (Figure 5), and further that while the neurons typically increase in activity during appetitive or consummatory behaviors (Figure 6), sizable populations were found decreasing in activity during such behaviors (Figure S7). Yet, when these neurons are bulk activated, mice eat voraciously (Figure 1-2) and when the neurons are ablated, mice eat less (Figure 3).

While a fair number of the same cells overall were detected during the hotplate and coldplate tests (*n* = 33), only 6 neurons remained in the same cluster between both sessions. We also observed 2 additional neurons that flipped from cluster 1 in the hotplate test to cluster 2 in the coldplate test. Thus, it appears that largely different populations of LH^PV^ neurons are modulated by hot and cold noxious stimuli. This information has been added together with mouse-by-mouse breakdown data in Figure 1 —figure supplement 2.

– The rabies tracing experiment is well-done and informative but is a somewhat indirect way of showing preferential innervation of different vlPAG populations. Directly addressing this is a technically difficult problem and perhaps the subject of a follow-up study but a discussion of the caveats would be helpful. One possible caveat in this experiment is that, for the sake of argument, a simply larger population of excitatory vlPAG neurons may skew the representation of retrogradely-labeled LH^PV^ neurons despite perhaps an equal rate of connectivity with excitatory and inhibitory neurons.

We sincerely appreciate the reviewer’s understanding of the technical challenges to determine the preferential targeting of LH^PV^ axonal projections onto neurons within the vlPAG. We agree that determining how LH^PV^ neurons regulate vlPAG microcircuitry will expand our understanding of how activation of this LH^PV^→vlPAG pathway modulates nociceptive responses to noxious stimuli. This is indeed a subject for our future studies which will require the use of techniques such as ChR2-assisted circuit mapping (CRACM) from LH^PV^-ChR2^+^ axonal projections onto postsynaptic vlPAG neurons followed by single-cell RT-qPCR analysis. As suggested, we have expanded this narrative in the revised version of the Discussion (Discussion, page 21, lines 419 – 423).

– Another outstanding question is whether the same population of LH^PV^ neurons are innervating both the vlPAG (antinociception) and the LHb (aversion) or if they are entirely different subpopulations. One way of approaching this experimentally is to inject two different conjugated CTBs, one in the vlPAG and the other in the LHb to determine if retrogradely-labeled LH^PV^ neurons overlap or not. If this experiment is not possible, addressing this in the discussion would be helpful.

It is still unknown whether LH^PV^ axonal projections to their target regions follow a one-to-one or one-to-many architecture. This is certainly an important question that has yet to be determined. We thank the reviewer for suggesting a way to address this point. We have tried the CTB approach on these neurons without success. However, we would like to highlight that our behavioral data suggest that these might be independent populations since we did not observe aversive-like effects during LH^PV^→vlPAG stimulation or antinociception during LH^PV^→LHb stimulation. Of note, our LH^PV^→LHb data also demonstrate that the antinociception evoked by activating the LH^PV^→vlPAG pathway was not due to antidromic stimulation effects. Future experiments will be needed to determine that these are indeed independent populations of LH^PV^ neurons. We have added some significant discussion regarding this issue (Discussion, page 21-22, lines 426 – 433).

– In the Discussion, the authors refer to LH^PV^ neurons as a whole in the context of pain modulation when their data actually implicates only a comparatively small subpopulation of these neurons in modulating pain responses, evidenced by their in vivo monitoring and rabies tracing. This language should be adjusted in the discussion given that LH^PV^ neurons are clearly not functionally monolithic. Furthermore, a short discussion of any known heterogeneity among LH^PV^ neurons would be useful here.

We apologize again for this ambiguity. As stated above, we used the CellReg pipeline to register cells across sessions and found that cluster 1 and cluster 2 neurons were variable across sessions. This information has been added together with mouse-by-mouse breakdown data in Figure 1—figure supplement 2. Thus, this argues against constant fractions of LH^PV^ neurons that can always be considered the antinociceptive or pronociceptive clusters, respectively. We would speculate that the response of each individual LH^PV^ neuron may be determined by several factors including the stimulus intensity or modality.

– Finally, in terms of the writing, the manuscript is overall quite well-written.However, the introduction could be improved. The authors start with a general description of pain and pain behavior and quickly transition to the role of the lateral hypothalamus in pain. It would be useful to make that transition more fluid and more detailed given that this is rationale for studying these cells in the context of pain. This could be accomplished by a more general description of the LH and how the repertoire of behaviors associated with the LH are relevant to pain responses. For example, the section of text: "While the LH circuits controlling food intake and reward have received intense focus over the past several years, those governing nociception have gone understudied by comparison. Thus, with its diverse array of neuronal populations, uncovering genetically defined LH circuits that regulate pain may bring forth a novel therapeutic target" should be expanded upon somewhat and include some key references. At the moment, there are no references here.

We appreciate the reviewer’s helpful recommendations to improve our Introduction. We have added several references surrounding that sentence and have introduced some prior work regarding LH^PV^ neurons to segue from the introduction of lateral hypothalamic circuits into the summary of the current study (pages 3 – 4, lines 46 – 61).

– Also, the description of the role of the LH in pain in the introduction is sparse and should include a slightly more detailed description of the literature with key references. This can be included either as part of the introduction or in the discussion. Also, there are some key references missing here. An example includes a set of papers describing a role for an LH substance P projection to the PAG in mediating antinociception (Holden and Naleway, 2001; Holden et al., 2002; Pizzi and Holden, 2008; Pizzi et al., 2009).

As suggested, we have expanded this part and added these citations to the Introduction.

Reviewer #4:The miniscope calcium imaging data are excellent. For all imaging studies, summary statistics on which cells or how many cells are from which specific animal should be graphically shown somewhere. From the current display one cannot tell if the active and inhibited cells are from just one animal or are observed in all animals. It would beneficial to see several "per animal" metrics, i.e. showing the data in Figure 1 E and I not as a merge of all neurons across animals but for each individual mouse, as well as show the proportion of active and inhibited cells per subject.1. The title states "sensory and affective pain", which I read initially as the paper would present data on the dichotomy of the sensory and emotional qualities of pain perception, as originally defined by Melzack 1968. However, the assays used (primarily reflexive tests of hypersensitivity) measure only sensory aspects of behavior. In the text though (in comparison to the title), the authors use "affect" to refer to "pain suppressed behaviors" / "pain induced co-morbidities", which I agree are partly captured by the nest building assay. The title to more accurately reflect the data, and remove "affective pain".

We understand the reviewer’s concern for our use of these terms. However, in addition to the nest building assay we would also suggest that the effects of LH^PV^ neuronal activation on formalin-induced changes in place preference also capture changes in affective pain; this has been interpreted as such by others in previous work (e.g., Alhadeff et al., 2018 Cell, Figure 4: “Hunger attenuates negative affective components of pain”) even though the same manipulation had a primary effect of reducing formalin-induced paw licking (measure of sensory pain; Figure 1: “Hunger attenuates response to inflammatory pain).

We acknowledge that several studies have demonstrated that affective pain can be selectively manipulated without having effects on sensory pain per se (e.g., Johansen et al., 2001; Corder et al., 2019). Yet, our work attempts to emphasize that simply treating sensory pain neither implies that (a) affective/motivational effects of pain are also improved nor that (b) normal behaviors are resumed. Thus, we wanted to highlight that we incorporated the nest building and place preference experiments to specifically examine these issues, but we acknowledge that LH^PV^ neuronal activation likely reduces sensory pain first, which then has secondary effects on affective pain. Clearly, pain is generally multifaceted and its different components interact substantially to evoke a physically painful, emotionally unpleasant experience. However, even some clinical studies have shown that certain manipulations can induce changes in sensory but not affective pain descriptors (Schilder et al., 2018). Thus, we think that to postulate that the effects of LH^PV^ neurons are *specific* for sensory pain would also not be technically accurate, given that the assays of affective pain did yield positive effects. To avoid confusion, we have changed the title to “An excitatory lateral hypothalamic circuit orchestrating pain in mice.”

2. The miniscope data find two Clusters of cell-types functionally defined by being active or inhibited to noxious stimuli. This result is nice. However, it is disconnected from the remainder of the data, and as currently presented is a very elaborate, but far superior, Fos-like experiment showing that some cells are active while others are not, but it is not clear how and to what function these cell-types contribute to results presented after Figure 1. To avoid a difficult and time-consuming miniscope experiment to image projection neurons to PAG vs. habenula, I suggest a tracing-topology study to link this data to the other very important and interesting results in Figure 6 on differential results seen in the LH projections to the PAG vs habenula. For example, injection of a retrograde-AAV-FlpO recombinase into the PAG (and in a separate cohort, injection into habenula) of Pv-Cre mice with a Cre-ON/Flp-On intersectional fluorophore into the hypothalamus plus a noxious stimulus to induce FOS in these cells. Several conclusions could be drawn to compliment the miniscope functional data and the behavioral data, namely 1) are there collateral projections from LH◊PAG that also project to the habenula; 2) what proportion of these projection cells are nociceptive (FOS+); 3) are there functionally distinct cell-types (e.g. Clusters 1 and 2) that preferentially send projections to PAG or habenula (i.e. Fluorphore+ and FOS+/FOS-). This type of minor experiment might also provide additional insight to understand the two activity Clusters and how the bulk optogenetic/chemogenetic activation of LH^Pv^ vlPAG neurons is antinociceptive when this experimental design "turns on" the nociception-inhibited Cluster-2.

We agree that this experiment would yield several interesting findings. However, it is based on the idea that cluster 1 or 2 neurons remain constant over time and across stimuli. For our revision, we performed cell registration across the five calcium imaging sessions (Figure 1—figure supplement 2). We found 33 neurons detected in both the hotplate and coldplate sessions in which we examined time-locked responses. Of those 33 neurons, only 6 neurons remained in a particular cluster in both sessions (e.g., cluster 1 in both the coldplate and hotplate), and an additional 2 neurons switched to the opposite cluster. Therefore, it seems likely that, in general, the responses of LH^PV^ neurons are either not consistent over time or dependent on the type of stimulus applied. Thus, FOS-induced cells would likely be different depending on the applied stimulus, which further limits the interpretability of the proposed experiment if cells cannot rigidly be classified into these cluster designations.

3. I would consider not referring to the 7-day timepoint post Spared Nerve Injury, a model of "chronic neuropathic pain". This is still an acute surgical neuropathy model, whereas most "chronic" designations should be reserved for the 3+ week timepoint, at a minimum.

We appreciate the reviewer’s attention to phrasing on this part. As suggested, we changed these occurrences to “persistent”.

4. No axonal terminals can be seen in the image of Figure 4A nor Supp Figure 3. Are there other images from these mice illustrating that this projection connects with this more posterior portion of the vlPAG? The group's prior 2019 paper shows dense LH^Pv^ innervation of the very anterior PAG (superocularmotor region), in contrast to some fiber placements in this article, which are almost 1.0 mm apart. In addition to the new images to confirm axons under the fiber tracts, I would request that images and quantification be provided for LH-PV axon densities across the anterior-posterior axis of the PAG. This will also be very helpful for the reader to link the past work with the current manuscript, as well as make sense of the choice of A-P coordinates for the fibers (-4.0 and -4.8 [Leptin-Cre study]) and for the RABV tracing experiment (which was done at the anterior -3.8 coordinate which was also used in the 2019 paper). Alternatively, performing a patch-clamp experiment (as done by the group in the 2019 paper) of Vglut2 or Vgat PAG neurons in the posterior PAG would confirm this connection, since the provided images do not show any axons in this region.

We apologize for this ambiguity. We wanted to depict a general example for fiber placement given that we have four different groups (i.e., ArchT + GFP control and ChR2 + GFP control). For better clarification, we have included images for LH^PV^ axonal projections in the vlPAG in the revised version.

Related to the vlPAG coordinates for the *Pvalb^Cre^* mice studies, the discrepancy between vlPAG targets of AP −3.87 in the previous study, AP −3.9 for RABV, and AP −4.0 for behavior in the current study was a trivial adjustment in our opinion (≤ 130 µm difference; smaller than standard virus spread). We made this adjustment in order to try to get the optical fibers to be closer to the center of the terminal field in the PAG. The LH^PV^ terminal field is quite longitudinally dispersed. Therefore, the difference of ∼0.1 mm will not make any difference and is not a basis for anterior/posterior delineation of the LH^PV^ terminal field. The requested axonal density study has fortunately already been done by Marco Celio’s group in this publication [Celio et al., 2013 J Comp Neurol. “Efferent connections of the parvalbumin-positive (PV1) nucleus in the lateral hypothalamus of rodents”]. Note especially Figure 9a-d (where axons are observed between AP −3.9 and AP −4.24) and the schematic in Figure 11. Therefore, the AP −4.0 mm coordinate is justified to use.

For the LH^LEPR^→PAG experiments, we implanted the optical fibers at (AP −4.8 mm) because LH^LEPR^ neurons project to a more posterior area of the PAG compared to the LH^PV^ axonal projections. Of note, we and others previously mapped the axonal projections of LH^LEPR^ neurons (Schiffino et al., 2019; Leinninger et al., 2009). We have included this narrative in the Results and Methods sections on the revised version for better clarification (Results, page 14, line 273; Methods, page 31, lines 556 – 558).

5. Even though it was stated by the authors that it created some clutter, I would still suggest to show all individual dots and lines for behavior throughout the figures (perhaps make the lines 50% transparent)

We have reformatted figures to show individual data points except for (i) the 5-min binned formalin paw licking graph, since the individual data points are plotted in the next panel, (ii) the morphine dose-response curves which contain 5 lines that overlap significantly already, and (iii) the isobolograms which are plotted as mean ± 95% CI.

Reviewer #5:In 2019 the Aponte lab (10.1038/s41598-019-48537-y) reported that a small cluster of lateral hypothalamic neurons that express the calcium-binding protein parvalbumin (LH-PV neurons) modulate nociception in mice. They showed that photostimulation of these neurons suppresses nociception to an acute, noxious thermal stimulus, and that photoinhibition potentiates thermal nociception. They also showed that these neurons form functional excitatory synapses on neurons in the ventrolateral periaqueductal gray (vlPAG), and that photostimulation of these axons mediates antinociception. Finally, they showed that the anti-nociceptive effect appears to occur independently of opioidergic mechanisms. Many of these findings are replicated here.In the present study they add to these findings by demonstrating, with the use of calcium imaging from behaving mice, that the LH-PV neurons respond to noxious stimuli. They also demonstrate that projections of these neurons to vlPAG affect both sensory and affective aspects of pain, whereas projections to the habenula appear to affect only the affective/aversive components.The conclusions of this paper are mostly supported by the data, but some detailed aspects could to be clarified (laid out below).1. Some of the effects reported appear rather small, and some reported differences might be driven by outliers. For example, data in Figure 1 d,g,h,k,l represent changes smaller than 2 standard deviations, or less than one Z score. Data in Figure 4i suggest very small effects on paw withdrawal thresholds. A consideration of whether these small changes are functionally meaningful would be particularly useful. Differences depicted in data in Figure 1 o,p,q appear to driven by a small number of outliers; even if these data survive tests of statistical outliers, one wonders why the vast majority of experiments show no differences.

We respectfully disagree with the statement “the vast majority of experiments show no difference.”

For Figure 1d, these sample traces depict some relatively large (2-3 SD) transients occurring after hotplate contacts, comparable to previous publications showing individual representative traces (see Jennings et al., 2015 Figure 4i).

For Figure 1g,h,k,l, please note that one criterion used to select cells into these clusters were that they achieved ± 1 Z-score, on average across trials, at some point following the stimulus (note the range of the red/blue values in the heatmaps plotting individual neuron activity, range ± 2.5 Z-score). When these traces are averaged within each cluster to plot the activity traces, since the time of the maximal Z-score was asynchronous across neurons, this has the effect of creating a constant Z-score value lower than the absolute maximum of any of the neurons during their time of highest activity.

For Figure 4i, we specifically acknowledged this small effect in the original text – “Due to the modest effects of LH^PV^→vlPAG activation on mechanical thresholds, we predicted that this pathway may be more effective during inflammatory than neuropathic pain conditions.”

Related to this, it would be useful to know what criteria were used to ensure that parametric analyses are appropriate. It is not clear why, in some comparisons, both Bonferroni and Dunnett's multiple comparisons are used on the same datasets. And, depicting variances as confidence intervals, instead of SEM, will likely be more informative.

We understand the reviewer’s concerns. Our original analysis used a two-way ANOVA, with the repeated measure factor of treatment (i.e., the same cell was tracked across phases of the formalin test). The QQ plot of residuals appeared mostly linear, with a few points at one tail indicating perhaps a slight left skew. Since a non-parametric version of a mixed-model (or even unmatched) two-way ANOVA is unavailable in our statistical software, we proceeded with this report since we felt the matching values over the test was a vital factor and more appropriate than switching to independent tests.

Nonetheless, we have conducted non-parametric analyses, comparing each phase of these formalin test data to control in independent tests (i.e., contralateral vs no injection and ipsilateral vs no injection) with Mann-Whitney non-parametric tests corrected for multiple comparisons with the Holm-Sidak method and found that the results from our two-way ANOVA remain. While contralateral formalin induced significant increases in calcium transients over no injection in the acute (*p* = 0.048), interphase (*p* = 0.0078) and inflammatory (*p* = 0.048) phases, ipsilateral formalin did not. We have substituted these comparisons in the revised version. Thus, the inclusion of the Bonferroni comparison has been removed from that analysis. It was a feature of a previous version of Prism to conduct post-hoc tests between overall group data but that appears to have been recently removed. It was not a critical piece of information regarding the interpretation of these data, and the Dunnett’s post-tests, which were more important, are now essentially replaced by the Mann-Whitney tests.

Of note, we have changed the error bars of the formalin calcium transient dataset to 95% confidence intervals.

Sample sizes are quite small. Although a large number of neurons are depicted, they were collected from a small number (e.g. 3) mice. At the very least, showing data from individual mice (instead of pooling data from all mice) will help determine how reproducible the results are.

The results from the individual mice are now shown in the new supplemental figure (Figure 1 —figure supplement 1), along with the contributions of each mouse to the cluster 1 and cluster 2 designations.

2. Some neurons appear to increase their activity in response to stimuli, whereas others decrease their activity (Figure 1). It would be informative if the authors discuss this intriguing finding.

We found 33 neurons detected in both the hotplate and coldplate sessions in which we examined time-locked responses. Of those 33 neurons, only 6 neurons remained in a particular cluster in both sessions (e.g., cluster 1 in both the coldplate and hotplate), and an additional 2 neurons switched to the opposite cluster. Therefore, it seems likely that, in general, the responses of LH^PV^ neurons are either not consistent over time or dependent on the type of stimulus applied. We have now commented on this finding in the discussion (Discussion, page 23 – 24, lines 459 – 476).

3. The doses of morphine that were effective appear rather high (Figure 7). Discuss?

The doses of morphine used in the current study (i.e., 3.2 – 32 mg/kg) for morphine-alone treatments, are a common range reported throughout the mice literature in assays of acute thermal pain. Of note, we used up to 32 mg/kg as it was necessary for us to obtain 100% MPE to accurately calculate the ED50 values.

1. Miller et al., 2011 Psychopharmacology (Berl) “Effects of morphine on pain-elicited and pain-suppressed behavior in CB1 knockout and wildtype mice” find a near-identical dose-response curve for i.p. morphine in the hotplate test (see Figure 1) as we report in the present study.

2. Neelakantan et al., 2015 Behav Pharmacol “Distinct interactions of cannabidiol and morphine in three nociceptive behavioral models in mice” find a near-identical dose-response curve for i.p. morphine in the hotplate test (see Figure 2g) as we report in the present study.

3. Stone et al., 2014 PLoS One *“*Morphine and clonidine combination therapy improves therapeutic window in mice: synergy in antinociceptive but not in sedative or cardiovascular effects” report < 50% MPE at 10 mg/kg i.p. in mice for the tail-flick assay.

4. Do individual LH-PV neurons project to both vlPAG and habenula? If so, can we exclude the possibility that terminal photostimulation antidromically activated LH neurons and their unintended axonal targets?If these are independent projections, can the authors discuss how the LH projections to vlPAG vs to habenula might be regulated or balanced during different states?

It is still unknown whether LH^PV^ axonal projections to their target regions follow a one-to-one or one-to-many architecture. This is certainly an important question that has yet to be determined. We thank the reviewers for suggesting various ways to address this point. However, these concerns seem to have mainly arisen from the assumption that certain LH^PV^ neurons consistently comprise the cluster 1 or cluster 2 archetype. Our cell registration analysis now demonstrates that this is not the case. Furthermore, our behavioral data suggest that these might be independent populations since we did not observe aversive-like effects during LH^PV^→vlPAG stimulation or antinociception during LH^PV^→LHb stimulation. Of note, we have added some significant discussion regarding this issue (Discussion, page 21 – 22, lines 426 – 433).

[Editors' note: further revisions were suggested prior to acceptance, as described below.]

The manuscript has been improved but there are some remaining issues that need to be addressed, as outlined below:The revisions have exacerbated a problem in terminology. Pain is a percept. Noxious stimulation is a stimulus. Pain behavior is a package of skeletal muscle-mediated behaviors that ambiguously may include (or not) autonomic reactions. Thus, the title is problematic – you have little information on the pain percept (CPP and that is it). The bulk of your data speaks to pain behavior. And stimuli are not painful until proven so. Stimuli should be described as noxious (if they are indeed in that range).Many of these issues converge in the sentence "As such, rodent studies searching for new pain interventions have begun to investigate ethological behaviors like nesting that are suppressed by pain (e.g., forgoing standard life activities) as well as the affective/emotional component of pain with assays of pain-induced aversion (e.g., comorbid depression) (5, 11, 12) identifying specific brain pathways capable of managing these multiple components of chronic pain and developing strategies for targeting them for translational use will advance the search for novel pain therapies."Possible changes include: new pain interventions/therapies → new analgesic interventions therapies; suppressed by pain → suppressed during pain behavior; assays of pain-induced aversion → assays of noxious stimulus-induced aversion; components of chronic pain → components of chronic pain behavior.

We appreciate the importance of semantics in this context and in the preclinical literature at large. We have adjusted the wording throughout the manuscript as suggested above when discussing the results from this study in mice. However, in cases where we allude to future potential treatments or discuss conditions in humans, we have left the word “pain” intact. In addition, we have changed the title to “An excitatory lateral hypothalamic circuit orchestrating pain behaviors in mice.”

The suggestion that "the bulk activation of LH^PV^ neurons with optogenetics or chemogenetics likely activates more than the necessary number of them (hot responders + cold responders) to evoke uniform antinociception effects in a given assay" is not clear. Please explain how this could work. Related to this, if the responses to the noxious stimulation are so easily dismissed in favor of "bulk activation," what are the implications, the import, if any, of these responses? If without import, then why are they shown?

We have revised this section of the discussion as follows:

“Therefore, it seems likely that, in general, the responses of LH^PV^ neurons are either not consistent over time or dependent on the type of stimulus applied. […] However, further work will be needed to elucidate how noxious stimuli are responded to and encoded by LH^PV^ neuronal activity.” (Discussion, page 24)

These findings are important for at least three reasons. First, they demonstrate that responding during noxious stimulation is part of the “natural repertoire” of LH^PV^ neurons. Thus, activating them via optogenetics or chemogenetics likely does not induce some primary effect totally unrelated to nociception (e.g., fear) which then has a secondary effect of suppressing nociception. Rather, suppressing nociception is likely directly affected via these manipulations. Second, they indicate that different types of stimuli engage different individual LH^PV^ neurons. As such, future investigations may link specific LH^PV^ neurons to specific noxious stimuli for targeted antinociception to that specific stimulus. For example, in an arthritis patient, reducing mechanical hypersensitivity (maladaptive) without affecting thermal sensing (normal) could be an ideal way to selectively manage pathological pain. Third, they replicate the broad phenomenon throughout the single-cell literature that even genetically similar neurons do not generally respond in a homogenous manner. We elaborated on the divergence of LH^VGAT^ neuronal responses to food-oriented behaviors in the discussion and rebuttal already. Similarly, although arcuate hypothalamic AGRP neurons are commonly discussed to uniformly decrease in neuronal activity upon access to food (when using fiber photometry; Su et al., 2017; Beutler et al., 2020; Mazzone et al., 2020), the only manuscript that directly measured AGRP neuronal activity using in vivo extracellular electrophysiology showed that ∼30% of AGRP neurons do not show a decrease in activity upon presentation of food/feeding (Mandelblat-Cerf et al., 2015), a finding also somewhat visible when using a single-photon miniscope to record AGRP neuronal activity (Betley et al., 2015). Likewise, neurons in the MnPO that control fluid homeostasis respond in a heterogeneous manner to oral fluid access or intragastric infusion (Zimmerman et al., 2019). Thus, we do not think our findings here on the responses of LH^PV^ neurons are at all discordant with the general findings of how neuronal populations respond to certain stimuli.